

# The western Andes at ~20–22°S: A contribution to the quantification of crustal shortening and kinematics of deformation.

Tania Habel[1], Martine Simoes[1], Robin Lacassin[1], Daniel Carrizo[2,3], German Aguilar[2]

[1] Université Paris Cité, Institut de physique du globe de Paris, CNRS, F-75005 Paris, France
[2] Advanced Mining Technology Center, Facultad de Ciencias Físicas y Matemáticas, Universidad de Chile, Avenida Tupper 2007, Santiago, Chile
[3] now at GeoEkun SpA, Santiago 7500593, Chile

*Correspondence to*: Martine Simoes (simoes@ipgp.fr)

**Abstract.** The Andes are an emblematic active Cordilleran orogen. Mountain-building in the Central Andes (~20°S) started by Late Cretaceous to Early Cenozoic along the subduction margin, and propagated eastward. In general, the structures sustaining the uplift of the western flank of the Andes are dismissed, and their contribution to mountain-building remains poorly constrained. Here, we focus on two sites along the western Andes at ~20–22°S, in the Atacama Desert, where structures are well exposed. We combine mapping from high-resolution satellite images with field observations and numerical trishear forward modeling to provide quantitative constraints on the kinematic evolution of the western Andes. Our results confirm the existence of two main structures, once our field observations are combined with regional data: (1) the Andean Basement Thrust, a west-vergent thrust system placing Andean Paleozoic basement over Mesozoic strata; and (2) a series of west-vergent thrusts pertaining to the West Andean Thrust System, deforming primarily Mesozoic units. Once restored, we estimate that both structures accommodate together at least ~6–9 km of shortening across the sole investigated ~7–17 km-wide field sites. This multi-kilometric shortening represents only a fraction of the total shortening accommodated along the whole western Andes. The timing of the main deformation recorded in the folded Mesozoic series can be bracketed between ~68 and ~29 Ma – and possibly between ~68 and ~44 Ma – from dated deformed geological layers, with a subsequent significant slowing-down of shortening rates. Even though negligible when compared to total shortening across the whole orogen, the contribution of the structures forming the West Andes has been relatively significant at the earliest stages of Andean mountain-building before deformation was transferred eastward.

## 1 Introduction

Along the western margin of South America (Figure 1), the oceanic Nazca plate plunges beneath the South American continent, with a present-day convergence rate of ~8 cm/yr at ~20ºS, according to the NUVEL-1A model (DeMets et al., 1994). The subduction megathrust absorbs most of this convergence in the form of large earthquakes (magnitude Mw ≥ 8). A small fraction of it – presently ~1 cm/yr at 20°S (e.g. Brooks et al., 2011; Norabuena et al., 1998) – contributes to the deformation of the



**Figure 1. Simplified geological and structural map of the western Central Andes at ~20–22°S**, Northern Chile (modified from Armijo et al., 2015), and average topographic profile (top; ve: vertical exaggeration). The two main structural ensembles are here the Marginal Block and the Western Cordillera. The Marginal Block encompasses the Coastal Cordillera and the longitudinal valley of the Atacama Bench (or Central Depression). The Western Cordillera includes the West Andean Thrust System (WATS), a basement high (Cordillera Domeyko), and the modern volcanic arc. A large part of the WATS is hidden beneath blanketing Cenozoic deposits and only outcrops in a few places. The Andean Basement Thrust (ABT) separates the WATS and the basement high of the Western Cordillera. The location of Figures 4 (Pinchal area) and 8 (Quebrada Blanca area) is given by black boxes. Inset: Location of the map (red box) within the Central Andes along the South American Continent. WAT: West Andean Thrust (after Armijo et al., 2015); Cz: Cenozoic; Mz: Mesozoic; Pz-Pc: Paleozoic and Precambrian.




upper plate over millions of years and to the formation of one of the largest reliefs at the Earth's surface: the Andean Cordilleras and the Altiplano-Puna plateau in between.

Andean mountain-building initiated by Late Cretaceous-Early Cenozoic along the western Andes of the Bolivian Orocline (between 16–22°S), and proceeded since then with the progressive eastward propagation of deformation onto the South American continent (e.g. Anderson et al., 2017; Armijo et al., 2015; Barnes et al., 2008; Barnes & Ehlers, 2009; Charrier et al., 2007; DeCelles et al., 2015; Eichelberger et al., 2013; Elger et al., 2005; Faccenna et al., 2017; Kley & Monaldi, 1998; McQuarrie et al., 2005; Oncken et al., 2006; Sheffels, 1990; and references therein). Most local and mountain-wide previous

studies have essentially focused on the structures of the Altiplano-Puna plateau and on those of the various cordilleras to the east, but the structures located along the western flank of the orogen have remained up to now relatively under-studied, and their contribution to the significant topographic relief (Figure 1) and crustal thickness (e.g. Allmendinger et al., 1990; Introcaso et al., 1992; Isacks, 1988) poorly understood.

In most classical models of Andean mountain-building, the western flank is described as a passive monoclinal-like crustal-

scale flexure (e.g., Isacks, 1988; Lamb, 2011, 2016; McQuarrie, 2002). However, in the late 1980's, Mpodozis and Ramos (1989) described west-vergent thrusting along the western Andean margin. Later, other authors also described various thrusts, mostly west-vergent (but not only), at several localities along the western Andean flank (e.g. Charrier et al., 2007; Farías et al., 2005; Fuentes et al., 2018; Garcia & Hérail, 2005; Martínez et al., 2021; Muñoz & Charrier, 1996; Victor et al., 2004), but they generally gave these thrusts a minor role in the building of the western flank of the orogen. Only further south, at the

latitude of Santiago de Chile (~33°30'S), a clear west-vergent fold-and-thrust-belt has been documented along the western Andes (Armijo et al., 2010; Riesner et al., 2017, 2018). This thrust belt emerges at the active San Ramon Fault in front of the capital city of Santiago de Chile, and has absorbed a significant amount of crustal shortening (Riesner et al., 2017, 2018).

At 33°30'S, the orogen is relatively younger and narrower than in the Bolivian Orocline ~1300 km further north. In contrast, at ~20–22°S, where the Andes-Altiplano system is much wider and structurally more complex, the contribution of structures

along the western Andes is probably small compared to the >300 km total shortening (e.g. Anderson et al., 2017; Barnes & Ehlers, 2009; Eichelberger et al., 2013; Elger et al., 2005; Faccenna et al., 2017; Kley & Monaldi, 1998; McQuarrie et al., 2005; Oncken et al., 2006; Sheffels, 1990) across the entire >650 km wide orogen, but their role at the start of orogenic building may have been significant (Armijo et al., 2015). One of the difficulties in better quantifying the contribution of these structures is that a large part of the deformation is hidden under blanketing mid-upper Cenozoic deposits and volcanics (Armijo et al.,

2015; Farías et al., 2005; SERNAGEOMIN, 2003; Victor et al., 2004). A quantitative analysis of this deformation and its kinematics is only possible at the few sites along the western flank where deformed Mesozoic series crop out and which are accessible despite the hostile desert conditions in North Chile.

In this study, we provide quantitative data to better constrain the geometry of structures, the shortening they accommodated and their kinematics of deformation over time in two of the few areas along the west Andean flank where the underlying

deformed Mesozoic layers are exposed (Figure 1). The Pinchal area, at ~21°30'S, exhibits a west-vergent thrust that brings the Paleozoic basement of the Cordillera Domeyko over folded Mesozoic units. In the Quebrada Blanca zone, ~80 km further





north, the excellent exposure of folded Mesozoic series allows for a more quantitative estimate of the shortening and of the timing of the main deformation episode. These two study areas only give a limited view on the deformation of the whole western Andean flank (Figure 1). Despite these limitations, we find that the shortening of these structures is multi-kilometric,
revealing that the contribution of the west Andean flank to Andean mountain building is not negligible. Additionally, we show that the main deformation recorded by the folded Mesozoic units occurred sometime between ~68 and ~29 Ma (and possibly between ~68 Ma and ~44 Ma), further emphasizing that these structures mostly participated to the early stages of mountain-building.

## 2 Geological Context of the Andes (~20–22°S)

### 2.1 General geological framework

At ~20–22°S, the Central Andean mountain-belt is characterized by its largest width (>650 km), highest average elevation (~4–4.5 km above sea level, hereafter a.s.l., Figure 1), thickest crust (70–80 km, e.g. Heit et al., 2007; Tassara et al., 2006; Wölbern et al., 2009; Yuan et al., 2000; Zandt et al., 1994) and greatest total shortening (>300 km, e.g. Anderson et al., 2017; Barnes & Ehlers, 2009; Eichelberger et al., 2013; Elger et al., 2005; Kley & Monaldi, 1998; McQuarrie et al., 2005; Sheffels,
1990). Here, from west to east, the Andean margin is constituted of: (1) the subduction margin, including the Peru–Chile Trench, the oceanward forearc, and the Coastal Cordillera that reaches altitudes >1 km and that corresponds to the former Mesozoic volcanic arc; (2) the Atacama Bench or Central Depression, at an altitude of ~1 km, corresponding to a modern continental forearc basin, well expressed in the morphology and topography of North Chile; and (3) the strictly speaking Andean orogen (e.g. Charrier et al., 2007; McQuarrie et al., 2005; Oncken et al., 2006). Following the terminology of Armijo
et al. (2010, 2015), the morpho-tectonic units located west of the Andean orogen constitute the Marginal Block (i.e. the oceanward forearc, the Coastal Cordillera and the Atacama Bench) (Figure 1).

At ~20–22°S latitude, the Andean orogen is composed, from west to east, of: (1) the Western Cordillera (Figure 1), including the Cordillera Domeyko and the modern volcanic arc (following here the terminology of e.g. Armijo et al., 2015; McQuarrie, 2002; Eichelberger et al., 2013; Garzione et al., 2017; Oncken et al., 2006); (2) the Altiplano Plateau, a high-elevation internally
drained low-relief basin; (3) the Eastern Cordillera; (4) the Interandean zone (or Cordillera Oriental); and (5) the Subandean ranges, east of which the South American craton underthrusts the Andes (e.g. Armijo et al., 2015; Isacks, 1988; McQuarrie et al., 2005; Oncken et al., 2012). The building of the Andean mountain-belt stricto sensu proceeded since the Late Cretaceous - Early Cenozoic at ~20–22ºS and was associated with crustal shortening and thickening (e.g. Amilibia et al., 2008; Andriessen & Reutter, 1994; Armijo et al., 2015; Arriagada et al. 2006; Barnes et al., 2008; Bascuñan et al., 2016; Charrier et al., 2007;
DeCelles et al., 2015; Faccenna et al., 2017; Henriquez et al., 2019; McQuarrie et al., 2005; Mpodozis et al., 2005; Oncken et al., 2006). Based on the regional syntheses by McQuarrie et al. (2005), Oncken et al. (2006), Charrier et al. (2007), Armijo et al. (2015), Garzione et al. (2017) and Horton (2018), the across-strike growth of the orogen is summarized as follows: (1) by Late Cretaceous, the Mesozoic arc and backarc basin (formed during the early Andean cycle) was located at the position of



the present-day forearc, and most of the Andes showed mainly flat topography; (2) by Late Cretaceous - Early Cenozoic,
orogenic growth initiated primarily along the western margin of the present-day Altiplano; (3) by ~45–30 Ma, shortening
vanished along the western flank of the Andes, and was transferred to the Eastern Cordillera; (4) by ~25 Ma, deformation
ended in the Eastern Cordillera and migrated to the Interandean Belt; (5) from ~10 Ma until present, deformation within the
Subandean Belt proceeded with the underthrusting of the Brazilian Craton beneath the Andes. It is therefore clear that the
Andean shortening started along the western Andes and subsequently propagated eastward, progressively enlarging the orogen
to form the different cordilleras and the Altiplano plateau in between.

## 2.2 Geological setting of the Western flank of the Andes at ~20–22°S

The Andean western flank is formed of three tectono-stratigraphic units at ~20–22°S, aside from the present-day volcanic arc.
Starting from the East (oldest and deepest units, exposed at high altitudes) to the West (youngest units, lower altitudes), these
are (Figure 1): (1) Andean basement consisting of metamorphic rocks of Precambrian and Paleozoic ages; (2) folded volcano-
sedimentary deposits of Mesozoic age (Triassic–Cretaceous), unconformably overlain by (3) less-deformed mid-upper
Cenozoic (Oligocene – Quaternary) volcanics and sedimentary cover. Magmatic intrusions locally alter these different units,
and are mostly Cenozoic (SERNAGEOMIN, 2003). This only pictures the first-order structuration of the western Andean
flank, as Mesozoic strata may be locally trapped in between two basement units, and Cenozoic layers may be unconformably
overlying older strata even to the east (Figure 1). Laterally, and in particular further south (i.e. south of the city of Calama,
~22°27'), the structural organization of the western flank of the Andes is more complex, and the description proposed here
does not directly apply.

### 2.2.1 Stratigraphic and geologic background

The pre-Andean basement rocks formed during the Late Proterozoic and Paleozoic, when the Amazonian craton was
progressively assembled from various terranes (e.g. Charrier et al., 2007; Lucassen et al., 2000; Ramos, 1988; Rapela et al.,
1998). At the end of this period of subduction and continental accretion, intensive magmatic activity (volcanism and major
granite intrusions) welded together the basement during the Late Carboniferous to Early Permian (Charrier et al., 2007; Ramos,
2008; Vergara & Thomas, 1984).
The Mesozoic deposits (Triassic to Cretaceous), found today along the west Andean flank, formed in a proto-Andean arc and
backarc basin system during the early period of the Andean cycle (e.g. Charrier et al., 2007; Mpodozis & Ramos, 1989).
Marine and continental sediments are interbedded with volcano-magmatic rocks (Aguilef et al., 2019; SERNAGEOMIN,
2003). These Mesozoic units attain locally thicknesses up to ≥10 km (e.g. Buchelt & Tellez, 1988; Charrier et al., 2007;
Mpodozis & Ramos, 1989).
A regional erosional surface called the Choja Pediplain (Galli-Olivier, 1967) developed during the Eocene to Early Oligocene
(~50–30 Ma) (e.g. Armijo et al., 2015; Victor et al., 2004 and references therein). Above this angular unconformity, the up to
~1600 m thick (Labbé et al., 2019) Cenozoic deposits of the Altos de Pica Formation (Galli & Dingman, 1962) are composed





of continental clastic sediments, interbedded with volcanic layers (Victor et al., 2004). The oldest documented age within the Altos de Pica Formation is of ~24–26 Ma from dated ignimbrites (Farías et al., 2005; Victor et al., 2004). From there, an age of ~27–29 Ma for the base of the formation is inferred regionally when extrapolated to the basal erosional surface. The youngest ignimbrites within the Altos de Pica Formation are dated at ~14–17 Ma (Middle Miocene) (Vergara & Thomas, 1984; Victor

et al., 2004). From there and from other younger dated ignimbrites (Baker, 1977; Vergara & Thomas, 1984), Victor et al. (2004) deduced from stratigraphic correlations that the development of the Altos de Pica Formation finished by ~5–7 Ma (Late Miocene) at ~20–22ºS.

### 2.2.2 Structural and kinematic context

The Paleozoic basement of the Western Cordillera is disrupted at places in the form of various basement highs boarded by

reverse faults (Figure 1), such as the Sierra del Medio to the east and the Sierra de Moreno to the west at ~22°S (e.g. Haschke & Günther, 2003; Henriquez et al., 2019; Puigdomenech et al., 2020; Tomlinson et al., 2001) – not to be confused with the north–south trending strike-slip Domeyko Fault System, also called West Fissure System (or Falla Oeste) (e.g. Charrier et al., 2007; Reutter et al., 1996; Tomlinson & Blanco, 1997a, 1997b), east and out of our field study area. At ~20-22°S, various maps describe west-vergent thrusts in overall structural continuity, bringing the Paleozoic basement westward over folded

Mesozoic units (Aguilef et al., 2019; Haschke & Günther, 2003; SERNAGEOMIN, 2003; Skarmenta & Marinovic, 1981). Using apatite fission track dating, Maksaev and Zentilli (1999) proposed significant exhumation of the basement units between 50 Ma and 30 Ma, possibly related to basement overthrusting. Older exhumation ages (Late Cretaceous to Early Cenozoic (U-Th)/He zircon and apatite ages) are however provided by Reiners et al. (2015) for the western Andean basement at ~21°42'S, but from only one sample and without modeling. Together, these ages indicate that data remain missing to better quantify the

exhumation, uplift and timing of deformation of the basement thrusts reported along this part of the Western Andean flank. Further west, a series of mostly west-vergent thrusts affecting the Mesozoic to Cenozoic series have been inferred from seismic profiles (Victor et al., 2004; Fuentes et al., 2018; Martinez et al., 2021). Victor et al. (2004) determined ~3 km of shortening of the syn-tectonic Altos de Pica Formation layers, but they did not take into account the deformation of the underlying more deformed Mesozoic units. Other authors propose limited shortening on these older deeper layers (Fuentes et al., 2018; Martinez

et al., 2021), but the poor quality of the seismic profiles at these depths renders these interpretations quite tenuous and disputable. Haschke and Günther (2003) estimated that >9 km of shortening across the western flank in the outcropping Sierra de Moreno area (~21°45'S) occurred since the Late Cretaceous to Eocene on a west- and east-verging thrust system. Whether these various faults are connected at depth onto an eastward dipping master fault (Victor et al., 2004; Haschke and Gunther, 2003; Armijo et al, 2015) or whether their are steeply dipping single planar faults reactivated from the earlier Andean basins

(Fuentes et al., 2018; Martinez et al., 2021) remains debated. It follows that even if published data document the existence of various faults along the western Andean front at ~20–22°S, their geometry, kinematics and total amount of shortening have not yet been satisfactorily evaluated.

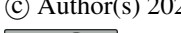



## 3 Data and Methods

Unconformable slightly deformed mid-upper Cenozoic clastic sediments and ignimbrites commonly hide the folded Mesozoic

layers and their contact with the basement (Figure 1). Field investigations are limited to the few sparse areas where the erosion of the Cenozoic cover has exposed the underlying structures (Aguilef et al., 2019; SERNAGEOMIN, 2003). In this study, we focus on two relatively accessible outcrop sites (Figure 1): (1) at ~21°30'S, where the Paleozoic basement thrusts over the Mesozoic (Skarmenta and Marinovic, 1981). This zone will be referred to as the Pinchal area (next to Cerro Pinchal, 4193 m a.s.l.). (2) At ~20°45'S, where folded Mesozoic units can be observed. This zone is hereafter named Quebrada Blanca area,

after its largest canyon.

### 3.1 Available Data

The most detailed - even though large-scale - existing geological map for the Pinchal area is the Quillagua map (1:250,000 scale, Skarmenta and Marinovic, 1981). For the Quebrada Blanca area, the recent Guatacondo map (1:100,000 scale, Blanco & Tomlinson, 2013) provides detailed and updated information on the stratigraphy and structure. There, the structure of the

folded Mesozoic rocks has been preliminarily mapped and qualitatively described by Blanco and Tomlinson (2013), Armijo et al. (2015) and Fuentes et al (2018).

Enhanced cartographic details can be deduced from high-resolution satellite imagery. We use Google Earth imagery (Landsat 7, DigitalGlobe) whose resolution varies from a few meters to a few tens of meters depending on the zones. In addition, this work benefits from very high-resolution imagery from the European Pléiades satellites. Using the MicMac software suite

(Rosu et al., 2014; Rupnik et al., 2016), we calculate high-resolution DEMs from tri-stereo Pléiades imagery, with a 0.5 m resolution. These DEMs are down-sampled to a resolution of 2m to enhance data treatment and calculations (e.g. stratigraphic projection and image processing). Relative vertical accuracy may reach ~1m, depending on local slope.

Field observations were acquired during two field surveys in March 2018 and January 2019. Difficult accessibility and field logistics in the remote and desert Pinchal area only allow detailed complementary field observations on a relatively limited

area. Observation points and the off-road track followed to reach our field site in the Pinchal area are provided as supplementary material.

### 3.2 Structural maps

To establish structural maps, we do 3D-mapping of stratigraphic layers, after Armijo et al., (2010) and Riesner et al. (2017). Layers are traced and correlated on Google Earth satellite images. The so-obtained georeferenced traces are projected on the

DEM-derived topographic map to obtain their altitude, and on geological maps for stratigraphic referencing. Field observations allow ground verifications and provide supplementary details, such as minor thrusts and folds, the observation of polarity criteria or local dip angles.



The approach used here is mainly limited by local geological complications. Continuous mapping of Mesozoic strata is locally complicated where incision of Cenozoic strata is limited, where magmatic intrusions and associated hydrothermalism alter the

structural geometries, where layers with no well-expressed bedding such as marls are present (ex: Pinchal area), or where landslides or recent sediment deposits hide the underlying deformation pattern. Therefore, geometrical observations and detailed mapping of the structures may be locally difficult, in some zones impossible. These difficulties cause some uncertainties in precisely correlating mapped layers, but only result in limited metric to decametric errors and do not modify our large-scale (km) results and interpretations.

**3.3 Structural cross-sections**

We use structural measurements, field observations and the obtained structural map to build cross-sections of the two investigated areas.

In the Pinchal area, because of limited canyon incision, marls, and frequent blanketing of the structures by Cenozoic cover, we build our structural cross-sections mainly from field observations (strike and dip angles, polarity criteria, first-order

stratigraphic column), with additional information taken from satellite imagery.

In contrast, in the Quebrada Blanca area, we build our cross-section mostly from mapping on satellite imagery. Here, we follow the approach proposed in Armijo et al. (2010) and described in detail in Riesner et al. (2017). The mapped georeferenced horizons are projected on the high-resolution Pléiades DEMs. Using a 3D-modeler, we project these layers along swath profiles chosen where Mesozoic strata crop out the best, where folds appear cylindrical and where topographic relief is most significant.

This approach allows for getting more precisely the large-scale structural geometries by averaging the usual local minor variations in strikes and dips that derive from direct multiple field measurements. From there, we successfully obtain the overall sectional geometry of folded layers, and by comparing with the structural map, we determine the approximate locations of the major synclinal and anticlinal axes. By respecting the classical rule of constant layer thickness, we derive fold geometries.

The limits of our interpretations mostly relate to the difficulty of unambiguously correlating stratigraphic layers, and to the fact that layers may not keep constant thicknesses. As local topographic relief is reduced to a few hundred meters at most, the construction of cross-sections is mostly restricted to extrapolating derived average surface dip angles at depth.

**3.4 Crustal shortening and kinematic modeling**

We apply a line-length-balancing approach to the obtained cross-sections to determine shortening related to folding only. This

result is independent of the geometry of the associated faults at depth, does not account neither for penetrative deformation nor for slip on underlying faults, and stands here as a conservative minimum.

From surface geology, we have no precise indication on the structure and geometry of faults and layers at depth. To make a step forward in our estimates of crustal shortening, we consider the simplest structural geometries where faults root at least at the base of the folded series. From there, we model anticlinal geometries using a numerical trishear approach (e.g.



Allmendinger, 1998; Erslev, 1991). We use the code FaultFold Forward (version 6) (Allmendinger, 1998) in order to jointly model thrust displacement and anticlinal folding. Trishear models the deformation distributed within a triangular zone located at the tip of a propagating fault (Erslev, 1991). This forward modeling relies on a set of parameters that are here adjusted by trial and error to fit observed structural geometries. By adding sedimentary layers at various steps during ongoing deformation, we model syntectonic deposition and subsequent deformation, in order to reproduce deformation of Cenozoic layers.

Additional information on trishear modeling, together with the range of tested parameters, are provided later and in supplementary material. We recognize that our best-fit model parameters may not be unique. This is not expected to impact much estimated total shortening for a specific structural interpretation as this result depends mostly on the modeled cross-sections. We cannot discard the possibility that faults are steeper and root deeper. If this were the case, crustal shortening would be lower. These various points are further discussed in section 7.2 and in the supporting information.

Deformation is expressed in terms of shortening (in km) and of relative shortening at the scale of the investigated sites (in %). Relative shortening is hereafter defined as the ratio of the estimated shortening by the initial length of the undeformed section.

## 4 Basement thrust and deformed Mesozoic series at Pinchal (~21º30'S)

### 4.1 Field observations

Because our observations are in contradiction with previous stratigraphic and structural interpretations of the folded Mesozoic
series, we hereafter describe in detail our field observations. We subsequently discuss and compare them to previous interpretations, and propose a solution reconciling these observations with regional stratigraphic knowledge.

### 4.1.1 Stratigraphic observations

In the landscape (Figure 2), the three main tectono-stratigraphic units are : (1) the metamorphic basement, (2) the Mesozoic sedimentary series (with a continuum from continental upward to marine facies) and (3) the continental Cenozoic cover. The
first-order stratigraphic column (Figure 3) is hereafter described from the oldest to the youngest units. Field pictures of sedimentary formations are provided in supplementary material to complement the forthcoming descriptions. We acknowledge not to have any constraint on the absolute ages of these series, but the relative stratigraphic ages are deduced from the kilometer-scale structural geometry and from clear sedimentary or structural polarity criteria observed in the field. Thicknesses are inferred only locally, and thickness variations cannot be excluded.

The Paleozoic basement (Figure S1) dominates the eastern part of the Pinchal area, and is composed of mainly coarse-grain granodiorites and diorites, as well as metamorphic rocks comprising gneisses, migmatites and mica-schist, consistent with previous descriptions in the area (Skarmenta & Marinovic, 1981).

The older part of the outcropping Mesozoic series consists of continental deposits (Figure S2), with a high content of Paleozoic lithics and volcano-clastic and tuffitic low-rounded conglomerates, of greenish, beige and brownish colors. Clast sizes vary
from a few millimeters to a few decimeters. At places, these rocks bear sedimentary polarity criteria such as grain-grading,



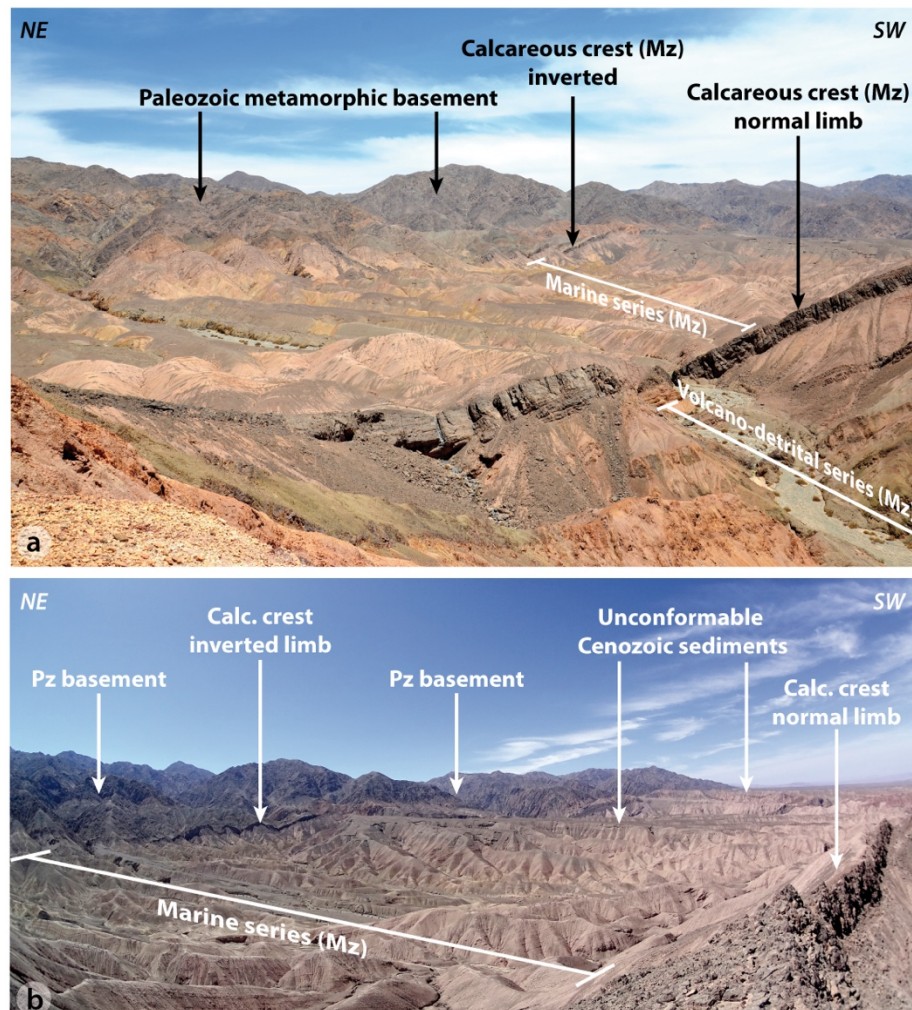

**Figure 2. Landscape field overviews of the Pinchal area depicting the main tectono-stratigraphic units.** The Paleozoic (Pz) basement stands clearly out in the background, characterized by its darker color and higher altitudes. The Mesozoic (Mz) series in the central part and in the foreground bear a marine part and a volcano-detrital part, delimited by an outstanding calcareous (Calc.) crest. Unconformable Cenozoic erosional surfaces, with limited fluvial deposits can also be observed. View points of both pictures are located on Figure 4.

grain-sorting, cross-bedding and tangential beds (Figure S3). In the eastern part of the Pinchal area, we locally observe below this series dark green detrital pelites (lutites) (Figure S4). On the basis of petrographic and sedimentological correlations, these detrital Mesozoic sediments recall units mapped as Triassic north of the Pinchal zone (between 21°–21°30'S) in the Quehuita area (Aguilef et al., 2019).

In paraconformity, a characteristic limestone layer marks the beginning of a marine sequence, evidencing a marine transgression. We refer to this layer as the "calcareous crest" as it is prominent in the landscape (Figure 2) and can be easily



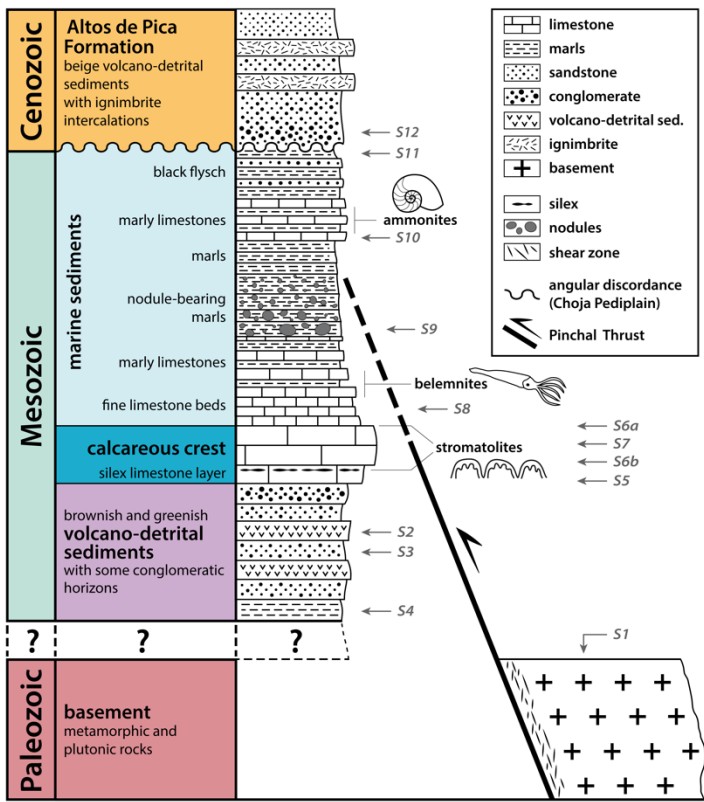


**Figure 3. First-order stratigraphic column of the Pinchal area** derived from field observations obtained mainly along Quebrada Tania (Figures 4 and 5a) where the Mesozoic series seems to be most complete. Thicknesses of the stratigraphic units are not at scale on the figure, but are given in the main text (section 4.1.1). By analogy to regional descriptions, these layers are suspected to be Triassic at the base, and Jurassic in the case of the marine fossiliferous levels (see section 4.1.3 for additional details). The description of Cenozoic units is here
completed based on the work of Victor et al. (2004). Color-code in line with maps (Figures 1, 4 and S14) and cross-sections (Figure 5). In the Pinchal area, the Paleozoic basement overthrusts folded Mesozoic series along the Pinchal Thrust, so that part of the deeper and older Mesozoic series may be missing here (as depicted by "?"). See Figures S1–S12 and corresponding captions (in supplementary material) for detailed sedimentologic descriptions.


used as a reference layer in the field or on satellite images. The base of the calcareous crest is characterized by silex layers or nodules (Figure S5). Upsection, numerous stromatolites (Figure S6) and bivalves (Figure S7) are found. Its thickness varies between a few meters (less than 10 m) in the eastern part, to ~10–20 m to the west.

The calcareous crest is overlain by thin-bedded (cm–dm) limestone layers of rose-beige color (Figure S8), over a thickness of 295 ~50–100 m. Going up-section, the marine series becomes more marly, more beige, and with less limestone layers, evidencing a deeper marine paleo-environment. Belemnite fossils were encountered in the lower part of this limestone-to-marl sequence. Characteristic calcareous oval concretions of variable diameter (cm to m) (Figure S9), are pervasive at the transition from marly limestones to marls. The marls bear ammonites, which we have not precisely identified. These ammonites could be Perisphinctes, Euaspidoceras, Mirosphinctes and Gregoryceras, according to the notice of the Quillagua geological map



(Skarmenta & Marinovic, 1981) if applicable here. In this case they would be of Middle Jurassic age (Bajocian to Callovian). The series from the thin-bedded limestones to the top of the beige marls is ~200 m thick along one of the canyons (Quebrada Tania).

Upsection, the beige marls become more calcareous again, with thin limestone layers (Figure S10). Finally, this marine sequence ends with black marls containing layers of beige sandstones (mm to few cm – rarely dm – thick) (Figure S11),

indicative of a detrital component in a probable deep-seated basin, comparable to the "flysch" series of the Alpine basins (Homewood & Lateltin, 1988). This unit is hereafter called "black flysch", and has a minimum thickness of ~50 m.

Continental-clastic Cenozoic deposits (Altos de Pica Formation), unconformably overlie this folded Mesozoic series, over the Choja erosional surface (Galli & Dingman, 1962; Galli-Olivier, 1967; Victor et al., 2004) (Figures 2 and 3). They are mainly composed of alluvial fan deposits sourced from the mountain front immediately to the east, locally interlayered or covered by

ignimbrites. We encountered red arenites at the base of the Cenozoic series in the western part of the Pinchal area (Figure S12). The age of the oldest sedimentary deposits above this erosional surface is regionally inferred to be ~27–29 Ma (Victor et al., 2004, see also section 2.2).

### 4.1.2 Structural observations

Figure 4 illustrates our structural map of the area. Two ~east–west cross-sections show detailed surface observations along

two accessible representative canyons: Quebrada Tania and Quebrada Martine (Figures 5a,b). The Quebrada Tambillo incises deeper into folded units, so that surface structural observations can be further extrapolated at depth (Figure 5c).

The easternmost part of our study area is marked by a west-vergent thrust bringing the metamorphic basement over folded Mesozoic units (Figures 2 and 6a). This basement thrust is hereafter named the Pinchal Thrust. The C/S-fabric ("Cisaillement/Schistosité") observed within the thrust shear zone indicates top-to-the-west thrusting (Figure 6b). The Pinchal

Thrust roughly follows a north-south direction (Figure 4). This contact often resumes to a single basement thrust (Figure 5a,c), but may also show local geometrical complexities, with secondary thrusts and branches, eventually involving basement with stripes of  trapped Mesozoic units, as along Quebrada Martine (Figure 5b).

Folded Mesozoic units are observable west of the Pinchal Thrust (Figures 2 and 4). From east to west, an asymmetric and overturned syncline (Figure 7a) is followed by a relatively symmetric anticline (Figure 7b).

The eastern limb of the syncline is inverted and locally highly faulted and folded (Figure 5 and Figure S13 in supplementary material). Within this inverted limb, the series goes westward (and upsection) from sheared lutites beneath the Pinchal Thrust, followed by Mesozoic detrital series with conglomerates, to the Mesozoic marine series from the calcareous crest upsection to the marly limestones. The overturned strata dip steeply (50–70ºE). Penetrative small-scale deformation is observed pervasively within the marine Mesozoic series in the form of numerous local small folds, kinematically indicative of an inverted fold limb

(used here as a structural polarity criterium) (Figure S13a), and local secondary shear zones and thrusts (Figure S13b).

Going westward, as observed in detail along Quebrada Tania (Figure 5a), the eastern part of the black flysch bears small-scale folds characteristic of an inverted fold limb, whereas normal limb folds (used here also as structural polarity criteria) are





**Figure 4. Structural map of the Pinchal area (at ~21°30'S)** derived from mapping in the field and on satellite imagery (location on Figure
1). White thin lines highlight Mesozoic layers mappable on satellite images. Thick blue line depicts the calcareous crest, which is used as a
marker bed (Figure 2). A–A' and B–B' sections locate the topographic profiles used for the surface cross-sections of Quebrada Tania and
Quebrada Martine, respectively (Figures 5a-b). In the case of the Quebrada Tambillo cross-section, a topographic swath profile was used
along C–C'. The fold axes are relatively well defined for the synclinal fold, but less well constrained for the anticlinal fold because only
observable along Quebrada Tambillo. Black dots refer to the location of field photographs, and are numbered according to the figures where
these pictures are reported. Additional field pictures provided in supplementary material are located on the map of Figure S14. Background
hillshaded DEM produced from tri-stereo Pléiades imagery. PT: Pinchal Thrust; Q: Quebrada (Chilean name for "canyon").



Figure 5. Cross-sections along (a) the Quebrada Tania (A–A' on Figure 4), (b) the Quebrada Martine (B–B' on Figure 4), and (c) the Quebrada Tambillo (C–C' on Figure 4). Reported dip angles have been measured in the field. Faults are outlined in black, and dashed when they are only observable at a local spatial scale. Only larger faults (continuous lines) are mapped on Figure 4. Fold axes are depicted above their surface trace, based on our field observations, and their orientation illustrates the deduced orientation of the corresponding axial planes. Grey numbers with arrows point out to field pictures and indicate the associated figure. In the case of the Quebrada Tania section (a), the sedimentary polarity criterion (β) indicated to the west of the section has been observed ~1 km further downstream than reported here. For the Quebrada Martine section (b), note the stripe of continental Mesozoic rocks trapped in between two strands of the Pinchal Thrust. Sub-surface interpretation from surface observations is reported with transparent colors in the case of the Quebrada Tambillo section (c). Note the different spatial scales of the three sections. PT: Pinchal Thrust.





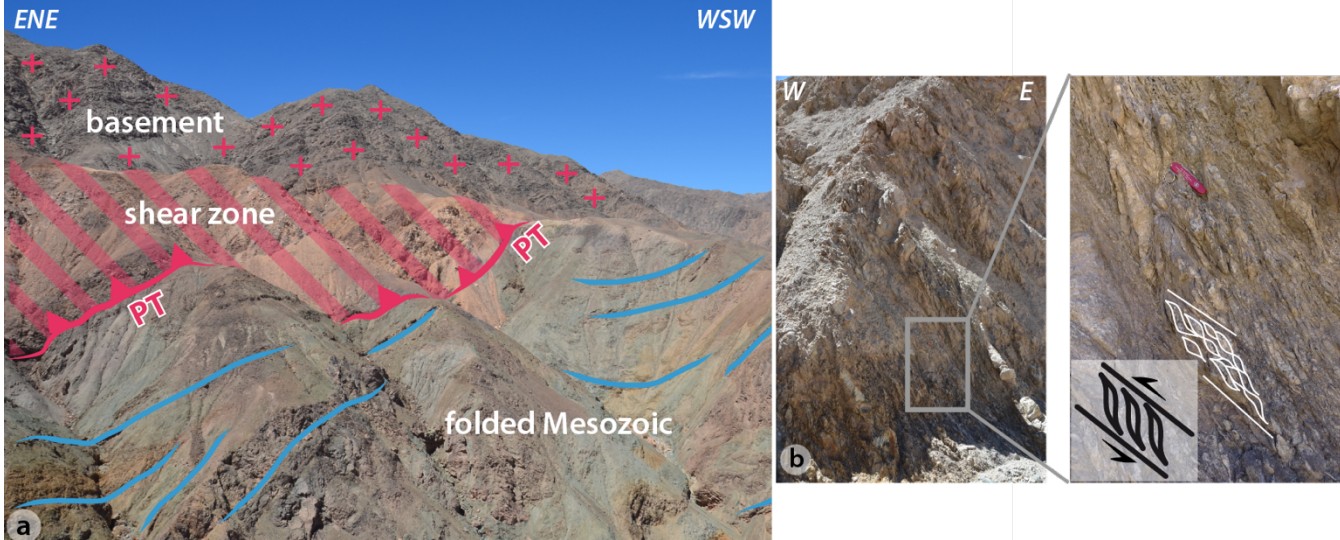

**Figure 6. Field characteristics of the Pinchal Thrust.**
(a) Field view of the Pinchal Thrust (PT), thrusting the dark-grayish Paleozoic basement over the greenish folded Mesozoic units. Reddish rocks on the hanging wall to the east-northeast correspond to the thrust shear zone (hatched area in picture). Location on Figure 4. Non-interpreted photograph can be found in the supporting information (Figure S15).
(b) Shear band with characteristic C/S-fabric (for "Cisaillement/Schistosité") indicative of top-to-the-west thrusting. Observation within the shear zone in the hanging wall of the Pinchal Thrust (deformed metamorphic basement).

observed slightly further west: the axis of the overturned west-vergent syncline therefore passes through the black flysch. Part of the Mesozoic series is missing, as overthrusting within the flysch and (marly) limestones is observed frequently along Quebrada Tania (Figure 5a). The overturned syncline is therefore broken by a secondary thrust fault striking approximately parallel to the Pinchal Thrust and roughly coinciding with the synclinal fold axis (Figures 4-5). Westward, the normal western limb of the syncline encompasses the whole Mesozoic series from the black flysch down-section to the Mesozoic volcano-detrital series, with more gentle dip angles (20–40ºE) (Figures 2 and 5). Penetrative deformation is observed to be limited here.

The continental Mesozoic layers of the normal limb of the syncline flatten toward the west. The section along Quebrada Tambillo (Figure 5c) shows a broad, overall symmetrical, anticlinal fold (Figure 7b). Its axial plane is steep, dipping ~80°E. Smaller, secondary folds with westward decreasing wavelength and amplitude are found at the western front of this large anticline. Field logistics did not permit further detailed structural observations.

The folded Mesozoic units are unconformably covered by sheet-like, river-incised Cenozoic fluvial deposits, forming aggradational terraces deposited above erosional surfaces at different elevations, of varying spatial extent and of probably different ages (Figure 2b). The majority of these erosional surfaces show a westward tilt (Figure 5c). Further west, the Cenozoic deposits become thicker and bury the westward extent of the folded Mesozoic units. Westward thickening of the Cenozoic layers is observed along Quebrada Tambillo and indicates growth strata at the front of the western anticline (Figures 5c and 7b).



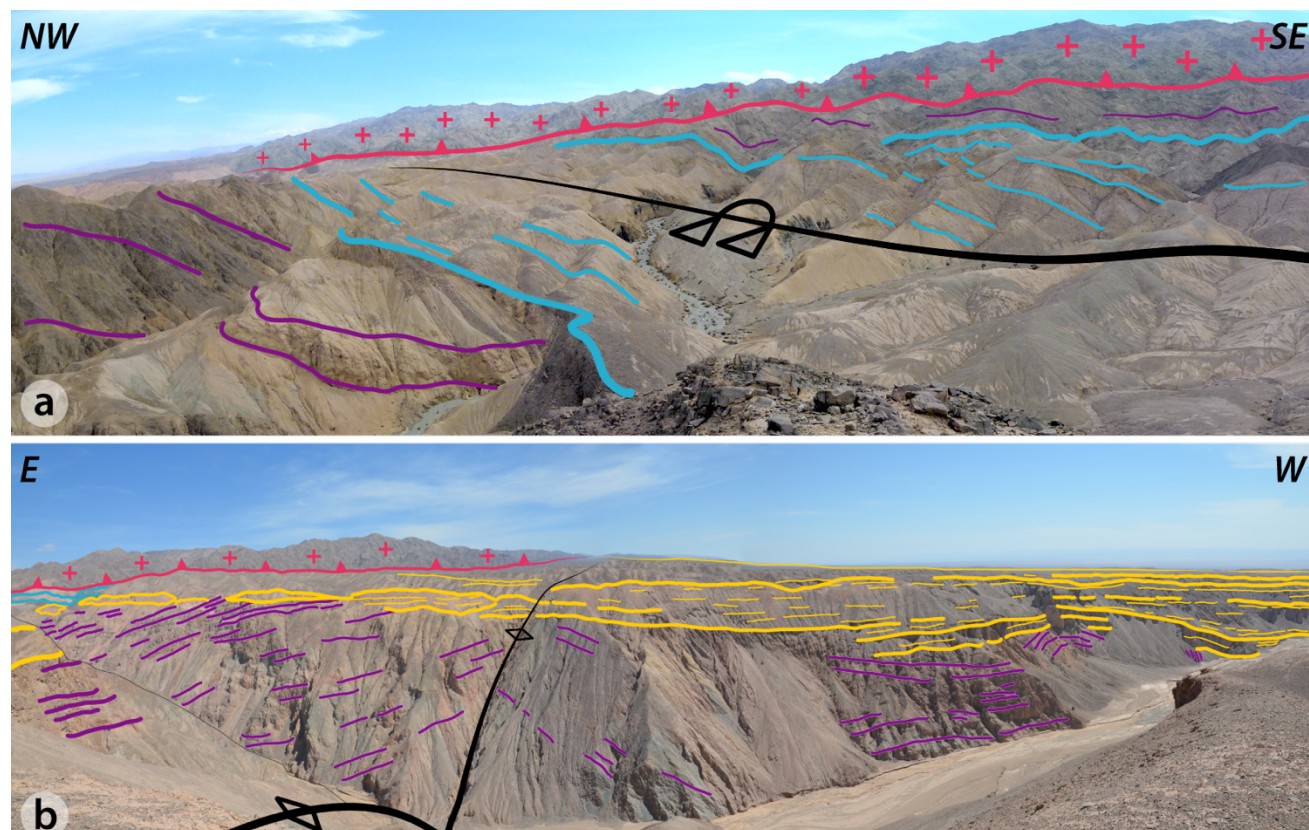

**Figure 7. Field pictures of the two major folds within the Pinchal area** (location on Figure 4). Non-interpreted photos can be found in the supporting information (Figures S16).

(a) Panoramic view over the north-eastern part of the Pinchal area. The Paleozoic basement (red crosses) overthrusts the Mesozoic units (blue and violet horizons) along the Pinchal Thrust (red line with triangles). The topographic low locates the synclinal axis. The calcareous crest on both sides is highlighted by the thick blue lines. For better visibility, Cenozoic erosional surfaces covered by thin deposits are not highlighted.

(b) Panoramic view along Quebrada Tambillo, in the southern part of the Pinchal area. The ~200 m deep incised canyon reveals the geometry of the large western anticline affecting Mesozoic layers (violet) underneath the unconformable Cenozoic strata (yellow). The fold axis (black line) probably coincides with an approximately vertical fault, also well observable on satellite imagery. Note also the repetition of smaller folds with westward decreasing amplitude and wavelength discernable beneath the westward thickening Cenozoic growth strata to the right of the picture. The Mesozoic calcareous crest (blue) and the Paleozoic basement (red crosses) over the Pinchal Thrust (red) appear in the far eastern background.

### 4.1.3 Comparison to previous stratigraphic and structural interpretations

In the Pinchal area, a basement thrust was reported in the Quillagua 1:250,000 geological map (Skarmenta & Marinovic, 1981). In this map, the Mesozoic units are interpreted as pertaining to the Jurassic Quinchamale formation, deposited in a backarc basin context and composed of an Oxfordian (~157–163 Ma) and a younger Kimmeridgian (~152–157 Ma) sub-unit. Based on this age interpretation and relying on a regionally established Mesozoic stratigraphy where marine sequences are followed



upward by younger clastic deposits, Skarmenta & Marinovic (1981) interpreted the main structure of the Pinchal zone as an anticline.

Our field investigations confirm the existence of a basement thrust, but contradict the earlier interpretation of the folded Mesozoic series and of the local Mesozoic stratigraphy. Even though we do not know the absolute ages of the folded sedimentary series, our structural and sedimentary field observations allow for clearly constraining the relative stratigraphic

ages of the folded Mesozoic units, from either structural or sedimentary polarity criteria, and unambiguously indicate that detrital continental units are here stratigraphically below a marine sequence (Figure 3). In the case that the marine strata are Jurassic in age from their likely fossiliferous content, the older continental clastic units could be Triassic, by comparison to recent observations not far from the Pinchal area (Aguilef et al., 2019).

Given this, even though the Pinchal stratigraphic sequence may look in contradiction with the regionally known stratigraphy,

it may rather be viewed as complementary: the detrital component observed below marine series may be older than the Jurassic and Cretaceous marine-to-continental upward succession that has been well described regionally. In this sense, the Pinchal area may provide a key outcrop to refine our knowledge of older series, possibly Triassic.

Detailed field pictures of the various stratigraphic and sedimentological observations are provided in supplementary material for reference. In any case, we recall that relative ages are only needed here for the scope of this study to decipher the general

structure and deformation pattern.

## 4.2 Structural Interpretation

The cross-section of Quebrada Tambillo (Figure 5c) summarizes our interpretation of the sub-surface structural geometry of the Pinchal area. Shortening estimates are summarized in Table 1.

Based on the dip angle of the C/S-fabric of the shear-zone (Figure 6b) and on the mapping of the thrust on satellite imagery,

we estimate that the Pinchal Thrust dips ~40ºE in the near-surface, locally less such as along Quebradas Tania and Martine (Figures 5a-b). All secondary strands of the Pinchal Thrust are expected to root at depth onto the main shear-zone. The secondary thrust breaking the core of the syncline, roughly parallel to the Pinchal Thrust (Figure 4), may be described as an out-of-the-syncline thrust (Mitra, 2002) and probably also connects onto it at depth. A similar reasoning is proposed to all small-scale thrusts and décollements observed within the inverted synclinal limb.

Considering that the folds west of the Pinchal Thrust develop above underlying thrusts is a reasonable and classical assumption. These thrusts are expected to root at least at the base of the outcropping folded Mesozoic series (Figure 5c), or deeper. From there, it can be extrapolated that the thrusts root at least 2 km beneath the topographic surface (i.e. at ~0.2 km a.s.l.), assuming that the layer thickness is constant over our study area. We cannot discard the possibility that the thrusts root deeper and are steeper from surface geology alone. To the west of our field area, at the front of the anticline, the small-scale folds with

westward decreasing wavelength and amplitude (Figure 7b) are interpreted as the possible expression of disharmonic folding within the forelimb of the anticline and/or of a thrust ramping-up toward the sub-surface (Figure 5c).



|  | Pinchal | Quebrada Blanca |
|---|---|---|
| Folding<br>(*line-length balancing*) | West anticline: ~0.6 km<br>East anticline: > 0.4 km<br>Total: > 1 km | West anticline: 1.8 km<br>East anticline+syncline: 2 km<br>Total: 3.8 km |
| Folding + thrusting<br>(*trishear modeling*) | West anticline: 3.1 km | West anticline: 6.6 km |
| Basement thrusting<br>(*structural section*) | Pinchal Thrust : >2.6 km | - |
| Total | Folds (folding + thrusting):<br>> 3.5 km<br>Total (folds + PT): > 6.1 km | Folds (folding + thrusting): ><br>8.6 km |


**Table 1: Shortening values on the various structures documented in this study.** Shortening associated to folding is estimated from line-length balancing on the various folds of the two investigated sites. Additional constraints on shortening are provided for the western anticlines from trishear modeling; these include folding of these anticlines and thrusting on the associated ramp. Trusting on the Pinchal Thrust (PT) is deduced from its sub-surface geometry and the minimum thickness of the folded Mesozoic series. See text for additional details.


Using the simplified geometry of layers along Quebrada Tambillo (Figure 5c), line-length balancing results in a minimum of ~1 km of shortening absorbed by folding only, from the Pinchal Thrust to the front of the anticline. Because of the pervasive presence of small-scale folding and thrusting (Figures 5a-b and S13), in particular within the inverted limb of the overthrusted

syncline, this estimate represents a minimum value. A significant – but unconstrained – amount of shortening is surely to be added, as well as the offsets on the interpreted underlying thrusts and on the Pinchal Thrust. The thrust offsets largely depend on the thrust geometry and therefore on the structural interpretation. An estimate of the contribution of thrusting will be provided below (section 6.2).

The minimum thickness of the Mesozoic series is estimated to ~2.2 km from the normal limb of the syncline along Quebrada

Tambillo. Thus, it can be considered that the strict minimum exhumation of the basement is equally of ~2.2 km. Assuming a constant 40°E dip angle, this yields a minimum displacement of ~2.6 km on the Pinchal thrust only.



## 5 Structure of the folded Mesozoic series at the Quebrada Blanca (~20°45'S)

### 5.1 Stratigraphy of the Quebrada Blanca area

The stratigraphy at ~20°45'S is well described in the Guatacondo geological map (Blanco & Tomlinson, 2013). Unlike in
Pinchal, basement rocks do not crop out in the investigated zone (Figure 8), but larger scale maps (e.g. SERNAGEOMIN,
2003) show Paleozoic basement units further east and higher in the topography (Figure 1).

The Mesozoic units of the Quebrada Blanca are of Jurassic to Cretaceous age (Blanco & Tomlinson, 2013). They have been
deposited in a back-arc basin context in successive transgression–regression sequences (Charrier et al., 2007), and are
subdivided into three formations: (1) The Late Oxfordian Majala Formation, a clastic unit of sandstones, shales and
stromatolitic limestones of transitional marine origin (Blanco et al., 2012; Blanco & Tomlinson, 2013; Galli-Olivier, 1967);
(2) the Late Jurassic / Early Cretaceous Chacarilla Formation, a fluvial clastic sequence (Blanco & Tomlinson, 2013; Dingman
& Galli, 1965); and (3) the Late Cretaceous Cerro Empexa Formation, an andesitic volcanic and continental sedimentary unit
(Blanco et al., 2000; Blanco & Tomlinson, 2013; Dingman & Galli, 1965). The Majala and Chacarilla Formations both bear
detritic reddish and beige sediments. The Cerro Empexa Formation appears greyish and massive in the field. In the Quebrada
Blanca area, uranium-lead (U/Pb) dated zircons from this formation bear ages between ~75 and ~68 Ma (Blanco et al., 2012;
Blanco & Tomlinson, 2013; Tomlinson et al., 2015) (Figure 8).

Magmatic intrusions and hydrothermalism occur locally, and hide the eastern continuation of the folded Mesozoic series. Some
of these intrusions are dated by uranium-lead (U/Pb) on zircons at ~44 Ma (Blanco & Tomlinson, 2013) (Figure 8).

The Cenozoic deposits of the Altos de Pica Formation here also overlie the Mesozoic series, over the Choja Pediplain angular
unconformity (Galli-Olivier, 1967) (see also section 2.2). The age of the basal deposits of the Altos de Pica Formation is
regionally estimated to ~27–29 Ma (Blanco & Tomlinson, 2013; Victor et al., 2004).

### 5.2 Structural observations

The structural map of the Quebrada Blanca area is reported on Figure 8. Although the cartography of the folds is complicated
by the blanketing Cenozoic cover (notably in the west and south), and by magmatic intrusions and hydrothermalism
(particularly to the east), three large-scale folds are observable: a wide syncline in the center (Higueritas syncline), bounded
by two anticlines to the west (Chacarilla anticline) and east (fold names from Blanco & Tomlinson, 2013; Fuentes et al. 2018).
The scale of these folds is multi-kilometric (Figure 8). The cross-section of Figure 9 illustrates the asymmetry of the folds:
both anticlines have steeper western limbs (dip angles vary between ~50–80ºW), whereas their eastern limbs have more gentle
dip angles (varying between ~20–50ºW) (Figure 9a). Despite the fact that the eastern flank of the eastern anticline is widely
hidden by magmatic intrusions and hydrothermalism, its southern part is well observable in the field (Figure 8). The central
Higueritas syncline is wider and more symmetric, with dip angles of ~40–50° on both limbs. The anticlines involve the Majala
and Chacarilla Formations, while the core of the syncline bears the Cerro Empexa Formation. Overall, the documented folds
show a clear west-vergence (Figure 9c).



**Figure 8. Structural map of the Quebrada Blanca zone (at ~20°45'S)**, refined from Armijo et al. (2015) (location on Figure 1). Colored lines report mappable layers. For visibility, only major, well-correlated layer traces are represented here. Black boxes locate the swath profiles from which layers were projected for the construction of the structural east–west cross-section (Figure 9). The A–B section corresponds to the topographic profile used for this same cross-section. Strike and dip measurements are extracted from 3D-map- ping (see section 3.3) or observed in the field. Strike symbols without dip value are derived from satellite imagery. Thick black lines correspond to major fold axes. Field pictures are located (with view direction), and numbered according to the associated figure. Ages from uranium-lead (U/Pb) radioisotope dating on zircon are taken from the Guatacondo geological map (Blanco & Tomlinson, 2013). Letters C, D, E, F, G and H to the north-east (within the folded Chacarilla and Majala Formations) report the layers illustrated on Figure S17 in supplementary material. Background hillshaded DEM produced from tri-stereo Pléiades imagery. Cz: Cenozoic; K: Cretaceous; Jr: Jurassic; Q: Quebrada.



**Figure 9. East–west cross-section of the Quebrada Blanca site**, established from the projection of selected, well-expressed layers mapped on satellite imagery. APF: Altos de Pica Formation.
(a) Observations, reporting the geometry of projected layers and associated dip angles, together with their stratigraphic ages (color-code).
(b) Sub-surface interpretation and extrapolation of observations.
(c) East–west cross-section based on (a) and (b). Interpretation at depth is indicated with transparent colors, in contrast with sub-surface observations. Extrapolation above the topographic surface is drawn with dashed lines. Ages from uranium-lead (U/Pb) radioisotope dating on zircon are taken from the Guatacondo geological map (Blanco & Tomlinson, 2012). The ~27–29 Ma age of the basal deposits of the Altos de Pica formation is derived from regional considerations (Victor et al., 2004).



In the field, we observe small-scale deformation within both anticlines (Figure 9). A series of anticlines with westward decreasing amplitude and wavelength (of a few tens to a few hundreds of meters – to be compared to the ~4 km wavelength of the main anticline) are observable on the western edge of the Chacarilla anticline (Figures 9c and 10). In the field, at least one of these small-scale folds is affected by a minor thrust. Additionally, within the eastern large-scale anticline, a thrust-

510  affected small-scale fold is observed (Figures 9c and Figure S17 in supplementary material), and confirms the west-vergence at this smaller scale.

The Cenozoic detrital units are unconformably deposited above the folded Mesozoic series. Thin sheet-like river-incised Cenozoic surfaces remain in the central part, becoming more dominant to the South and West (Figure 8). These superficial erosional surfaces show an overall westward tilt (Figure 9). Westward thickening of the Cenozoic layers deposited above the

515  erosional Choja surface is clearly observed at the front of the western anticline (Figure 10) and reveals the presence of growth strata.

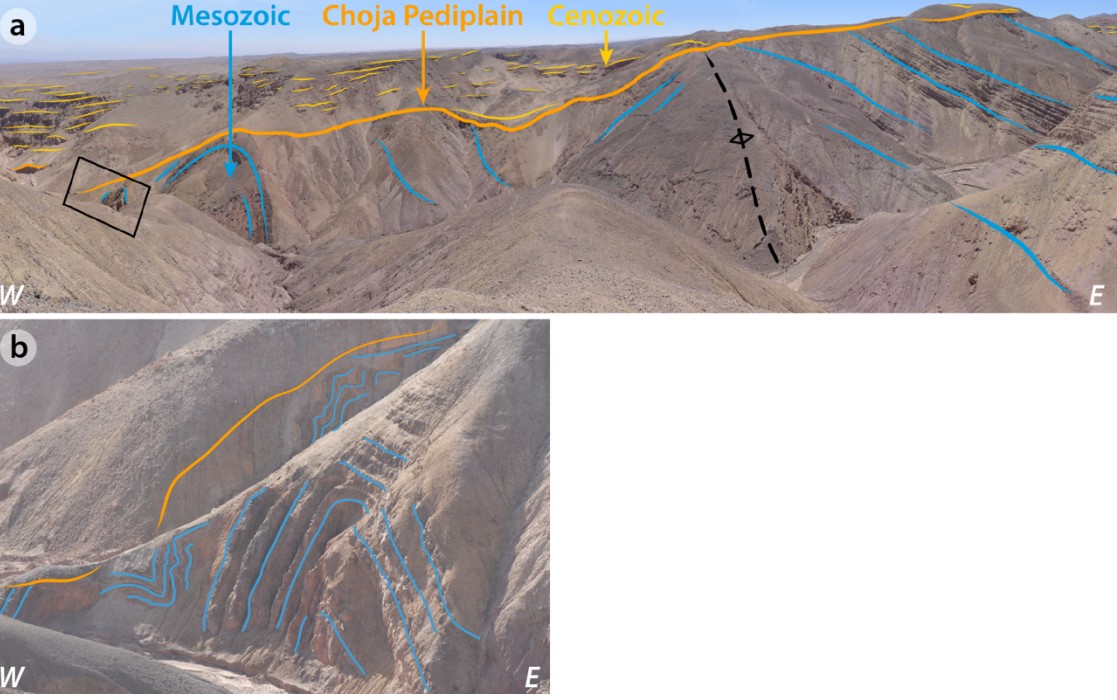

520  **Figure 10. Field picture of the western limb of the western anticline in the Quebrada Blanca area.** Non-interpreted photo- graphs are provided in supplementary material (Figure S18). Location on Figures 8 and 9.
(a) Series of folds with westward decreasing amplitude and wavelength (hundreds to tens of meters) observed at the front of the western anticline.
(b) Detailed view of the westernmost outcropping small-scale anticlines, located on (a) by the black box.



## 5.3 Structural interpretations

As for the Pinchal area, we interpret the Quebrada Blanca folds as related to ramp thrusts rooting at least at the base of the folded Mesozoic units - ie at the base of the Late Jurassic series - (Figure 9c), or deeper. Assuming constant layer thicknesses, it can be deduced that the thrusts root at least 4 km beneath the current topographic surface (i.e. at least at –2 km a.s.l.) (Figure 9c). In our cross-section of Figure 9c, the interpreted thrust needs to deepen eastward to balance the proposed section. From surface geology alone, we cannot discard the possibility that the thrust is steeper below the documented anticlines and roots deeper.

The secondary frontal folds with westward decreasing wavelength (Figures 9c and 10) can be explained as disharmonic folds within the forelimb of the large western anticline and/or be interpreted as reflecting the existence of a shallow thrust (Figure 9c). Such a feature is also in good agreement with secondary (steeper) thrusts affecting these anticlines (Figure 9).

Line-length-balancing of the cross-section of Figure 9c results in ~3.8 km of shortening solely related to folding (Table 1). This value is only a minimum as it does not account neither for slip on the related thrusts at depth nor for the observed small-scale deformation.

## 6 Kinematics of shortening of the folds and thrusts of the Pinchal and Quebrada Blanca areas

### 6.1 Timing of deformation

The projection of mapped strata indicates that the Mesozoic series is overall concordant in Quebrada Blanca (Figure 9). The cross-section of the Guatacondo map (Blanco & Tomlinson, 2013) suggests however the presence of a minor angular unconformity (<10º) at the base of the Cerro Empexa formation, not observed here from our large-scale high-resolution mapping. As this local unconformity does not produce any major change in the geometry of layers from Jurassic to Cretaceous, we consider it to be minor, in particular with respect to the main large-scale folding documented here. Folding therefore mostly post-dates the deposition of all these series. In the Quebrada Blanca area, the youngest folded layers of the Cerro Empexa Formation bear U-Pb ages of 68.9±0.6 Ma and 68±0.4 Ma (Blanco & Tomlinson, 2013) (Figures 8 and 9c). We can therefore conclude that the main deformation episode post-dates ~68 Ma, even though we cannot exclude earlier but minor deformation when compared to the observed large-scale folding (Figure 9c).

Magmatic intrusions dated at ~44 Ma intrude the folded Mesozoic units, and appear cartographically not affected by folding (Blanco & Tomlinson, 2013) (Figure 8). This suggests that the major part of the folding occurred during the ~68–44 Ma time interval. However, without additional observations of the deformation – or not – of these intrusions (geometry of the contact with surrounding host units, mineral deformation…), we cannot unequivocally conclude here from this simple cartographic observation.



Even though we suspect that the deformed series of the Pinchal zone are Triassic to Jurassic (section 4.1), we do not have any
absolute ages of the folded units. Therefore, we postulate that the main deformation here also post-dates ~68 Ma by analogy
to our observations at the Quebrada Blanca.

The folded Mesozoic units are unconformably covered by the Cenozoic Altos de Pica Formation at both investigated sites.
This is also the case for the Pinchal Thrust and secondary thrusts at few places in the Pinchal zone (Figure 4). The presence of
growth strata at the front of the westernmost anticlines in both study areas, over the erosional Choja Pediplain, suggests that
some deformation proceeded after ~29 Ma, during deposition of the Altos de Pica Formation. However, the deformation
recorded by folded Mesozoic units appears of greater intensity than that of the Cenozoic growth layers (Figures 5c and 9c).
Given this, we propose that the main folding of the Mesozoic layers documented here can be bracketed to a maximum time
span of ~40 Myr, sometime between ~68 Ma and ~29 Ma, with additional relatively minor deformation after ~29 Ma. Possibly,
the main deformation period could be shorter (~24 Myr at most), sometime between ~68 Ma and ~44 Ma, with minor
shortening after the Eocene intrusions. In the case of the Pinchal Thrust, we can only propose from our observations that
thrusting took place prior to ~29 Ma.

## 6.2 Additional constraints on shortening

Because the underlying thrusts have not reached the surface (Figures 5c and 9c), we assume fault-propagation-folding to be
the dominant mode of deformation in both study areas. To estimate the amount of shortening that is taken by thrusting at depth,
some assumptions on the footwall and thrust geometries are needed.

The simplest interpretation is to consider a thrust ramp parallel to the layers of the backlimb of the anticlines, rooting at the
base of the series involved in folding, as in the sections of Figures 5c and 9c. Disharmonic folding at the front of the anticlines
is likely related to the local thickening of layers at the front of a shallow upward propagating ramp, as in the frontal triangular
zone of trishear folds. To further quantify the shortening across the sections of Figures 5c and 9c, we do kinematic trishear
modeling  (e.g. Allmendinger, 1998; Erslev, 1991) of the westernmost anticlines documented at the Quebrada Tambillo
(Pinchal area) and Quebrada Blanca. This approach accounts for folding, slip on propagating thrust-faults, and models the
deformation distributed at the tip of these evolving faults. The trishear formalism relies on a set of parameters that are adjusted
here by trial and error so as to fit the proposed structural geometries of the anticlines. The values of these parameters are within
the range considered in previous studies (e.g. Allmendinger, 1998; Allmendinger & Shaw, 2000; Cristallini & Allmendinger,
2002; Hardy & Ford, 1997; Zehnder & Allmendinger, 2000). Here we present our best-fitting model, which allows for
reproducing satisfactorily our structural sections, acknowledging that it is not unique. Further details are provided in
supplementary material. Tables S1 to S3 provide the set of parameters used for our modeling.

The structural geometries of the westernmost anticlines of the study sites are reproduced (Figure 11), and the evolution of
deformation is modeled over time taking into account the Cenozoic growth strata. The various stages of deformation are shown
in Figures S19 and S20 in the supplementary material. We find that the geometries of the western anticlines can be reproduced
with a cumulative shortening of 3.1 km for Quebrada Tambillo (Pinchal area), and of 6.6 km for Quebrada Blanca (Figure 11).



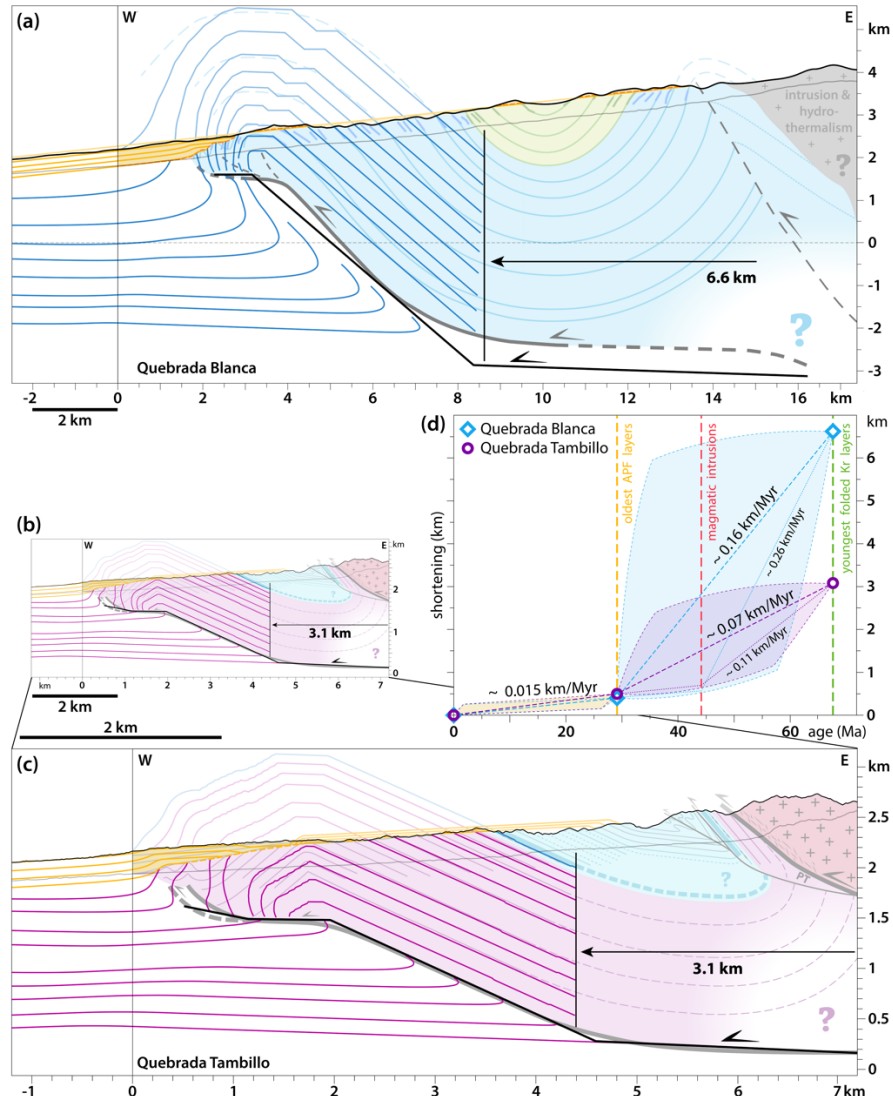

**Figure 11. Kinematics of folding of the western anticlines of the Quebrada Blanca and Pinchal zones as deduced from field observations and trishear modeling.** Modeling was performed with FaultFold Forward v.6 (Allmendinger, 1998).

(a-c) Final stages of the best-fit models in the case of (a) the Quebrada Blanca area; (b) the Quebrada Tambillo (Pinchal area), shown here at the same scale as (a). (c) Detailed and enlarged view of our results for the Quebrada Tambillo (Pinchal area). Note the large scale-difference between the sections of the two investigated sites (a,b). Thicker lines outline model results, while transparent lines and colors refer to the cross-sections of Figures 9c and 5c. These lines are color-coded according to the stratigraphic level they represent, as in the original cross-sections. Black lines report the modeled thrusts and horizontal arrows report the model total shortening. PT: Pinchal Thrust.

(d) Shortening vs. time, as deduced from trishear modeling of the western anticlines of the Quebrada Blanca and Pinchal areas, and the ages of deformed layers. The three temporal benchmarks correspond to the age of the youngest folded Cretaceous (Kr) unit (~68 Ma), to the age of magmatic intrusions (~44 Ma) that are cartographically discordant, both derived from the Guatacondo geological map (Blanco & Tomlinson, 2013 – see also Figures 8 and 9c), and to the ~29 Ma age of the oldest Cenozoic layer of the Altos de Pica Formation (APF) (Victor et al., 2004) above the Choja erosional surface. It is possible that most deformation occurred prior to ~44 Ma, as deduced from the age of the intrusions cartographically seemingly post-dating folding (Figure 8), even though this argument is to be taken with caution. Our

results underline two phases of deformation, with a slowing down of deformation since ~29 Ma at least, possibly even before. Intermediate stages of the trishear modeling are reported on Figures S19 and S20 (supplementary material) for the cross-sections of Quebrada Tambillo and Quebrada Blanca, respectively. Model parameters are indicated in Tables S1–S3 in supplementary material.



The above shortening values only account for the deformation (folding and thrusting) absorbed across the modeled westernmost anticlines. Synclinal folding accounts for an additional minimum shortening of ~0.4 km as deduced by line-length-balancing in the Pinchal area, leading to a total of >3.5 km of shortening across the Mesozoic units along the Quebrada Tambillo section. This includes folding, as well as slip on the detachment and western thrust ramp (Table 1). When adding the minimum ~2.6 km of thrusting deduced on the Pinchal Thrust, we get >6.1 km of total shortening across the whole Pinchal

area. Similarly, in the Quebrada Blanca area, the easternmost anticline and syncline take up ~2 km of shortening by folding deduced by line-length-balancing, leading to a minimum amount of shortening of ~8.6 km across the whole Quebrada Blanca section, including folding and slip on the underlying detachment and western ramp (Table 1).

The two investigated sites take up different amounts of shortening. This may relate to the fact that the across-strike extent of the two sections are significantly different (~7 km long section for the Quebrada Tambillo vs. ~17 km long section for the

Quebrada Blanca). The calculated shortenings similarly represent ~47% and ~34% of shortening when scaled to the extent of the Quebradas Tambillo and Blanca sections, respectively. Differences between these sections may also relate to the depth at which the interpreted thrusts root (altitude of ~0.2 km for Quebrada Tambillo vs. depth of ~2 km for Quebrada Blanca, relative to sea level) (Figure 11). Lateral variations in deformation can also not be excluded.

The shortening values estimated above depend on the proposed sub-surface structural interpretations, and more specifically on

the thrust geometries. They are to be considered lower bounds only within the considered structural framework. We favored the simplest geometry where the thrusts root at the base of the folded series and connect at depth, but cannot discard from local field observations alone the possibility that they are steeper single planar faults that root deeper. Folding estimated from line-length balancing only and our favored structural interpretation therefore provide a lower and upper bounds on shortening estimates, respectively. However, as further discussed in section 7.2, large-scale considerations on the overall topography and

geology at the scale of the whole Western Andean flank tend to favor our local structural interpretations of Figures 5c and 9c, and from there the shortening estimates from trishear kinematic models.

**6.3 Kinematics of shortening**

Trishear modeling allows for simulating the evolution of thrust slip and folding in the case of the westernmost anticlines of the two investigated sites. By adding syntectonic layers while deformation proceeds, we also reproduce the overall geometry

of the base of the Cenozoic Altos de Pica Formation deposits and of the subsequent growth strata (Figures S19 and S20). Syntectonic surfaces and layers are prescribed an initial 3–6° W dipping angle, similar to the present-day regional topographic slope (Figure 1). From there, we find that ~0.5 km and ~0.4 km of shortening are needed to reproduce the first-order geometry of the base of the Altos de Pica Formation at the front of the Quebrada Tambillo (Pinchal area) and Quebrada Blanca sections, respectively, using the previous trishear models adjusted to our final cross-sections. When compared to the minimum 3.1 km

and 6.6 km of total shortening accumulated since ~68 Ma across the westernmost anticlines of these two sections, this indicates that the ~29 Ma old basal Cenozoic layers above the Choja surface record at most only 16% and 6% of this total shortening,



respectively. We have tested the possibility of initial horizontal Cenozoic syntectonic layers. In this case, a post- ~29 Ma shortening of 0.8 km at most is needed to best adjust the observed geometry of the basal Altos de Pica Formation layers, even though a good fit to both the geometry of the growth strata and of the finite fold structure cannot be satisfactorily found.

These results are then used to quantitatively describe the evolution of shortening over time across the westernmost anticlines of the two investigated sections, with account on the timing of deformation discussed in section 6.1 (Figure 11d). We find that shortening rates were on average of ~0.07–0.16 km/Myr over the time span ~68–29 Ma. They could have been even as high as ~0.11–0.26 km/Myr if considering that the main deformation phase is confined to ~68–44 Ma. Subsequently, deformation rates decreased to an average value of ~0.015 km/Myr after ~29 Ma, starting possibly earlier.

These average values are most probably minimum values, within the framework of our modeled structural interpretations. Indeed, thrusting and folding are here only modeled for the westernmost anticlines of our study sites, and do not account for the shortening cumulated neither across the other structures nor on the Pinchal Thrust. Also, the main phase of deformation prior to ~29 Ma could have lasted less than the ~68–29 Ma or ~68–44 Ma considered time intervals, respectively (Figure 11d). In the case that the underlying faults are steeper and root deeper, these minimum values would be lower, but this interpretation
is not favored here (section 7.2).

Our results therefore quantitatively emphasize our former qualitative conclusion that the major phase of deformation occurred sometime between ~68 and ~29 Ma, with a significant subsequent slowing down of deformation rates afterwards, possibly as soon as ~44 Ma or earlier (Figure 11d), a general conclusion that is not dependent on the proposed sub-surface thrust geometries.

**7 Discussion**

**7.1 The Andean Basement Thrust**

**7.1.1 Evidencing a major basement thrust system along the West Andean flank (~20–22°S)**

Here, we have further documented the Pinchal Thrust, which brings basement units of the Sierra de Moreno westward over folded Mesozoic units. Our study in the Pinchal area suggests that this thrust bears local complexities with several strands and
minor splays, most probably related to the reactivation of structures in the initial pre-Andean back-arc basins. Laterally, the geological map of Skarmenta and Marinovic (1981) clearly documents this structure from ~21°15'S to 21°35'S, and possibly down to ~22°S with some structural complexities by ~21°35'S with the junction of two possible strands of this basement thrust. Similar basement thrusts have been described all along the Cordillera Domeyko between ~20°S and ~22°S. North of the map by Skarmenta and Marinovic (1981), the Quehuita (up to ~21°11'S) and Choja (between ~21°08'S–21°01'S) Faults are west-
vergent thrusts bringing basement over folded Mesozoic sediments (Aguilef et al., 2019). North of ~21°S, intrusions, hydrothermalism and surface volcanics hamper clear observation of similar basement thrusts. Such basement thrusts, if existent, would however provide a reasonable mechanism for the exhumation and exposure of basement rocks east of the



folded Mesozoic units and at higher elevations, at the latitude of Quebrada Blanca (~20°45'S) (Figure 1). For these reasons, we cannot tell with any certainty whether a thrust contact similar to that described in Pinchal (this study) and further north
(Aguilef et al., 2019) exists at this latitude, but such structure is to be suspected.

South of the map by Skarmenta and Marinovic (1981), in the Sierra de Moreno at ~21°45'S, Haschke and Günther (2003)'s section report a basement thrust over folded Mesozoic units, in agreement with the  style of deformation documented here, but with a relatively minor displacement compared to our results in Pinchal. This thrust is here called the Sierra de Moreno Thrust. Together with the (1:1,000,000) Geological map of Chile (SERNAGEOMIN, 2003), Haschke and Günther (2003)'s map
suggests that this basement thrust is cartographically continuous southward to the southern end of the Sierra de Moreno, at ~22°05'S. This possibly documents its lateral termination.

As a conclusion, there exists a series of west-vergent basement thrusts all along the Western Andean flank, with various strands mapped as local basement faults, as in our study (Figure 4) or in other maps (Aguilef et al., 2019; Haschke & Günther, 2003; SERNAGEOMIN, 2003; Skarmenta & Marinovic, 1981; Tomlinson et al., 2001). Altogether, these thrusts appear as a major
structural boundary all along the western Andean flank, bringing the basement of the Cordillera Domeyko westward over folded Mesozoic units of the earlier Andean basins (Figure 1)– and therefore contributing to the uplift of the western margin of the Altiplano. They form a segmented thrust system extending laterally over at least ~120 km north-south (Figure 1), that we propose to name here the Andean Basement Thrust (hereafter ABT) system.

We interpret the ABT to dip eastward beneath the Western Cordillera, at least >2 km (Pinchal area) or >4 km (Quebrada
Blanca area) beneath the present-day topographic surface. Deeper and eastward, this thrust probably connects to a crustal-scale ramp, as needed to sustain the large-scale uplift and topographic rise of the Western Andes (Figure 1), following the earlier ideas by Victor et al. (2004) and Armijo et al. (2015). Such crustal-scale structure has been termed the West Andean Thrust (or WAT) by Armijo et al. (2015).

### 7.1.2 Shortening and timing of deformation of the Andean Basement Thrust

We estimated that the Pinchal Thrust (as part of the ABT system) accommodated a strict minimum of ~2.6 km of shortening over a horizontal distance of ~1 km in Pinchal. This shortening would be associated with basement exhumation, but limited thermochronological data do not permit to evaluate this precisely. These data are presently absent locally in Pinchal, but sparsely exist at a regional scale when considering the ABT system over its whole extent. From apatite fission track dating in basement samples taken ~20 km east and south-east of our two study sites, Maksaev and Zentilli (1999) inferred at least 4–5
km of basement exhumation occurring between ~50–30 Ma. This is in good agreement with our results in terms of amount of uplift that would result from basement overthrusting on the ABT, and above the WAT further east. Older thermochronological ages – (U-Th)/He zircon and apatite ages of ~91 Ma and ~57 Ma, respectively – were found by Reiners et al. (2015) from the basement of the Quebrada Arcas, ~30 km south of Pinchal, in a structural setting equivalent to that documented here. These ages do not contradict the previous estimates on total exhumation by Maksaev and Zentilli (1999), even though modeling
would be needed here to precisely test this. However, they question the exact timing of basement exhumation, and, from there,





of thrusting over the ABT. In the absence of properly analyzed and modeled samples closer to the ABT, it is difficult to assess more precisely its timing or amount of exhumation, uplift and shortening.

At a few places, the Pinchal segment of the ABT is covered by Cenozoic deposits. Given this observation and with existing thermochronological ages, we postulate that the ABT was most probably active sometime by Late Cretaceous to Early
Cenozoic – and that its activity had ceased by Early Miocene. This suggests it may have been coeval with folding of the Mesozoic units documented immediately further west – or starting slightly before.

## 7.2 The West Andean Thrust System at ~20–22ºS

### 7.2.1 Evidencing a west-vergent thrust system along the West Andean flank (~20–22°S)

The west-vergent folds described here as deforming Mesozoic units at ~20–22ºS are interpreted to form above thrusts that root
at least at the base of the folded series. A similar system of folds and faults affecting Mesozoic units is expected to extend further north and south than just the two sites described here, most probably over the entire zone of ~20–22ºS (Figure 1), even though a large part north of Quebrada Blanca is covered by Cenozoic strata. This is deduced from existing maps and previous works (e.g. Aguilef et al., 2019; Haschke & Günther, 2003; SERNAGEOMIN, 2003; Skarmenta & Marinovic, 1981). It therefore probably spreads out over a north-south distance of at least ~200 km – and possibly more as folded Mesozoic
sediments are mapped on the (1:1,000,000) Geological map of Chile (SERNAGEOMIN, 2003) in the north- and south-ward continuation of the two zones investigated here.

Further west, structures at depth are hidden beneath Cenozoic deposits. Seismic profiles from the Chilean Empresa Nacional del Petroleo (ENAP), as re-interpreted by Victor et al. (2004), Jordan et al. (2010), Fuentes et al. (2018), Labbé et al. (2019) or Martínez et al. (2021), show also a series of several blind mostly west-vergent thrust-faults. These west-verging thrust faults
are interpreted by Fuentes et al. (2018) and Martínez et al. (2021) as single planar deep-reaching thrusts reactivating the faults bounding the earlier Andean basins. However, even though such geometries cannot be discarded from local poorly resolved seismic data, they cannot satisfactorily explain the large-scale geometry of the western Andean flank as noted earlier by Victor et al. (2004). Indeed, only a ramp-flat-ramp geometry of a basal master fault deeping eastward beneath the Western Cordillera can account for the overall large-scale topographic rise of the entire western plateau margin (Figure 1) (Victor et al., 2004,
Armijo et al., 2015). The blind west-vergent thrust-faults found all along the western flank at ~20–22°S can therefore reasonably be interpreted as connecting onto such an east-dipping master fault (or detachment). By integrating our local observations into these regional large-scale considerations, we favor the earlier interpretations of Victor et al (2004) and Armijo et al (2015).

Altogether, these data suggest that all these thrust faults, either hidden below Cenozoic deposits or deduced from outcropping
folds, most possibly pertain to a common west-vergent thrust system, found all along the Western Cordillera of North Chile (20-22°S). We propose to name this thrust system the West Andean Thrust System (or WATS). The WATS at ~20–22°S



therefore extends laterally over at least ~200 km, and across-strike over a much wider region (~50 km, maybe locally more) than the two ~7–17 km wide sites investigated in this study (Figure 1).

**7.2.2 Shortening across the West Andean Thrust System (~20-22°S)**

By excluding the possibility of steep deep-rooting single faults from the above large-scale considerations, we favor our local structural interpretations of Figures 5c and 9c - and from there the associated shortening estimates. The WATS of northern Chile (~20–22°S) therefore accommodates a minimum shortening of ~3–9 km, as quantified from the ~7–17 km wide investigated areas (not including the contribution of the ABT in Pinchal). At ~21°45'S, ~30 km south of the Pinchal area, Haschke and Günther (2003) report a minimum shortening of >9 km from a ~50 km wide cross-section in the Sierra de Moreno

area and further east. Within the ~8–10 km wide area encompassing an equivalent of the WATS and ABT, they estimate a minimum shortening of ~4 km (i.e. a minimum of ~30% of shortening), a value consistent with our results. This study of Haschke and Günther (2003) is to our knowledge the only other work attempting to estimate the minimum total shortening absorbed by the WATS. It becomes obvious that the various structures of the WATS in northern Chile, wherever they are (Quebrada Blanca, Pinchal or Sierra Moreno areas), all absorb multi-kilometric shortening, at the scale of only one to three

major folds and thrusts.

This conclusion further emphasizes that the ~3–9 km of shortening proposed here from the folds of the Quebrada Blanca and Pinchal areas (when excluding the contribution of the ABT in Pinchal) are under-estimates of the total shortening across the whole WATS. When applying the minimum ~34–47% shortening estimated across our two investigated sites to the ~50 km across-strike extent of the whole WATS, we find a possible crustal shortening of ~26–44 km, a value consistent – even though

in the high range – with the ~20–30 km qualitatively estimated by Armijo et al. (2015) by scaling with structural relief and crustal thickness. These estimates should however be taken with caution and as possible upper bounds, as deformation is localized on thrust faults (Haschke and Gunther, 2003; Victor et al., 2004; Fuentes et al., 2018; Martinez et al. 2021; this study) and not homogeneously distributed. A precise quantification of the deformation recorded by buried folded Mesozoic units west of our study sites is at the moment not possible from available seismic profiles.

**7.3 Temporal evolution of deformation along the western Andes (~20–22°S)**

Our investigations underline that the deformation of the Quebrada Blanca and Pinchal areas is not linearly distributed over time, and can be assigned to two main periods: (1) a period of major deformation sometime between ~68–29 Ma (possibly ~68–44 Ma); and (2) a subsequent period of moderate deformation from ~29–0 Ma (starting possibly earlier) (Figure 11d). This is deduced for the westernmost anticline of both study sites from trishear modeling, but the reduction in deformation rates

is expected at the scale of the whole investigated sites as the difference in the deformation cumulated by Mesozoic units and by post ~29 Ma Cenozoic layers can be qualitatively – but clearly – intuited from our field observations and cross-sections (Figures 5, 9 and 11). Westward, deformation is mostly well-imaged on seismic profiles for Cenozoic post- ~29 Ma growth



strata and remains less well-resolved for underlying Mesozoic units (Victor et al, 2004; Labbé et al, 2016; Fuentes et al, 2018; Martinez et al, 2021), reflecting the fact that Mesozoic units could here also be much more deformed than Cenozoic layers.

In this study, we find ~0.4–0.5 km of post ~29 Ma shortening on one single most frontal fault and fold in the case of our two investigated sections (Figures S19, S20), that is over a distance of ~5–8 km. Based on the ENAP seismic profiles in the westward prolongation of our study areas, Victor et al. (2004) determined a post ~29 Ma shortening of ~3 km, accommodated by several west-vergent thrusts within the ~40 km wide Atacama Bench. All these values are in overall good agreement when setting them to the same spatial scale, as they consistently represent ~6–8% of shortening. Compared to the minimum ~3–6

km of ante- ~29 Ma shortening (or ~34–47% of shortening) quantified on one single structure in this study (Figures S19, S20), the post ~29 Ma shortening is clearly of limited importance.

The deformation slow-down, starting by ~29 Ma at latest and possibly earlier by ~44 Ma, could therefore be regional across the entire WATS. This reasoning applies to the WATS but may also hold for the ABT. If the age of basement thrusting is not precisely known, it most probably occurred by Early Cenozoic (Maksaev & Zentilli, 1999) or even Late Cretaceous - Early

Cenozoic (deduced after Reiners et al., 2015), and had ceased by ~29 Ma (see discussion in section 7.1).

This proposed time window for major folding and possibly for thrusting over the ABT is generally consistent with the main Incaic phase of deformation inferred by various authors as the main period of Andean mountain-building stricto sensu (e.g. Charrier et al., 2007; Cornejo et al., 2003; Pardo-Casas & Molnar, 1987; Steinmann, 1929). The simplest interpretation on the post ~29 Ma decline of the shortening rate is that it results from the slow-down of the same protracted regional compressional

event which caused the formation of the west-vergent WATS and ABT. With the presently available data at 20–22°S, we cannot exclude that this slow-down may have started before ~29 Ma – possibly as soon as ~44 Ma, or even before  (section 6.3) – but definitely not afterwards.

## 7.4 Regional implications

Even though multi-kilometric, the shortening accommodated by the west-vergent structures of the western Andes outlined in

this study represents a modest contribution to the total crustal shortening of >300 km across the entire Central Andes at ~20ºS (e.g. Anderson et al., 2017; Barnes & Ehlers, 2009; Eichelberger et al., 2013; Elger et al., 2005; Faccenna et al., 2017; Kley & Monaldi, 1998; McQuarrie et al., 2005; Oncken et al., 2012; Sheffels, 1990).  It should however be recalled that the deformation absorbed across the western Andes took place mostly in the early stages of the Andean orogeny, sometime between ~68–29 Ma (possibly ~68–44 Ma) in the case of the WATS, starting possibly earlier for the ABT – in any case during the Incaic phase.

In fact, when replaced within the temporal evolution of Andean mountain-building at these latitudes (e.g. Armijo et al., 2015; Charrier et al., 2007; McQuarrie et al., 2005; Oncken et al., 2006) (see section  2.1), the early multi-kilometric shortening evidenced here represents a major contribution to initial Andean deformation, which has been most often neglected in orogen-wide studies. The slowing down of deformation across the western Andean flank by ~29 Ma – and possibly starting after ~44 Ma – may have accompanied the jumping and transfer of deformation towards the East (i.e. towards the eastern Altiplano and

further east, e.g. Isacks et al., 1988; McQuarrie et al., 2005; Oncken et al., 2006).



## 8 Conclusion

In this study, we investigate and explore from two outcropping sites two major structural features within the western flank of the Andes at ~20–22°S: (1) the Andean Basement Thrust (ABT) system, which stands as a west-vergent, >120 km long system of ~north-south trending thrusts bringing Paleozoic basement over folded Mesozoic series; (2) the West Andean Thrust system

(WATS), which is a west-vergent thrust system deforming Mesozoic and Cenozoic sediments, mostly covered by the Cenozoic Altos de Pica Formation, but cropping out in few (up to ~10–20 km wide) places along the mountain flank. The WATS extends over at least ~200 km north-south and ~50 km across-strike. Even though our investigations only rely on two limited outcropping sites, our deductions have regional implications when compared and up-scaled with previous results.

Using field and satellite observations, we build structural cross-sections and quantify the recorded shortening at two key sites

along the western mountain flank. We find a minimum shortening of >2.6 km on the ABT and of >3–9 km on the few exposed structures of the WATS. This shortening – derived from outcrop areas of limited extent – corresponds only to a fraction of the entire deformation at the scale of the whole Western Cordillera at ~20–22°S. When set on scale with the extent of the investigated structures, it implies the possibility of multi-kilometric shortening across the western flank of the Andes, possibly up to 26–44 km.

We further exploit the differential deformation recorded by folded Mesozoic layers and Cenozoic growth strata of the post ~29 Ma Altos de Pica Formation. We show that the outcropping WATS was mainly active between ~68–29 Ma (possibly ~68–44 Ma), and that its deformation rates significantly decreased after ~29 Ma (a decrease that may have started earlier, e.g. by ~44 Ma). By comparison to previous studies of the blind portions of the WATS west of our study sites, we propose that such slowing-down of deformation rates was regional rather than local. In addition, field observations and published

thermochronological results of basement exhumation suggest that this temporal evolution of deformation rates may also hold for the ABT. We therefore propose that the post ~29 Ma (or post ~44Ma) decline in shortening rates resulted from the regional slowing-down of the same protracted compressional event that caused the formation of the west-vergent WATS and ABT, most probably accompanying the transfer of Andean deformation towards the Altiplano Plateau, Eastern Cordillera, and further eastward.

**Data availability**

Pleiades satellite imagery was obtained through the ISIS program of the CNES under an academic license, and is not available for open distribution. On request, the DEMs calculated from this imagery can be provided to any academic researcher, however after approval from the CNES (contact: isis-pleiades@cnes.fr, with copy to lacassin@ipgp.fr and simoes@ipgp.fr, and referring to this manuscript). Numerical computations for the DEMs were performed using the free and open source MicMac

software suite (Rosu et al., 2014; Rupnik et al., 2016) freely available at https://micmac.ensg.eu/index.php. For cartographic mapping, we also used Google Earth imagery (Landsat 7, DigitalGlobe) freely accessible at https://earth.google.com. All geological maps used in this work are cited in the main text and in the reference section. Our own maps are provided in the



main text. All field measurements and observations have been collected by us during our field missions (march 2018, january 2019) and are provided in the main text, in the figures and in the supporting information. The trishear kinematic modeling was

conducted    using    FoldFault    Forward    version    6    (Allmendinger,    1998),    freely    available    at http://www.geo.cornell.edu/geology/faculty/RWA/programs/faultfoldforward.html

**Supplementary Data**

**Author contribution**

RL and MS designed the study and TH carried it out. TH designed all figures. TH, MS and RL prepared the manuscript, with

the contribution of all co-authors. All authors participated to field work and to the various scientific discussions.

**Competing interests**

The authors declare that they have no conflict of interest.

**Acknowledgements**

This study was supported by grants from CNRS-INSU (program TELLUS-SYSTER) and from the Institut de physique du

globe de Paris (IPGP) (PI: RL). Field work was also funded by the Andean Tectonics Laboratory of the Advanced Mining Technology Center, University of Chile (PI: DC). Earlier work on this zone by RL and DC was supported by ANR project MegaChile (grant ANR-12-BS06-0004-02) and LABEX UnivEarthS project. TH benefited from a PhD grant attributed by the French Ministry of Higher Education and Research. Pleiades satellite imagery was obtained through the ISIS program of the CNES under an academic license and is not for open distribution. The authors thank A. Delorme for his technical assistance

in producing the DEMs, using the free and open-source MicMac software suite (Rosu et al., 2014; Rupnik et al., 2016) freely available at https://micmac.ensg.eu/index.php.. Numerical computations for the DEMs were performed on the S-CAPAD platform, Institut de physique du globe de Paris (IPGP). The kinematic modeling was made using FoldFault Forward version 6 (Allmendinger, 1998). R. Armijo and the late R. Thiele are warmly thanked for the fruitful discussions that led over the years to this work and manuscript. We also benefited from discussions with C. Creixell, N. Blanco, A. Tomlinson and F. Sepulveda

(SERNAGEOMIN), from the valuable help of M. Riesner for the 3D mapping, and that of L. Barrier for facies and polarity identifications. L. Barrier and N. Bellhasen are also thanked for discussions that inspired and led to trishear modeling. Comments by L. Giambiagi, C. Mpodozis and R. Allmendinger on an earlier version of this manuscript are acknowledged. This study was partly supported by IdEx Université de Paris ANR-18-IDEX-0001.





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
