# Peer review of "A contribution to the quantification of crustal shortening and kinematics of deformation across the Western Andes (~20–22°S)."

_EGUsphere, 2022_

## Referee Comment (RC1)

Title: The western Andes at ~20–22°S: A contribution to the quantification of crustal shortening and kinematics of deformation
Author(s): Tania Habel et al.
MS No.: egusphere-2022-629
MS type: Research article

**General comments:**

In this manuscript Habel et al. constrain the timing and quantify the amount of tectonic shortening in a part of the Western Andes in northern Chile. They processed high-resolution satellite images, field observations, updated geological and structural maps in order to confront this dataset to numerically-modeled balanced cross-sections of the study area.

As it has never been done before, results and discussions are of a great interests for the community working on the Andes. The Western Cordillera is often mentioned in the literature, but the shortening has never been quantified accurately. Results, based on the study of pluri-kilometer geological objects, partially fill the data gap regarding shortening rates and timings of (re)activation of structures in the Western Cordillera. These data allow to better frame the deformation of the Western Andes and should be taken into account in future attempts to restore Andean shortening rates.

I particularly appreciated the topic, the scientific approach and the efforts of the authors in order to reach their scientific objectives. This work is worth publishing. Results are discussed and satisfactorily confronted to the literature already published. I have detailed in the sections below a few points of science to clarify.

However, the presentation of the manuscript still needs some work. It is sometimes difficult to follow the reasoning. For instance, some parts of the results are interpretations and some interpretation sections include reporting of results. Some sentences are difficult to follow (especially figure captions). I would suggest to shorten them (please see detailed comments below). Some significant information are missing and some less relevant are available. The presentation of the supplementary document is quite different from the manuscript. I advise authors to homogenize the presentation.

I am not native English speaker but some of the vocabulary and grammar seem to be incorrect. I have indicated errors I was able to identify, but I advise authors to have the manuscript proofread by a specialized company or a native English-speaking colleague if possible.

I recommend a round of minor revision, especially to fix presentation issues.

\*\*\*\*\*\*\*\*\*\*\*\*\*\*\*\*\*\*\*\*\*\*\*\*\*\*\*\*\*\*\*\*\*\*\*\*\*\*\*\*\*\*\*\*\*\*\*\*\*\*\*\*\*\*\*\*\*\*\*\*\*\*\*\*\*\*\*\*\*\*\*\*\*\*\*\*\*\*\*\*\*

**Review criteria:**

*1. Does the paper address relevant scientific questions within the scope of SE?*

→ Yes. The understanding of the timing and the quantification of deformation in the Andean Western Cordillera is of a great interest for the geological community working in the Andes. Methods, results and discussion definitely fit with the scope of Solid Earth journal.

*2. Does the paper present novel concepts, ideas, tools, or data?*

→ Yes, new ideas and data are presented in this manuscript.

*3. Are substantial conclusions reached?*

→ Yes. Results have been satisfactorily analyzed, discussed, and conceptualized. The conclusions are of importance for a better understanding of the building of the Central Andes.

*4. Are the scientific methods and assumptions valid and clearly outlined?*

→ Overall, yes. Some information about the cross-section restoration modeling process are missing. Please see detailed comments below.

*5. Are the results sufficient to support the interpretations and conclusions?*

→ Yes. Results were satisfactorily exploited to support authors' statements. The limitation associated to the results were clearly discussed.

*6. Is the description of experiments and calculations sufficiently complete and precise to allow their reproduction by fellow scientists (traceability of results)?*

→ Overall, yes. Some information about the modeling process are still missing.

*7. Do the authors give proper credit to related work and clearly indicate their own new/original contribution?*

→ Yes.

*8. Does the title clearly reflect the contents of the paper?*

→ Yes. The title should be even more assertive by showing that the shortening of the Western Andes estimated in this study is not so negligible at the scale of the entire Andes.

*9. Does the abstract provide a concise and complete summary?*

→ Yes, the abstract is complete.

*10. Is the overall presentation well-structured and clear?*

→ No, it needs a considerable amount of work, especially in the organization of the results, interpretation and discussion sections.

*11. Is the language fluent and precise?*

→ No, Some paragraphs are difficult to follow. I advise authors to have the manuscript proofread by a native English-speaking colleague if possible and refer to technical corrections below.

*12. Are mathematical formulae, symbols, abbreviations, and units correctly defined and used?*

→ Yes.

*13. Should any parts of the paper (text, formulae, figures, tables) be clarified, reduced, combined, or eliminated?*

→ Yes. Parts of the manuscript and figures should be clarified. Please see detailed comments below. Paragraphs have to been clarified and shorten by removing redundant information.

*14. Are the number and quality of references appropriate?*

→ Yes, the state of the art is almost complete and authors' results are satisfactorily confronted with the literature available in the discussion.

*15. Is the amount and quality of supplementary material appropriate?*

→ Yes. Some parts need to be clarified. Please see detailed comments hereafter.
* * *
**Specific comments / Scientific questions and issues:**

Manuscript

L22-24: I do not think that the authors can say that the shortening of the Western Andes is "negligible" even at the scale of the entire range. The study area only represents approximately 1/5 in width of the Western Cordillera; Authors should expect higher shortening for the Western Cordillera. Taking into account the authors' maximum shortening estimate, it could represent 15% of total shortening integrated to the entire Andes. Authors should be a bit more assertive about this important result because most of the deformation of Western Cordillera is hidden.

L61-62: How much shortening has been estimated in the thrust belt framed by San Ramon fault ? Authors should add this information (if available) in order to compare it with their results. Is this tectonic shortening thick or thin-skinned, or both in Santiago de Chile area?

L100: The Eastern Cordillera and the Cordillera Oriental are the same… The Cordillera Oriental is not the Interandean zone. The nomenclature should be clarified here. The Interandean zone is a transition zone between the Eastern cordillera and the Subandean zone with a specific tectonic style and specific geological units involved (Kley, 1996).

L107-115: Authors should also discuss paleo-altimetry estimates in this paragraph, which is (at least partly) linked to the deformation. Please see Sundell et al. (2019) and others for instance. In this section, authors should also mentioned the Bolivian Orocline bending that affected the area, to highlight potential relationships with the migration of the deformation eastward (Müller et al., 2002; for instance).

L125: Why is the structural organization become more complex southward? Please consider to add a short explanation here.

L226: Are variations in stratigraphic thicknesses have been taken into account (error computation in shortening estimates)? And if yes, how? Does it affect significantly the shortening estimates?

L237-243: Authors should consider to briefly detail here what was the set of parameters explored for cross-section restoration? Even if it is available in the supplementary document. Also, what were the criteria to define the best solution of the forward modeling approach? Authors should had this significant information in this paragraph.

L574-582: This part should be in the method section. Not in the discussion. Furthermore, authors should had first-order missing information here, such as:
  - Briefly describe what are parameters investigated (trishear).
  - What were criteria to define the "best-fitting model". Are specific metrics used? if qualitative only, which parameters have been taken into consideration? Authors should describe a bit more the methodology.

L585-586: Are there errors on cumulative shortening estimates? Did the authors test other modeling setups with similar results (or close results), which also satisfies the present-day geometry of the structures.

L692-696: Exhumation and uplift are different. Authors should not compare it directly. I did not find the information about the amount of uplift related to the shortening modeling. Authors should explain how uplift has been computed and what is the value of uplift taken into account for comparison.

L790-795: Authors should briefly explain hypothesis dealing about the deformation transfer from west to east in the frame of their discussion.

Supporting information

Text S1: How many forward models have been run? Information partly appear in the caption of Table S1, but it has to be written in the text also (in the supplementary and in the manuscript). How was the range of parameters explored? How the authors decided to sample the range of parameters in order to cover the parameter space in the most representative way. Authors should add these information.

\*\*\*\*\*\*\*\*\*\*\*\*\*\*\*\*\*\*\*\*\*\*\*\*\*\*\*\*\*\*\*\*\*\*\*\*\*\*\*\*\*\*\*\*\*\*\*\*\*\*\*\*\*\*\*\*\*\*\*\*\*\*\*\*\*\*\*\*\*\*\*\*\*\*\*\*\*\*\*\*\*

**Technical corrections:**

Manuscript

L12/L14/L45: "western Andes" should be written "Western Andes" with a capital letter (w). Authors should check it in the entire manuscript.

L14-15: "Our results […] regional data". Authors should consider to rework this sentence that is difficult to follow. I think the part "once our […] regional data" is not necessary.

L45: Authors should consider to add a reference (e.g. Jaillard et al., 2000; Jaillard & Soler, 1996) to justify Andean mountain-building initiation.

L47-49: Authors should considered to cite only 2 or 3 articles here, in order to focus on the most relevant papers that deal with Andean deformation migration. Some of the papers cited here do not deal exclusively with this topic.

L49-53: This sentence is too long and difficult to follow. I suggest to shorten it.

L50-51: To justify previous studies in the "various cordilleras to the east" authors should consider to cite Gérard et al. (2021) to support their statements.

L57: "locations" instead of "localities" maybe?

L67: "onset" instead of "start" maybe?

L69: Authors should refer to Figure 1 to help the reader. "blanketing Cenozoic deposits and volcanics".

L73: Authors should briefly describe here what kind of quantitative data have been processed.

L86-115: This state of the art is quite complete but, it is too difficult to follow. In order to be clearer, I would suggest to start the geological framework from the full Andes' scale and then to focus on the study area. That is to say, to keep L86-90 as it is. Then to put L97-115 in a row and next, focusing on the study area (L90-96).

L91-92: "elevation" instead of "altitude". Please check throughout the manuscript.

L95-96: Merge "Figure 1" into the bracket located before. For instance: (…Atacama Bench; Figure 1). Please check throughout the manuscript.

L98: Authors do not need to quote "following here the terminology of …" I would suggest just to cite the literature. It will be easier to read.

L118-119: Authors usually described units from west to east before. Here it is reversed. I suggest to keep this west-east logic, it will be easier for the reader.

L124: Please add the city of Calama on Figure 1.

L149-153: These sentences are confusing. Please consider to rework this section. There are too many comas.

L150: Where are located these Sierras on Figure 1? Sierra del Medio and Sierra de Moreno? Authors should consider to add these information on Figure 1.

L189: Authors should add a reference for "European Pléiades satellites images".

L191: A space is missing between "2" and "m".

L249-251: This is a paragraph for a discussion. Authors should consider to put this section in the interpretation/discussion section and not in the result section.

L254-255: "The first order […] units". Authors do not need to introduce figures like in a book chapter or in a thesis or report. Just to smoothly integrate it in the text to argue their points. This observation is valid for other sections in the manuscript.

L255-256: To my opinion, and this is valid for the entire manuscript, it would be easier for the reader if authors should refer to the exact figures in the supplementary document (in line with their statements; Figure S1 for instance) instead of just reporting that there is supplementary material here and there.

L314-315: Authors should not introduce figures at the beginning of sections, just integrate it in the text, according to their statements. It will be easier to read.

L395: The section 4.1.3 is an interpretation section. Authors must not merged results and interpretation this way. 4.1.3 part has to appear later in the manuscript. Authors should consider to had this section in the Discussion/Interpretation section.

L416: Same comment as above. Authors cannot present results and interpretations from Pinchal area and then present results and interpretations from Quebrada Blanca. The organization of the sections has to be reviewed. Authors should present all observations, stratigraphic, tectonic and modeling results for both area; and then interpret and discuss it later in the manuscript. Parts of section 4.2 has to be transferred in a distinct interpretation section (section 6 for instance, with new sub-sections for Quebrada Blanca and for Pinchal).

L417-418: In order to get straight to the point, these introductory sentences are not necessary. It makes the reading difficult with unnecessary information. Authors should simply refer to figures and tables to support their statements.

L418: There is an issue in results presentation. Authors should not refer to table 1 (presenting shortening values computed from cross-section restoration) before figures presenting cross-sections restoration/line-length balancing (Figure 11). It is very difficult to follow.

L425: For this assumption, authors should consider to add a reference to support their point.

L453: Results for Quebrada Blanca area should appear before interpretations from Pinchal area.

L525: Similar comment from above. Parts of this section are interpretations, and it should appear in a distinct and independent section later in the manuscript to avoid merging results and interpretation in the same section.

L554: It is not necessary to refer to previous sections: "section 4.1". Authors should remove this information.

L567: Section 6.2 is a mix between trishear method, results, interpretation and discussion. Results should appear earlier in the manuscript. Authors should review the structure of the manuscript.

L574-582: This part should be in the method section. Not in the discussion.

L617: "elevation" instead of "altitude".

L797-802: There are too many information here. Authors should consider to shorten this part to focus on first-order information.

L798: Authors may add "Chilean" before "Andes" to help the reader.

L855: The bibliography is not homogeneous. Some titles are in capital letters, typos are present. Please fix issues.

Figures

Figure 1:
- The text in the figure caption is not very clear. Authors should consider to rework the caption with smallest sentences.
- Authors should consider to add the topographic cross-section location on the geological map.
- Authors should consider to add major tectonic features location on the topographic cross-section to identify potential relationships between faults and topography.
- Cordillera Domeyko appears in the caption but does not appear on the geological map. Please add it.
- To simplify the caption, authors should remove abbreviation explanation for ABT, WAT, WATS, Cz, Mz, Px-Pc as it is already written in the figure. Or delete the abbreviation in the figure et keep it in the caption.
- In the inset, authors should consider to add an arrow to show the direction of convergence (Nazca vs. South American plate).

Figure 2:
- Labels "a" and "b" are on the figure but not in the caption. What is the difference between the two pictures ? Authors should maybe select one picture to avoid repetition of information.
- Figures 2 and 3 could be merged.

Figure 3:
- Approximative thicknesses of units should be indicated on the figure. Even if the log is not scaled.
- In the caption: "By analogy to regional description". Please cite the literature here.

Figure 4:
- It would be maybe better to call figure 4 before. At the beginning of section 4, to better organize this section.
- Authors do not need to add information twice (in the figure + in the caption). Field picture locations for instance.

Figure 5:
- Authors should not repeat three times "Figure 4" in the caption. It represents a repetition of the same information. It is difficult to read… I suggest to find a way to shorten these sentences.
- There are some French words in the figure: "Cisaillement, Schisosité". Please translate it.

Figure 7:
- L385/389: In the caption, the word "violet" should be replaced by "purple"?
- Authors should indicate dip marks of strata on the pictures to help the reader.

Figure 10:
- This figure is not well structured. Authors should add a symbol (on the figure) to show the relationship between picture a and b. Authors should also find a way to remove blank space at bottom right of the figure.

Figure 11:
- This Figure and associated results have to be presented before in the result section. Not in an interpretation part.
- L596-603: This text is not needed in the figure caption and should be written in the main text, otherwise the caption is too hard to follow.
- Parts of the caption are already displayed on the figure. To shorten the caption, authors should consider to remove duplicate information.

Supporting information

The style of the supplementary document is quite different from the manuscript. I advise authors to homogenize the presentation. English has to be proofread. In the manuscript, authors should refer to the exact figure label to clarify the text. Generally, figure captions should be shorten in order to optimize them.

Section (1)
- Figure S14 appears before Figure S13 in the text. Authors should check the order.
- Figures S15 and S16 are not introduced. Authors should consider to add this information.

Figure S4: Aguilef et al. 2019 citation in the caption is not referenced in the bibliography at the end of the supplementary document.

Figure S11: In the figure, authors should correct "bancs" into "bed" or "strata". And "Calcareous" into "Limestone".

Figure S14: I do not see "view directions" on the map. "Field picture" information appears twice in the legend and in the caption. To help the reader, authors should remove one of this information. "Quebrada" is not only Chilean but used in all the Andes. Authors should write "Spanish word for…". This latest observation is also valid for Figure 4.

Text S1: Some of the information presented here have to be written in the method section of the manuscript to help the reader. Please see detailed comments above.

Table S1: This table should be in the method section of the main manuscript. In the caption, authors should remove unnecessary parts (Best results […] Allmendinger, 2002) and add them instead in the text S1, or even in the manuscript.

Data Set S1: Parts of this text (although nice) are not appropriate for publication in a scientific journal. Although I love to imagine geologists in the field under a beautiful starry night, this is not required here. Authors must remain factual.

\*\*\*\*\*\*\*\*\*\*\*\*\*\*\*\*\*\*\*\*\*\*\*\*\*\*\*\*\*\*\*\*\*\*\*\*\*\*\*\*\*\*\*\*\*\*\*\*\*\*\*\*\*\*\*\*\*\*\*\*\*\*\*\*\*\*\*\*\*\*\*\*\*\*\*

Benjamin Gérard

**References cited in this review letter**

Gérard, B., Robert, X., Audin, L., Valla, P. G., Bernet, M., & Gautheron, C. (2021). Differential Exhumation of the Eastern Cordillera in the Central Andes: Evidence for South-Verging Backthrusting (Abancay Deflection, Peru). *Tectonics*, *40*(4), 1–29. https://doi.org/10.1029/2020TC006314

Jaillard, E., & Soler, P. (1996). Cretaceous to early Paleogene tectonic evolution of the northern Central Andes (0-18 degrees S) and its relations to geodynamics. *Tectonophysics*, *259*(2), 41–53. https://doi.org/10.1016/0040-1951(95)00107-7

Jaillard, E., Hérail, G., Monfret, T., Diaz-Martinez, E., Baby, P., Lavenu, A., et al. (2000). Tectonic evolution of the Andes of Ecuador, Peru, Bolivia and northermost Chile. *Tectonic Evolution of South America*, 481–559.

Kley, J. (1996). Transition from basement-involved to thin-skinned thrusting in the Cordillera Oriental of southern Bolivia. *Tectonics*, *15*(4), 763–775. https://doi.org/10.1029/95TC03868

Müller, J. P., Kley, J., & Jacobshagen, V. (2002). Structure and Cenozoic kinematics of the Eastern Cordillera, southern Bolivia (21°S). *Tectonics*, *21*(5), 1-1-1–24. https://doi.org/10.1029/2001tc001340

Sundell, K. E., Saylor, J. E., Lapen, T. J., & Horton, B. K. (2019). Implications of variable late Cenozoic surface uplift across the Peruvian central Andes. *Scientific Reports*, *9*(1), 1–12. https://doi.org/10.1038/s41598-019-41257-3

---

## Referee Comment (RC2)

[referee-annotated manuscript omitted]

---

## Author Comment (AC1)

We hereafter respond to the various comments and questions addressed by Benjamin Gerard (RC1) in his review. His review is entirely reported in black, and our responses in bold blue.

The suggested revisions do not question our results and conclusions, but will clearly help improve our manuscript by clarifying our arguments and their presentation. We thank Benjamin Gerard for his positive appreciation of our work, as well as for his various comments and suggestions

**General comments:**

In this manuscript Habel et al. constrain the timing and quantify the amount of tectonic shortening in a part of the Western Andes in northern Chile. They processed high-resolution satellite images, field observations, updated geological and structural maps in order to confront this dataset to numerically-modeled balanced cross-sections of the study area.

As it has never been done before, results and discussions are of a great interests for the community working on the Andes. The Western Cordillera is often mentioned in the literature, but the shortening has never been quantified accurately. Results, based on the study of pluri- kilometer geological objects, partially fill the data gap regarding shortening rates and timings of (re)activation of structures in the Western Cordillera. These data allow to better frame the deformation of the Western Andes and should be taken into account in future attempts to restore Andean shortening rates.

I particularly appreciated the topic, the scientific approach and the efforts of the authors in order to reach their scientific objectives. This work is worth publishing. Results are discussed and satisfactorily confronted to the literature already published. I have detailed in the sections below a few points of science to clarify.

However, the presentation of the manuscript still needs some work. It is sometimes difficult to follow the reasoning. For instance, some parts of the results are interpretations and some interpretation sections include reporting of results. Some sentences are difficult to follow (especially figure captions). I would suggest to shorten them (please see detailed comments below). Some significant information are missing and some less relevant are available. The presentation of the supplementary document is quite different from the manuscript. I advise authors to homogenize the presentation.

I am not native English speaker but some of the vocabulary and grammar seem to be incorrect. I have indicated errors I was able to identify, but I advise authors to have the manuscript proofread by a specialized company or a native English-speaking colleague if possible.

I recommend a round of minor revision, especially to fix presentation issues.

**We appreciate this various positive feedbacks on our work and will hereafter explain the modifications and corrections we propose to make our manuscript clearer and more easily readable.**

**Review criteria:**

*1. Does the paper address relevant scientific questions within the scope of SE?*

☞Yes. The understanding of the timing and the quantification of deformation in the Andean Western Cordillera is of a great interest for the geological community working in the Andes. Methods, results and discussion definitely fit with the scope of Solid Earth journal.

*2. Does the paper present novel concepts, ideas, tools, or data?*

☞Yes, new ideas and data are presented in this manuscript.

*3. Are substantial conclusions reached?*

☞Yes. Results have been satisfactorily analyzed, discussed, and conceptualized. The conclusions are of importance for a better understanding of the building of the Central Andes.

*4. Are the scientific methods and assumptions valid and clearly outlined?*

☞Overall, yes. Some information about the cross-section restoration modeling process are missing. Please see detailed comments below.

**See detailed answers hereafter**

*5. Are the results sufficient to support the interpretations and conclusions?*

☞Yes. Results were satisfactorily exploited to support authors' statements. The limitation associated to the results were clearly discussed.

*6. Is the description of experiments and calculations sufficiently complete and precise to allow their reproduction by fellow scientists (traceability of results)?*

☞    Overall, yes. Some information about the modeling process are still missing.

**See detailed answers hereafter**

*7. Do the authors give proper credit to related work and clearly indicate their own new/original contribution?*

☞Yes.

*8. Does the title clearly reflect the contents of the paper?*

☞Yes. The title should be even more assertive by showing that the shortening of the Western Andes estimated in this study is not so negligible at the scale of the entire Andes.

**We thank RC1 for this suggestion. It should be however reminded that we only clearly document a few km of shortening across our two field sites, in the Pinchal and Quebrada Blanca zones. Our proposal that shortening may not be that negligible across the whole western flank of the Andes derives from our reasoning when scaling our observations/results to the whole region - and as such may be discussed and debated. We therefore prefer to keep conservative and not too assertive or provocative in the title of the manuscript.**

*9. Does the abstract provide a concise and complete summary?*

☞Yes, the abstract is complete.

*10. Is the overall presentation well-structured and clear?*

☞No, it needs a considerable amount of work, especially in the organization of the results, interpretation and discussion sections.

**As further explained and detailed hereafter, we propose to re-organize slightly our manuscript to further separate field observations (in sections 4 and 5, for each one of the two field sites), structural interpretations and from there our various deductions on crustal shortening and kinematics (in section 6 - previously mixed in between sections 4-5 and 6), before proposing a discussion based on the regional up-scaling of our observations and deductions (section 7).**

*11. Is the language fluent and precise?*

☞No, Some paragraphs are difficult to follow. I advise authors to have the manuscript proofread by a native English-speaking colleague if possible and refer to technical corrections below.

**We will do our best to have our revised manuscript read and corrected by a native English colleague.**

*12. Are mathematical formulae, symbols, abbreviations, and units correctly defined and used?*

☞Yes.

*13. Should any parts of the paper (text, formulae, figures, tables) be clarified, reduced, combined, or eliminated?*

☞Yes. Parts of the manuscript and figures should be clarified. Please see detailed comments below. Paragraphs have to been clarified and shorten by removing redundant information.

**See detailed response hereafter.**

*14. Are the number and quality of references appropriate?*

☞Yes, the state of the art is almost complete and authors' results are satisfactorily confronted with the literature available in the discussion.

*15. Is the amount and quality of supplementary material appropriate?*

☞Yes. Some parts need to be clarified. Please see detailed comments hereafter.

**Specific comments / Scientific questions and issues:**

Manuscript

L22-24: I do not think that the authors can say that the shortening of the Western Andes is "negligible" even at the scale of the entire range. The study area only represents approximately 1/5 in width of the Western Cordillera; Authors should expect higher shortening for the Western Cordillera. Taking into account the authors' maximum shortening estimate, it could represent 15% of total shortening integrated to the entire Andes. Authors should be a bit more assertive about this important result because most of the deformation of Western Cordillera is hidden.

**As already pointed out, we only clearly document a few km of shortening across the two investigated field sites. Our proposal that shortening may be up to ~20-50 km across the whole Western Andes is only a deduction when interpreting the field sites as part of a thrust system and when scaling up our results to the whole western flank of the Andes.**

**~20-50 km represent 6-14% of the total 360 km shortening across the whole Andes. We agree that such values are not to be considered as negligible, in particular if we consider that they were accommodated at a time when no other main structures were active. However, we prefer to keep conservative so as to avoid unnecessary debate and controversy.**

**The term "negligible" may sound negative, we propose to modify slightly this sentence to clarify our point.**

L61-62: How much shortening has been estimated in the thrust belt framed by San Ramon fault ? Authors should add this information (if available) in order to compare it with their results. Is this tectonic shortening thick or thin-skinned, or both in Santiago de Chile area?

**The San Ramon fault is the most frontal fault of the West-Andean fold and thrust belt at the latitude of Santiago (33.5°S). This structure is mostly thin-skinned and has absorbed a total of 9-15 km over the last ~20-25 Myr (Riesner et al 2017), to be compared to the total 27-42 km across the whole Andes at this latitude (Riesner et al 2018) - ie 21-55% of the total Andean shortening at 33.5°S.**

**However, even though this information is available in the cited references, we recall that this section of the Andes is ~1300 km southward from the sites investigated here, and is therefore not directly comparable. To avoid confusing details and keep the focus of our manuscript on the Andes at ~20-22°S, we prefer not to add these details. The reader has all needed references in case interested.**

L100: The Eastern Cordillera and the Cordillera Oriental are the same... The Cordillera Oriental is not the Interandean zone. The nomenclature should be clarified here. The Interandean zone is a transition zone between the Eastern cordillera and the Subandean zone with a specific tectonic style and specific geological units involved (Kley, 1996).

**We agree and thank RC1 for this correction.**

L107-115: Authors should also discuss paleo-altimetry estimates in this paragraph, which is (at least partly) linked to the deformation. Please see Sundell et al. (2019) and others for instance. In this section, authors should also mentioned the Bolivian Orocline bending that affected the area, to highlight potential relationships with the migration of the deformation eastward (Müller et al., 2002; for instance).

**We agree with RC1 that elevation is a indirect indication of deformation - and as such data on paleo-altimetry could be mentioned in this paragraph. We recall however that this section is only meant at providing a general overview on the various morphotectonic units of the whole Andes and on their timing of deformation, needed to later put our results and findings in the context of the Andean orogeny. Details about paleo-altimetry or even the bending of the Bolivian orocline are somehow out of the scope of this section - and of our manuscript in general. We prefer to keep our manuscript focused and not to lengthen it with unnecessary details.**

L125: Why is the structural organization become more complex southward? Please consider to add a short explanation here.

**One possible explanation for the lateral structural variations along the Andes relates to structural inheritance from the earlier Mesozoic Andean basins. We will add a short explanation when revising the manuscript.**

L226: Are variations in stratigraphic thicknesses have been taken into account (error computation in shortening estimates)? And if yes, how? Does it affect significantly the shortening estimates?

Variations in stratigraphic thicknesses were not taken into account in computing shortening estimates. We do not have any evidence for significant variations in the case of Quebrada Blanca (Figures 9a-b). In the case of the Pinchal zone, we cannot document any such variations if existent, in particular by comparing the two limbs of the overturned syncline as the eastern limb is highly faulted and deformed.

We simply and conservatively indicate that this is a possible limitation of our interpretations - and a classical one in structural geology.

L237-243: Authors should consider to briefly detail here what was the set of parameters explored for cross-section restoration? Even if it is available in the supplementary document. Also, what were the criteria to define the best solution of the forward modeling approach? Authors should had this significant information in this paragraph.

Our preferred solution was defined by visually comparing the modeled and interpreted structural geometries. We thank RC1 for his suggestions. We will add more information on the trishear modeling in the main text, instead of keeping all relevant information in the supplementary data. We recall here that our trishear model is a forward model (and not a backward restoration) and that our preferred model is possibly a non-unique solution. This will be further emphasized in the revised manuscript.

L574-582: This part should be in the method section. Not in the discussion. Furthermore, authors should had first-order missing information here, such as:

- Briefly describe what are parameters investigated (trishear).

- What were criteria to define the "best-fitting model". Are specific metrics used? if qualitative only, which parameters have been taken into consideration? Authors should describe a bit more the methodology.

This section (section 6) is not part of the discussion (section 7), but presents the results of our modeling and interpretation of field observations.
We recall here that our preferred model is defined from visually comparing modeled and interpreted structural geometries, and that our preferred model may not a unique solution. As such, we believe that we should refer to a "preferred model", rather than to a "best-fit model" - this will be corrected when revising our manuscript.
This will be further emphasized and clarified in the revised manuscript.

L585-586: Are there errors on cumulative shortening estimates? Did the authors test other modeling setups with similar results (or close results), which also satisfies the present-day geometry of the structures.

We tested a wide range of parameters, within the range of values considered in the literature for these parameters (see explanation in the supplementary material, to be transferred partly to the main text when revising the manuscript). Within this range of acceptable values, we did not find a wide range of possible solutions reproducing our interpreted structural geometries. Given this, we estimate that our shortening values are determined with an uncertainty of 0.1-0.2 km. We recall here that our shortening estimates mostly depend on the structural geometries to be modeled (ie as deduced from our field interpretations) rather than on the detailed model parameters.

**This will be added and/or further emphasized in our revised manuscript.**

L692-696: Exhumation and uplift are different. Authors should not compare it directly. I did not find the information about the amount of uplift related to the shortening modeling. Authors should explain how uplift has been computed and what is the value of uplift taken into account for comparison.

**We agree with RC1.**
**In the case of the ABT system, these is no data on uplift, but rather on exhumation from thermochronology at the regional scale (Maksaev et al 1999, Reiners et al 2015). We compare these data to what can be deduced from our structural interpretations in the Pinchal zone. Our observations suggest >2.2 km of exhumation over the Pinchal Thrust, to erode the >2.2 km thick Mesozoic series and exhume the basement (section 4.2). Such exhumation is expected to be concomitant with overthrusting over the Pinchal thrust, as part of the ABT system. This amount of exhumation is consistent with the findings of Maksaev et al (1999) of 4-5 km of basement exhumation.**
**This will be rephrased and clarified in the revised manuscript, here but also in the revised section 4.2**

L790-795: Authors should briefly explain hypothesis dealing about the deformation transfer from west to east in the frame of their discussion.

**We do not understand the point raised here by RC1. We only aim at pointing out the fact that deformation across our field sites - and possibly across the whole Western Andean flank - seems to significantly slow down by the time deformation initiates further east (Eastern Cordillera) in the Andes. This will be tentatively rephrased if confusing.**

Supporting information

Text S1: How many forward models have been run? Information partly appear in the caption of Table S1, but it has to be written in the text also (in the supplementary and in the manuscript). How was the range of parameters explored? How the authors decided to sample the range of parameters in order to cover the parameter space in the most representative way. Authors should add these information.
**As already mentioned, this will be further detailed in the revised main manuscript and supplementary material.**

**Technical corrections:**

Manuscript

L12/L14/L45: "western Andes" should be written "Western Andes" with a capital letter (w). Authors should check it in the entire manuscript.

L14-15: "Our results [...] regional data". Authors should consider to rework this sentence that is difficult to follow. I think the part "once our [...] regional data" is not necessary.

L45: Authors should consider to add a reference (e.g. Jaillard et al., 2000; Jaillard & Soler, 1996) to justify Andean mountain-building initiation.

L47-49: Authors should considered to cite only 2 or 3 articles here, in order to focus on the most relevant papers that deal with Andean deformation migration. Some of the papers cited here do not deal exclusively with this topic.

L49-53: This sentence is too long and difficult to follow. I suggest to shorten it.

**These above corrections will be easily implemented in the revised manuscript.**

L50-51: To justify previous studies in the "various cordilleras to the east" authors should consider to cite Gérard et al. (2021) to support their statements.

**Although this contribution is undoubtedly interesting, we prefer not to mention it in our manuscript. Indeed, it is rather a local study in Peru, and if we were to cite this work, we should also cite any other work on the Altiplano-Puna... in contradiction with the simplification requested in our citations (see penultimate comment by RC1).**

L57: "locations" instead of "localities" maybe?

L67: "onset" instead of "start" maybe?

L69: Authors should refer to Figure 1 to help the reader. "blanketing Cenozoic deposits and volcanics".

L73: Authors should briefly describe here what kind of quantitative data have been processed.

**These above corrections will be implemented in the revised manuscript.**

L86-115: This state of the art is quite complete but, it is too difficult to follow. In order to be clearer, I would suggest to start the geological framework from the full Andes' scale and then to focus on the study area. That is to say, to keep L86-90 as it is. Then to put L97-115 in a row and next, focusing on the study area (L90-96).

**We do not really get the point raised by RC1. Indeed the first part (lines 86-90) refers to the overall Andean margin, then the second one (lines 90-96) to the proper Andes and their various units at the large-scale, before indicating the temporal evolution of deformation of these various units from the literature (lines 97-115). As such we already follow the recommendation of RC1 by progressively zooming into our study area. It would make no sense to indicate the temporal evolution of Andean morphotectonics units (lines 97-115) before even defining these units (lines 90-96).**
**We propose to slightly modify this section, so as to mention the topographic characteristics of the Andes, their crustal thickening and shortening after first defining the Andes along the Andean margin.**

L91-92: "elevation" instead of "altitude". Please check throughout the manuscript.

L95-96: Merge "Figure 1" into the bracket located before. For instance: (...Atacama Bench; Figure 1). Please check throughout the manuscript.

L98: Authors do not need to quote "following here the terminology of ..." I would suggest just to cite the literature. It will be easier to read.

**These above corrections will be implemented in the revised manuscript.**

L118-119: Authors usually described units from west to east before. Here it is reversed. I suggest to keep this west-east logic, it will be easier for the reader.

**We agree with RC1. The logic has been to describe morpho-structural units from west to east before, ie from the subduction trench to the South American continent. We have tried in our earlier versions of the manuscript to keep this geographic logic, but found it to be difficult as it supposes to describe stratigraphic units from the youngest to the oldest, ie the**

reverse of what is usually done. To keep a stratigraphic (ie temporal) logic from the oldest units to the youngest ones, here but also later in the manuscript when providing the detailed stratigraphy of the investigated field sites, we need to keep describing from east to west our region of interest. We will explain this logic briefly in the revised manuscript.

L124: Please add the city of Calama on Figure 1.

This correction cannot be implemented, as the city of Calama is out of the map of Figure 1. This is why we indicate its latitude in the main text.

L149-153: These sentences are confusing. Please consider to rework this section. There are too many comas.

This correction will be implemented in the revised manuscript.

L150: Where are located these Sierras on Figure 1? Sierra del Medio and Sierra de Moreno? Authors should consider to add these information on Figure 1.

These sierras are also slightly south of the map of Figure 1, except for part of the Sierra Moreno where the Pinchal zone is located. We will simplify and delete this part of the sentence mentioning local and unnecessary features.

L189: Authors should add a reference for "European Pléiades satellites images".

Rather than adding a reference, we will add the corresponding web site of the spatial program (https://earth.esa.int/eogateway/missions/pleiades) in the data availability statement.

L191: A space is missing between "2" and "m".

This correction will be implemented in the revised manuscript.

L249-251: This is a paragraph for a discussion. Authors should consider to put this section in the interpretation/discussion section and not in the result section.

We understand this comment by RC1. However, because the stratigraphic order observed in the field is the reverse of that published in the earlier geological map of the area (Skarmeta and Marinovic, 1981), with implications in terms of the sense of local structures (syncline instead of anticline) and in terms of regional stratigraphy with the possibility of Trias units, we prefer to mention this particular context early in our presentation of the Pinchal area. In contrast, in the case of the Quebrada Blanca area, we rely on earlier stratigraphic and geologic work.

L254-255: "The first order [...] units". Authors do not need to introduce figures like in a book chapter or in a thesis or report. Just to smoothly integrate it in the text to argue their points. This observation is valid for other sections in the manuscript.

This correction will be implemented in the revised manuscript.

L255-256: To my opinion, and this is valid for the entire manuscript, it would be easier for the reader if authors should refer to the exact figures in the supplementary document (in line with their statements; Figure S1 for instance) instead of just reporting that there is supplementary material here and there.

We do not understand this comment by RC1... as this is what we exactly do by refering to the exact figure to be found in the supplementary material. We will verify that this is done throughout the whole text.

L314-315: Authors should not introduce figures at the beginning of sections, just integrate it in the text, according to their statements. It will be easier to read.

**This correction will be implemented in the revised manuscript.**

L395: The section 4.1.3 is an interpretation section. Authors must not merged results and interpretation this way. 4.1.3 part has to appear later in the manuscript. Authors should consider to had this section in the Discussion/Interpretation section.

**We understand this comment by RC1. However, we believe that we need to discuss our field observations in the Pinchal zone in light of existing geological maps and regional stratigraphic knowledge, in particular because these are in contradiction with previous published work, before presenting anything about the Quebrada Blanca. As we present two different field sites with various degrees of confidence in previous work in light of our field observations, we find it confusing to strictly follow the classical manuscript organisation "data-results-interpretations-discussion" that would imply repetitions in going back and forth between the two field sites. At some point we need to discuss the strength of our field observations before presenting our interpretations, and this for each site, before combining our results together at the regional scale and discussing them. We therefore prefer to keep here this discussion on the strength of our stratigraphic considerations, before getting into the resulting structural interpretations.**

L416: Same comment as above. Authors cannot present results and interpretations from Pinchal area and then present results and interpretations from Quebrada Blanca. The organization of the sections has to be reviewed. Authors should present all observations, stratigraphic, tectonic and modeling results for both area; and then interpret and discuss it later in the manuscript. Parts of section 4.2 has to be transferred in a distinct interpretation section (section 6 for instance, with new sub-sections for Quebrada Blanca and for Pinchal).

**As the Pinchal and Quebrada Blanca zones are quite different in terms of scale, stratigraphy or structure, in particular with respect to previous works, we preferred to present observations and interpretations for each zone, separately, for clarity purposes and to avoid otherwise inevitable repetitions.**

**We understand the recommendations of RC1, but for these previous reasons, we would like to keep presenting and discussing the two zones separately, one after the other, at least where it comes to observations. However, following these suggestions, we propose to revise the structure of the manuscript as follows:**

**1) first present stratigraphic and structural observations of Pinchal (present sections 4.1.1 to 4.1.3) and Quebrada Blanca (present sections 5.1 and 5.2) zones in two separate sections (sections 4 and 5),**

**then 2) present all structural and kinematic interpretations of both zones in section 6 (including present sections 4.2, 5.3 into present section 6),**

**before 3) a general discussion (section 7) where our results and interpretations are up-scaled to the whole western Andean flank.**

L417-418: In order to get straight to the point, these introductory sentences are not necessary. It makes the reading difficult with unnecessary information. Authors should simply refer to figures and tables to support their statements.

**This correction will be implemented in the revised manuscript.**

L418: There is an issue in results presentation. Authors should not refer to table 1 (presenting shortening values computed from cross-section restoration) before figures presenting cross-sections restoration/line-length balancing (Figure 11). It is very difficult to follow.

When revising the structure of the manuscript as proposed a few lines above, any reference to Table 1 would appear later in the manuscript, in section 6. This issue should then be solved.

L425: For this assumption, authors should consider to add a reference to support their point.

This will be corrected in the revised manuscript, with references to classical papers by J. Suppe for instance.

L453: Results for Quebrada Blanca area should appear before interpretations from Pinchal area.

L525: Similar comment from above. Parts of this section are interpretations, and it should appear in a distinct and independent section later in the manuscript to avoid merging results and interpretation in the same section.

See previous answer about the proposed revision of the manuscript structure. Our proposed revision follows in its main lines this particular recommendation.

L554: It is not necessary to refer to previous sections: "section 4.1". Authors should remove this information.

This correction will be implemented in the revised manuscript.

L567: Section 6.2 is a mix between trishear method, results, interpretation and discussion. Results should appear earlier in the manuscript. Authors should review the structure of the manuscript.

Trishear results cannot appear earlier in the manuscript, as we need to present first the structural interpretations of our field sites before modeling them.
We agree that part of the text is here a repetition of the trishear method. This can be easily corrected by complementing section 3.4 (method) and by focusing only here on the model implementation and results. Some points need however to remain discussed here - such as the different spatial scales of the two sites, the meaning of these results with respect to our structural interpretations - so as to keep the discussion section (section 7) focused on up-scaling our results and interpretations and discussing their regional implications.

L574-582: This part should be in the method section. Not in the discussion. Furthermore, authors should had first-order missing information here, such as: - Briefly describe what are parameters investigated (trishear). - What were criteria to define the "best-fitting model". Are specific metrics used? if qualitative only, which parameters have been taken into consideration? Authors should describe a bit more the methodology.

As stated previously, we agree with RC1 and part of the text will be moved to the method section (section 3.4). Also to complement what is written about the model implementation, we will move to this section some key information initially provided in the supplementary material: what parameters investigated, range of values, etc. Finally, as proposed previously, the "best-fitting model" should be rather referred as the "preferred model" as we do a visual fit between model and structural section, and not a mathematical fit of the two geometries.

L617: "elevation" instead of "altitude".

L797-802: There are too many information here. Authors should consider to shorten this part to focus on first-order information.

L798: Authors may add "Chilean" before "Andes" to help the reader.

L855: The bibliography is not homogeneous. Some titles are in capital letters, typos are present. Please fix issues.

**These above corrections will be implemented in the revised manuscript.**

Figures

Figure 1: The text in the figure caption is not very clear. Authors should consider to rework the caption with smallest sentences. Authors should consider to add the topographic cross-section location on the geological map. Authors should consider to add major tectonic features location on the topographic cross-section to identify potential relationships between faults and topography. Cordillera Domeyko appears in the caption but does not appear on the geological map. Please add it. To simplify the caption, authors should remove abbreviation explanation for ABT, WAT, WATS, Cz, Mz, Px-Pc as it is already written in the figure. Or delete the abbreviation in the figure et keep it in the caption. In the inset, authors should consider to add an arrow to show the direction of convergence (Nazca vs. South American plate).

**The suggested corrections will be implemented.**

Figure 2: Labels "a" and "b" are on the figure but not in the caption. What is the difference between the two pictures ? Authors should mayve select one picture to avoir repetition of information. Figures 2 and 3 could be merged.

**We thank RC1 for these suggested corrections, which will be implemented in the revised manuscript.**

**As of merging figures 2 and 3, we do not agree with this idea as these two figures are quite different: one represents a landscape view (Figure 2), and the other the simplified stratigraphic column of the site (Figure 3).**

Figure 3: Approximative thicknesses of units should be indicated on the figure. Even if the log is not scaled. In the caption: "By analogy to regional description". Please cite the literature here.

**We already tried to report the thicknesses of the units, but because the log is not scaled this is not easy to implement for readability purposes. All thicknesses are reported in the text. Instead of citing the literature corresponding to regional stratigraphic descriptions, we rather refer to the text to avoid lengthening unnecessarily the caption.**

Figure 4: It would be maybe better to call figure 4 before. At the beginning of section 4, to better organize this section. Authors do not need to add information twice (in the figure + in the caption). Field picture locations for instance.

**The structural map of Figure 4 cannot be called before the structural descriptions, ie not before section 4.1.2. The stratigraphic description of section 4.1.1 is not related to this map but rather to Figure 3.**
**Any redondant information between figure and caption will be removed. In the case of located field pictures, the associated number still needs to be explained as the figure to which it refers.**

Figure 5: Authors should not repeat three times "Figure 4" in the caption. It represents a repetition of the same information. It is difficult to read... I suggest to find a way to shorten these sentences. There are some French words in the figure: "Cisaillement, Schisosité". Please translate it.

**Repetitions will be corrected in the revised manuscript.**

**As of the French word "Cisaillement-Schistosité".. well, this is in fact the technical word used even in English (which explains the "C-S" abbreviation used for the fabric), as some other structural words such as "décollement", etc.**

Figure 7: L385/389: In the caption, the word "violet" should be replaced by "purple"? Authors should indicate dip marks of strata on the pictures to help the reader.

**The word "violet" will be corrected.**
**We will also implement some dip marks, but will mostly try not to load too much the figure to keep it readable.**

Figure 10: This figure is not well structured. Authors should add a symbol (on the figure) to show the relationship between picture a and b. Authors should also find a way to remove blank space at bottom right of the figure.

**We will modify the figure so as to balance the blank space in the lower right corner of the figure (ex: moving lower panel to the middle), and so as to better illustrate the link between figures a and b (ex: adding an arrow).**

Figure 11: This Figure and associated results have to be presented before in the result section. Not in an interpretation part. L596-603: This text is not needed in the figure caption and should be written in the main text, otherwise the caption is too hard to follow. Parts of the caption are already displayed on the figure. To shorten the caption, authors should consider to remove duplicate information.

**We do not agree with RC1. This figure can only appear with section 6 as it presents and uses results from trishear modeling, together with geological time constraints. Therefore it must be in an interpretation section, even though this interpretation integrates modeling results. We recognize that time benchmarks, which provide key time constraints for our kinematic interpretation, are already explained in the main text. We will simplify the caption, but prefer to clarify and explain the key time lines that appear in the figure as these are important for our conclusions - and to understand the figure.**

Supporting information

The style of the supplementary document is quite different from the manuscript. I advise authors to homogenize the presentation. English has to be proofread. In the manuscript, authors should refer to the exact figure label to clarify the text. Generally, figure captions should be shorten in order to optimize them.

**We will homogenize the format of the supplementary material with that of the main text. We recall here that there is no indication of a specific format to be followed for supplementary material in the author guidelines of Solid Earth.**

**English will be verified. As of referring to the exact figures in the main text, this was already the case in the initially submitted manuscript.**

Section (1)

Figure S14 appears before Figure S13 in the text. Authors should check the order.

**This will be verified after revision of our manuscript, we thank RC1 for pointing this.**

Figures S15 and S16 are not introduced. Authors should consider to add this information.

Figure S4: Aguilef et al. 2019 citation in the caption is not referenced in the bibliography at the end of the supplementary document.

Figure S11: In the figure, authors should correct "bancs" into "bed" or "strata". And "Calcareous" into "Limestone".

Figure S14: I do not see "view directions" on the map. "Field picture" information appears twice in the legend and in the caption. To help the reader, authors should remove one of this information. "Quebrada" is not only Chilean but used in all the Andes. Authors should write "Spanish word for...". This latest observation is also valid for Figure 4.

**The above suggestions will be implemented in the revised manuscript.**

Text S1: Some of the information presented here have to be written in the method section of the manuscript to help the reader. Please see detailed comments above.

**See previous responses about moving details on the trishear modeling from supplementary material to the main text.**

Table S1: This table should be in the method section of the main manuscript. In the caption, authors should remove unnecessary parts (Best results [...] Allmendinger, 2002) and add them instead in the text S1, or even in the manuscript.

**We agree with RC1 that this information is important, however not crucial for the main message conveyed in our manuscript. To avoid unnecessarily lengthening the manuscript, we'd rather keep this table in the supplementary material.**

Data Set S1: Parts of this text (although nice) are not appropriate for publication in a scientific journal. Although I love to imagine geologists in the field under a beautiful starry night, this is not required here. Authors must remain factual.

**We liked the idea of letting the authors free for a couple words of fantasy - in the supplementary material and not in the scientific argumentation presented in the main text. Anyway, if asked by the editor, we will remove it - but we sincerely hope that the editor will let us keep this slight departure with strict uses since it has no consequence on the science.**

Benjamin Gérard

**References cited in this review letter**

Gérard, B., Robert, X., Audin, L., Valla, P. G., Bernet, M., & Gautheron, C. (2021). Differential Exhumation of the Eastern Cordillera in the Central Andes: Evidence for South-Verging Backthrusting (Abancay Deflection, Peru). *Tectonics*, *40*(4), 1–29. https://doi.org/10.1029/2020TC006314

Jaillard, E., & Soler, P. (1996). Cretaceous to early Paleogene tectonic evolution of the northern Central Andes (0-18 degrees S) and its relations to geodynamics. *Tectonophysics*, *259*(2), 41–53. https://doi.org/10.1016/0040-1951(95)00107-7

Jaillard, E., Hérail, G., Monfret, T., Diaz-Martinez, E., Baby, P., Lavenu, A., et al. (2000). Tectonic evolution of the Andes of Ecuador, Peru, Bolivia and northermost Chile. *Tectonic Evolution of South America*, 481–559.

Kley, J. (1996). Transition from basement-involved to thin-skinned thrusting in the Cordillera Oriental of southern Bolivia. *Tectonics*, *15*(4), 763–775. https://doi.org/10.1029/95TC03868

Müller, J. P., Kley, J., & Jacobshagen, V. (2002). Structure and Cenozoic kinematics of the Eastern Cordillera, southern Bolivia (21°S). *Tectonics*, *21*(5), 1-1-1–24. https://doi.org/10.1029/2001tc001340

Sundell, K. E., Saylor, J. E., Lapen, T. J., & Horton, B. K. (2019). Implications of variable late Cenozoic surface uplift across the Peruvian central Andes. *Scientific Reports*, *9*(1), 1–12. https://doi.org/10.1038/s41598-019-41257-3

---

## Author Comment (AC2)

We hereafter respond to the various comments and questions addressed by Patrice Baby (RC2) in his review. His review is entirely reported in black, and our responses in bold blue.

The suggested revisions do not question our results and conclusions, but will clearly help improve our manuscript by clarifying our arguments and their presentation. We thank Patrice Baby for this discussion, as well as for his various comments and suggestions.

The paper of Habel et al. presents a structural study of two sites of the western Andes in Chile (20-22°S), where the authors use numerical trishear forward modelling to evaluate minimum horizontal shortening and analyse the kinematic evolution of two fault-related anticlines.

Before being published, this paper must better document the structural observations, which are not always convincing (see below). These data can be used to construct in each area a balanced section to validate structural interpretations and calculate shortenings more rigorously. The authors need to better explain why they chose a fault propagation fold model rather than a tectonic inversion model in their structural interpretation.

These various issues on why we conservatively can't do much better on building cross-sections and calculating from there shortening, or the chosen structural interpretation (tectonic inversion vs. fault propagation fold) will be detailed hereafter. To summarize here in a few words:

- There are no constraints on the structure of the footwall of the faults inferred to generate the folds observed from surface geology. Given this, a more rigorous structural section and the associated shortening can't be proposed. Because of this, we do only propose a possible interpretation and discuss its implications.

- We do not oppose tectonic inversion to fault-propagation folding. In fact, the investigated regions were forming basins during the Mesozoic, and these basins have been inverted during the Cenozoic to form the structures documented here along the Western flank of the Andes. This inversion most probably re-used faults that structured these basins, variably along-strike, and these inverted faults may have formed the fault-propagation folds documented in our work. However, the main question relates here on whether these faults connect or not at depth onto a common detachment (as proposed by e.g., Victor et al 2004 and Armijo et al 2015), or whether they are independent from each other (as proposed in Fuentes et al 2018 and Martinez et al 2021, even though their later interpretation - in Fig 37.9 of Martinez and Fuentes 2022 - gets back to connecting these faults at depth onto an east-dipping detachment). Surface geology of the investigated field sites does not allow to choose between either one of these two models. However, from regional considerations we prefer the interpretation that these faults connect at depth onto a common detachment (see section 7.2.1).

**GENERAL COMMENTS:**

The title must be modified. The studied areas are too small to represent the entire western Andes. It would be interesting to locate the two sites on a regional cross-section through the western Andes.

We agree with RC2 that our two areas are too small to represent the entire Western Andes - this is largely recognized throughout the manuscript. However, based on our findings

from these two limited areas, we discuss the implications of these findings for shortening and kinematics of deformation at the scale of the entire Western Andes, from a regional reasoning.

The fact that we mention that our work is only "a contribution to the quantification of crustal shortening" does imply that we provide key additional data, not that we solve the problem at the scale of the entire Western Andes.

Adding a regional general cross-section of the region is a good idea, in particular in Figure 1, to structurally locate the investigated field sites. A section inspired from that shown in Figure 13 of Armijo et al 2015 would be appropriate. However, providing this section early in the manuscript may give the wrong impression that our interpretations are biased, and give room to unneeded criticism, even before we put forward our various arguments. As such, we'd rather avoid to revise the manuscript accordingly.

The stratigraphic and geologic background (2.2.1) needs a figure with a synthetic stratigraphic column.

This is already schematically provided in the legend of Figure 1, which corresponds to this part of the main text. As this is a very general regional overview, we do not see the need for more at this stage.

Structural and kinematic context (2.2.2):

In their last paper, Martinez and Fuentes (2022)(https://doi-org.insu.bib.cnrs.fr/10.1016/B978-0-323-85175-6.00037-7) show the importance of tectonic inversion of the Jurassic rift in this region. The analysed seismic sections are just west of the study areas of this paper and must be taken into consideration and discussed.

In fact, we already discussed the work and results of the group of Martinez and Fuentes, by citing their original work and data (Martinez et al 2021 and Fuentes et al 2018) rather than this later summarizing paper that does not provide any new interpretation of the seismic profiles.

As mentioned earlier, tectonic inversion is not in opposition with our interpretation. In fact, the continental and marine Mesozoic series described in our field sites were deposited initially in basins, that were lately inverted during Cenozoic compression to form the structures forming the Western flank of the Andes. We will clarify this in our manuscript.

In fact the two main previous opposing views of these structures were about their geometries at depth. On one hand, Martinez et al 2021 and Fuentes et al 2018 propose that they relate to the inversion of previous steep normal faults, not connected to each other, minimizing shortening estimates. On the other hand, Victor et al 2004 and Armijo et al 2015 propose that these faults form a thrust system connected at depth onto a common decollement, dipping eastward beneath the Andes. Seismic profiles to the west of our field sites are too poorly resolved at depth - let's be frank on this! - and field observations are too sparse and local to favor one or the other model. However, as already discussed in Victor et al 2004, and considered by Armijo et al 2015, only the thrust system model is able to provide a satisfactory structural framework for the large-scale structural organisation (deep basement to the east, shallower Mesozoic and Cenozoic units to the west) and topography (high to the east, low to the west, with a west-dipping continuous slope) of the Western Andes. Interestingly, in their latest interpretation, Martinez and Fuentes 2022

propose to connect these various faults at depth onto an east-dipping detachment (See their figure 37.9).

To summarize, only from regional considerations can we also favor the interpretation of a thrust system, as indicated in our final discussion (section 7.2), and the implications of this choice in terms of shortening were also discussed. We will revise our manuscript to further emphasize these points and the reasons for our final interpretations. In particular if parts of the manuscript are re-organized according to the recommendations of RC1, with a clearer separation between observations and structural interpretations, this can be easily done.

**Data and structural observations:**

It is important to better document the field data. For example, it is necessary to localise the field structural data in the structural map of Figure 4 to validate the cross-sections construction and structural interpretations.

All field data are reported on the map of Figure 4 and on the field sections of Figure 5. We cannot place on the map all measurements for obvious readability reasons, this is why we represented (some of) them along the field sections of Figure 5 - a section that is located on the map.

We will try to add some of our structural measurements - as long as the figure keeps readable and these additions are meaningful. We remind here that additional field observations were located on the map of Figure S14

We would like also to underline the fact that field data do not resume, to our sense, only to numerous analytical field measurements of strike and dip angles, but also encompass large-scale landscape observations - illustrated in the various field pictures provided.

Field pictures interpretations must be also validated by field data. These field data, as structural dip measurements, must be placed on the pictures. I am not at all convinced by the structural interpretation of the picture in Figure 7b. I can't really see the axis of the anticline of the Quebrada Tambillo, which is a key element of the structural interpretation. This picture interpretation must be absolutely validated by field measurements.

We chose to show field pictures to depict the various observed structures, for instance with the general trend in dip angles on either side of axial planes. This is to our sense much more meaningful here than detailed field measurements for our interpretations at the scale of the field sites. In figure 7b for instance, the axis of the anticline is defined by the change in the dip orientation of layers, with layers dipping to the east (left of the figure) on one side and layers dipping to the west (right on the figure) on the other side, whatever the absolute values of the dip angles.

However, we will try to add additional information on the field pictures, whenever appropriate and meaningful, either from direct field measurements, or from the projection of mapped 3D-layers on satellite images and DEMs - as long as the figures keep readable.

The structural map of the Quebrada Blanca zone shows structural dips values, which is not the case for the structural map of the Pinchal area. I understand that strike and dip measurements are extracted from 3D mapping. These 3D mapping and data extraction must be documented with some detailed illustrations.

**See our previous answers about representing structural measurements on the map of Figure 4 (Pinchal Zone). A detailed illustration of the structural inferences that can be made from 3D maps is already provided in the surface section of Figure 5a.**

**Structural interpretations:**

I don't understand why the authors didn't try to construct balanced cross-sections, the best way for thrust system modelling and calculation of shortening. The proposed interpretations are not geometrically validated. The footwalls of the thrusts have not been constructed (?).

**The footwalls of the thrusts were not drawn in our sections of Figures 5 and 9, as we recognize not to have any indication on their structure. The structure used to build our trishear models is just a proposition, as indicated in the text. As such no definitive balanced section can be honestly proposed just from our field observations. We will make sure this point is clear when revising the manuscript.**

The authors propose a model of fault propagation fold (or fault bend fold (?)) for each section. Why? Why not a tectonic inversion? How do you explain such a steep frontal ramp in the cross-section of Figure 9C? Are there lithologies compatible with the levels of detachment?

**As already indicated, tectonic inversion is not to be opposed to fault-propagation folding (see previous answers). We agree that structural inheritance from the earlier Andean Basin may have played an important role in localizing the thrusts at depth, or in controlling their steep geometry. This will be clarified in our revised manuscript.**

**This said, the observed folding is most simply explained by fault-related folding. Fault-propagation folding is favored here as the faults do not reach the surface, and as small-scale folds at the front of the western anticlines are possibly indicative of disharmonic folding (and therefore internal deformation) at the tip of an upward propagating ramp.**

**As of the lithologies at depth that would favor a decollement level, as can be deduced from our cross-sections (Figures 5c and 9c), these stratigraphic levels do not crop out in the investigated areas and we cannot tell more on the subject. There are just deduced from the surface geometries of the folded layers.**

**Our manuscript will be re-organized so as to better clarify these points. We propose to better separate field observations, and our choices of interpretations. But in any case this does not question our final conclusions and the field data provided.**

The calculation of shortening is confusing ("Folding" + "Folding + thrusting"(?)).

**We propose to move to section 6 all the structural reasoning leading to shortening calculations. By having all this together in one section, these various results should be clarified.**

The discussion would require an integration of results in a regional cross-section through the Western Andes.

**This has already been discussed previously**

**My detailed comments are highlighted in the attached pdf version.**

Please also note the supplement to this comment:
https://egusphere.copernicus.org/preprints/2022/egusphere-2022-629/egusphere-2022-6 29-RC2-supplement.pdf

We hereafter copy the various comments found throughout the reviewer's annotated PDF. Please note that some text sections were highlighted but not commented by the reviewer.

Line 1: The studied areas are too small to represent the entire western Andes unit

See previous answer to the comment on the title

Line 17: They are not really restored

Right, since we do not model the entire sections. This will be suppressed, thanks for the correction.

Lines 47-49: Add: Baby et al., 1997
https://doi.org/10.1130/0091-7613(1997)025<0883:NSCTCT>2.3.CO;2
and Rochat et al., 1999
https://www.researchgate.net/publication/233713934_Crustal_balance_and_control_of_the_erosive_and_sedimentary_processes_on_the_Altiplano_formation
Lines 101-102: Baby et al. (1997):
https://doi.org/10.1130/0091-7613(1997)025<0883:NSCTCT>2.3.CO;2
Lines 104-106: Baby et al. (1997):
https://doi.org/10.1130/0091-7613(1997)025<0883:NSCTCT>2.3.CO;2

We recall that RC1 recommended to limit here the citations only to the few most significant ones.. among the numerous possible from the extensive literature on the subject of Andean shortening across the whole range!

Figure 2: It would be appropriate to represent the Pinchal thrust on both images.

We agree that this could be a good idea, even though the aim of this figure is to present first-order observations on the stratigraphic organization at the scale of the landscape. We fear that adding the Pinchal Thrust could load too much the figure, we will however try to add this information and keep the figure readable.

Figure 3: It is hard to imagine that there are no more precise ages in the bibliography with all these fossils and volcanic levels.It would be appropriate to indicate on the figure more precise ages such as 27-29 Ma at the base of the Cenozoic, or the Triassic and Jurassic (Majala o Chacarilla Fm?).

We recall that the Pinchal zone is really remote and not easily accessible. The only geological map of the area dates back from the early 1980's (Skarmeta and Marinovic, 1981) and we believe from our field experience of this site that their stratigraphic correlation was done from an a priori knowledge of the regional stratigraphy rather by actual dating of the local layers.

Precise dating remains to be done here and we agree that there is a good potential for this. The SERNAGEOMIN (Geological and Mining service of Chile) will be working on this most probably over the next years. This is out of the scope of this manuscript.

Figure 3: Locate the décollement of the cross-section in Figure 5c?

This decollement is a structural interpretation proposed from our field observations, not a field observation and should therefore not be placed on the stratigraphic column of Figure 3.

Figure 5c: I am not convinced by this anticline. Field data are lacking

We would have liked to have more indications on why RC2 is not convinced by the existence of this anticline. As shown in Figure 7b, the axial plane and the two fold limbs re well defined at the landscape scale, with layers dipping eastward to the east and layers

**dipping westward to the west. In the absence of any field indication of inverted series... we get an anticline!**

Figure 5c: This thrust geometry is very arbitrary. The footwall of the thrust must be constructed to geometrically validate the interpretation. Is there a lithology compatible with this level of décollement? Why not tectonic inversion?

**To clarify our point, we propose to better separate field observations, sub-surface and deeper interpretations**

**As already mentioned, we have no clue about the structure of the footwall from field geology or even geophysics - this is why we had left it blank. Also, we already explained earlier why the tectonic inversion hypothesis is not to be opposed to fault-related folding. This will be further clarified.**

Figure 6: You must indicate values of the structural dips collected on the field and show them on the photo. Scale?

**We can add an approximate scale on Figure 6a - there is already a scale (Swiss knife) on the picture of Figure 6b). Dip measurements could be added on Figure 6a, but would not be meaningful here, at this scale, in the highly deformed footwall of the Pinchal Thrust. We rather choose not to do so.**

Figure 7: You must indicate values of the structural dips collected on the field and show them on the photo. Scale?

**We will try to add some dip angles, either measured in the field or deduced from 3D mapping, but will do so as long as the figure keeps readable. The objective of this figure is to offer a landscape view of the folds, dip angles are provided along the various field sections of Figure 5 where we feel they are more appropriate.**

Figure 9c: I'm not really convinced by this part. Lack of field data!

**Well... the small-scale folds are illustrated in the field picture of Figure 10b.... as indicated in the boxes of Figures 9a-b. To our sense, not much to be added here without a more precise comment.**

Figure 9c: This thrust geometry is very arbitrary. The footwall of the thrust must be constructed to geometrically validate the interpretation. Are there lithologies compatible with these level of detachment? Why not tectonic inversion? How do you explain such a steep ramp?

**As indicated in our previous responses, we propose to better separate in our revised manuscript surface observations and sub-surface (folds) deductions from deeper interpretations (thrust geometries).**

**See previous answers about tectonic inversion.**

Figure 10: I'm not really convinced by this part. Lack of field data!

**Same answer as for the comments on Figures 5c and 7 above, about the frontal anticline of the Pinchal zone. Also this anticline (Chacarilla anticline) has already been documented in previous published (cited) work: Blanco and Tomlinson 2001, Armijo et al 2015, Fuentes et al 2018.**

Figure 11a: How do you explain such a steep ramp?

**Steep ramps are commonly found, in particular in the case of inherited previous structures. This is probably the case here as Mesozoic layers were deposited initially in**

extensional basins. Here "steep" is only ~40-45°, as reported in table S3... so not that extremely steep. Such dip angle is derived from the geometry of the folded layers of the eastern limb of the Chacarilla anticline, and because of this should be here taken as a minimum.

Figure 11c: It is a fault bend fold!

We do not agree: it is a fault-propagation fold, as the fault has not yet reached the surface and deformation is disharmonic with layer thickening (and folding) at the tip of the ramp. But we agree that this fault-propagation fold is not far from becoming a fault-bend fold with additional incremental slip and fault propagation.

Figure 11: How do you explain the difference in the depth of the detachment between the two cross-sections?

The two sections are ~70-80 km away from each other (Figure 1), with clearly differing stratigraphies, most probably controlled by the details in the earlier local Andean Basin (structural and stratigraphic inheritance). Also the Pinchal section is located in the immediate footwall of the Pinchal Thrust (Figure 5c), and we cannot rule out the possibility that the thrusts generating the observed folding of the Mesozoic layers are here shallower splays of the Pinchal thrust. We will add a few words on this in the revised version of the manuscript.

---

## Author Response (AR1)

Martine Simoes
Institut de physique du globe de Paris
1 rue Jussieu
75238 Paris cedex05
France
tel: +33 6 28 70 15 42
e-mail: simoes@ipgp.fr

Editor of *Solid Earth* (EGU)

Dear Editor,

Please find enclosed our revised manuscript, now entitled "A contribution to the quantification of crustal shortening and kinematics of deformation across the Western Andes (~20-22°S)" by T. Habel and co-authors, submitted to the journal *Solid Earth*.

We thank Benjamin Gerard (RC1) and Patrice Baby (RC2) for their appreciation of our work, as well as for their various constructive comments and suggestions, which helped improve our manuscript  by clarifying our arguments and their presentation. We would like to emphasize that none of our final results and conclusions were questioned.

We hereafter respond in detail to the various comments and questions addressed by both reviews:

- the reviews are entirely reported in black,
- **our responses in bold blue (with references to lines and sections of the first version of the manuscript),**
- and the corresponding revisions detailed in red (with references to lines and sections of the revised manuscript)

We also provide a marked-up manuscript specifying our revisions. Major revisions to our manuscript are highlighted in yellow in this document. Highlighted figure captions indicate figures that have been modified to follow reviewer's recommendations.
A simplified version is provided first in the marked-up PDF (pages 1-39) where only highlighted major revisions are reported. It is followed by a version where all changes are indicated (pages 40-78), as provided by *Word*. Note that the lines are not numbered the same in these two parts (for an obscure reason when using *Word*). The simplified marked-up document has a line numbering identical to that of the revised manuscript.

We hope that you'll find now our manuscript suitable for publication in Solid Earth.

Martine SIMOES (on behalf of all co-authors)

**Response to the review by Benjamin Gerard (RC1)**

**General comments:**

In this manuscript Habel et al. constrain the timing and quantify the amount of tectonic shortening in a part of the Western Andes in northern Chile. They processed high-resolution satellite images, field observations, updated geological and structural maps in order to confront this dataset to numerically-modeled balanced cross-sections of the study area.

As it has never been done before, results and discussions are of a great interests for the community working on the Andes. The Western Cordillera is often mentioned in the literature, but the shortening has never been quantified accurately. Results, based on the study of pluri- kilometer geological objects, partially fill the data gap regarding shortening rates and timings of (re)activation of structures in the Western Cordillera. These data allow to better frame the deformation of the Western Andes and should be taken into account in future attempts to restore Andean shortening rates.

I particularly appreciated the topic, the scientific approach and the efforts of the authors in order to reach their scientific objectives. This work is worth publishing. Results are discussed and satisfactorily confronted to the literature already published. I have detailed in the sections below a few points of science to clarify.

However, the presentation of the manuscript still needs some work. It is sometimes difficult to follow the reasoning. For instance, some parts of the results are interpretations and some interpretation sections include reporting of results. Some sentences are difficult to follow (especially figure captions). I would suggest to shorten them (please see detailed comments below). Some significant information are missing and some less relevant are available. The presentation of the supplementary document is quite different from the manuscript. I advise authors to homogenize the presentation.

I am not native English speaker but some of the vocabulary and grammar seem to be incorrect. I have indicated errors I was able to identify, but I advise authors to have the manuscript proofread by a specialized company or a native English-speaking colleague if possible.

I recommend a round of minor revision, especially to fix presentation issues.

**We appreciate these various positive feedbacks on our work and will hereafter explain our point of view, and the modifications and corrections we made to our manuscript to make it clearer and more easily readable.**

**Review criteria:**

*1. Does the paper address relevant scientific questions within the scope of SE?*

☞Yes. The understanding of the timing and the quantification of deformation in the Andean Western Cordillera is of a great interest for the geological community working in the Andes. Methods, results and discussion definitely fit with the scope of Solid Earth journal.

*2. Does the paper present novel concepts, ideas, tools, or data?*

☞Yes, new ideas and data are presented in this manuscript.

*3. Are substantial conclusions reached?*

☞Yes. Results have been satisfactorily analyzed, discussed, and conceptualized. The conclusions are of importance for a better understanding of the building of the Central Andes.

*4. Are the scientific methods and assumptions valid and clearly outlined?*

☞Overall, yes. Some information about the cross-section restoration modeling process are missing. Please see detailed comments below.

**See detailed answers hereafter**

*5. Are the results sufficient to support the interpretations and conclusions?*

☞Yes. Results were satisfactorily exploited to support authors' statements. The limitation associated to the results were clearly discussed.

*6. Is the description of experiments and calculations sufficiently complete and precise to allow their reproduction by fellow scientists (traceability of results)?*

☞    Overall, yes. Some information about the modeling process are still missing.

**See detailed answers hereafter**

*7. Do the authors give proper credit to related work and clearly indicate their own new/original contribution?*

☞Yes.

*8. Does the title clearly reflect the contents of the paper?*

☞Yes. The title should be even more assertive by showing that the shortening of the Western Andes estimated in this study is not so negligible at the scale of the entire Andes.

**We thank RC1 for this suggestion. It should be however reminded that we only clearly document a few km of shortening across our two field sites, in the Pinchal and Quebrada Blanca zones. Our proposal that shortening may not be that negligible across the whole western flank of the Andes derives from our reasoning when scaling our observations/results up to the whole region - and as such may be discussed and debated. We therefore prefer to keep conservative and not too assertive or provocative in the title of the manuscript.**

Title slightly modified, but in response to a comment by RC2. Now: *A contribution to the quantification of crustal shortening and kinematics of deformation across the Western Andes ( ~20-22°S).*

*9. Does the abstract provide a concise and complete summary?*

☞Yes, the abstract is complete.

*10. Is the overall presentation well-structured and clear?*

☞No, it needs a considerable amount of work, especially in the organization of the results, interpretation and discussion sections.

As further explained and detailed hereafter, we re-organized our manuscript to further separate field observations (in sections 4 and 5, for each one of the two field sites), structural interpretations and from there our various deductions on crustal shortening and kinematics (in new section 6 that combines previous sections 4.2, 5.3 and 6), before

proposing a discussion based on the regional up-scaling of our observations and deductions (section 7).

*11. Is the language fluent and precise?*

☞No, Some paragraphs are difficult to follow. I advise authors to have the manuscript proofread by a native English-speaking colleague if possible and refer to technical corrections below.

We did our best for English corrections when revising the paper.

*12. Are mathematical formulae, symbols, abbreviations, and units correctly defined and used?*

☞Yes.

*13. Should any parts of the paper (text, formulae, figures, tables) be clarified, reduced, combined, or eliminated?*

☞Yes. Parts of the manuscript and figures should be clarified. Please see detailed comments below. Paragraphs have to been clarified and shorten by removing redundant information.

**See detailed response hereafter.**

*14. Are the number and quality of references appropriate?*

☞Yes, the state of the art is almost complete and authors' results are satisfactorily confronted with the literature available in the discussion.

*15. Is the amount and quality of supplementary material appropriate?*

☞Yes. Some parts need to be clarified. Please see detailed comments hereafter.

**Specific comments / Scientific questions and issues:**

Manuscript

L22-24: I do not think that the authors can say that the shortening of the Western Andes is "negligible" even at the scale of the entire range. The study area only represents approximately 1/5 in width of the Western Cordillera; Authors should expect higher shortening for the Western Cordillera. Taking into account the authors' maximum shortening estimate, it could represent 15% of total shortening integrated to the entire Andes. Authors should be a bit more assertive about this important result because most of the deformation of Western Cordillera is hidden.

**As already pointed out, we only clearly document a few km of shortening across the two investigated field sites. Our proposal that shortening may be up to ~20-44 km across the whole Western Andes is only a deduction when interpreting the field sites as part of a thrust system and when scaling up our results to the whole western flank of the Andes.**

**~20-44 km represent 6-14% of the total 360 km shortening across the whole Andes. We agree that such values are not to be considered as negligible, in particular if we consider that they were accommodated at a time when no other main structure was active. However, we prefer to keep conservative so as to avoid unnecessary debate and controversy.**

The term "negligible" may sound negative and we changed it to " a small fraction of the total shortening". We modified slightly the sentence to clarify our point (lines 22-23)

L61-62: How much shortening has been estimated in the thrust belt framed by San Ramon fault ? Authors should add this information (if available) in order to compare it with their results. Is this tectonic shortening thick or thin-skinned, or both in Santiago de Chile area?

The San Ramon fault is the most frontal fault of the West-Andean fold and thrust belt at the latitude of Santiago (33.5°S). This structure is mostly thin-skinned and has absorbed a total of 9-15 km over the last ~20-25 Myr (Riesner et al 2017), to be compared to the total 27-42 km across the whole Andes at this latitude (Riesner et al 2018) - ie 21-55% of the total Andean shortening at 33.5°S.

However, we recall that this section of the Andes is ~1300 km southward from the sites investigated here, and is therefore not directly comparable. To avoid confusing details and keep the focus of our manuscript on the Andes at ~20-22°S, we prefer not to add these details in our introduction. The reader has all needed references in case interested.

L100: The Eastern Cordillera and the Cordillera Oriental are the same... The Cordillera Oriental is not the Interandean zone. The nomenclature should be clarified here. The Interandean zone is a transition zone between the Eastern cordillera and the Subandean zone with a specific tectonic style and specific geological units involved (Kley, 1996).

Corrected (lines 94-95)

L107-115: Authors should also discuss paleo-altimetry estimates in this paragraph, which is (at least partly) linked to the deformation. Please see Sundell et al. (2019) and others for instance. In this section, authors should also mentioned the Bolivian Orocline bending that affected the area, to highlight potential relationships with the migration of the deformation eastward (Müller et al., 2002; for instance).

We agree with RC1 that elevation is a indirect indication of deformation - and as such data on paleo-altimetry could be mentioned in this paragraph. We recall however that this section is only meant at providing a general overview on the various morphotectonic units of the whole Andes and on their timing of deformation (from regional reviews), needed to later put our results and findings in the context of the Andean orogeny. Details about paleo-altimetry or even the bending of the Bolivian orocline are somehow out of the scope of this section - and of our manuscript in general. We prefer to keep our manuscript focused and not to lengthen it with unnecessary details.

L125: Why is the structural organization become more complex southward? Please consider to add a short explanation here.

One possible explanation for the lateral structural variations along the Andes relates to structural inheritance from the earlier Mesozoic Andean basins.
Short explanation added (lines 119-120).

L226: Are variations in stratigraphic thicknesses have been taken into account (error computation in shortening estimates)? And if yes, how? Does it affect significantly the shortening estimates?

**Variations in stratigraphic thicknesses were not taken into account in computing shortening estimates. We do not have any evidence for significant variations in the case of Quebrada Blanca (Figures 9a-b). In the case of the Pinchal zone, we cannot document any such variations if existent, in particular by comparing the two limbs of the overturned syncline as the eastern limb is highly faulted and deformed.**

**We simply and conservatively indicate that this is a possible limitation of our interpretations (lines 220-221) - and a classical one in structural geology.**

L237-243: Authors should consider to briefly detail here what was the set of parameters explored for cross-section restoration? Even if it is available in the supplementary document. Also, what were the criteria to define the best solution of the forward modeling approach? Authors should had this significant information in this paragraph.

**We thank RC1 for his suggestions.**
**Our preferred solution was defined by visually comparing the modeled and interpreted structural geometries.** This is now reported (lines 245, 590), and as such we now refer to a "preferred" model (lines 249, 594) rather than to a "best fit" solution.
We added more information on the trishear modeling in the main text, instead of keeping all relevant information in the supplementary data (lines 236-244). **We recall here that our trishear model is a forward model (and not a backward restoration) and that our preferred model is possibly a non-unique solution.**

L574-582: This part should be in the method section. Not in the discussion. Furthermore, authors should had first-order missing information here, such as:

- Briefly describe what are parameters investigated (trishear).

- What were criteria to define the "best-fitting model". Are specific metrics used? if qualitative only, which parameters have been taken into consideration? Authors should describe a bit more the methodology.

**This section (section 6) is not part of the discussion (section 7), but presents the results of our modeling and interpretation of field observations.**
**We recall here that our preferred model is defined from visually comparing modeled and interpreted structural geometries, and that our preferred model may not a unique solution.**
This has been clarified in the method section (lines 231-252).

L585-586: Are there errors on cumulative shortening estimates? Did the authors test other modeling setups with similar results (or close results), which also satisfies the present-day geometry of the structures.
**We tested a wide range of parameters, within the range of values considered in the literature for these parameters (see explanation in the supplementary material,** now partly transferred to the main text in the method section**). Within this range of acceptable values, we did not find a wide range of possible solutions reproducing our interpreted structural geometries. Given this, we estimate that our shortening values are determined with an uncertainty of 0.1-0.2 km** (now indicated lines 599-601)**. We recall here that our shortening estimates mostly depend on the structural geometries to be modeled (ie as deduced from our field interpretations) rather than on the detailed model parameters** (as clarified lines 250-251, 616-617)**.**

L692-696: Exhumation and uplift are different. Authors should not compare it directly. I did not find the information about the amount of uplift related to the shortening modeling. Authors should explain how uplift has been computed and what is the value of uplift taken into account for comparison.

We agree with RC1.

In the case of the ABT system, these is no data on uplift, but rather on exhumation from thermochronology at the regional scale (Maksaev et al 1999, Reiners et al 2015). We compare these data to what can be deduced from our structural interpretations in the Pinchal zone. Our observations suggest >2.2 km of exhumation over the Pinchal Thrust, to erode the >2.2 km thick Mesozoic series and exhume the basement. Such exhumation is expected to be concomitant with overthrusting over the Pinchal thrust, as part of the ABT system. This amount of exhumation is consistent with the findings of Maksaev et al (1999) of 4-5 km of basement exhumation.

This has been rephrased and clarified in the revised manuscript (line 538-541 and 690-692)

L790-795: Authors should briefly explain hypothesis dealing about the deformation transfer from west to east in the frame of their discussion.

We do not understand the point raised here by RC1. We only aim at pointing out the fact that deformation across our field sites - and possibly across the whole Western Andean flank - seems to significantly slow down by the time deformation initiates further east (Eastern Cordillera) in the Andes.

Supporting information

Text S1: How many forward models have been run? Information partly appear in the caption of Table S1, but it has to be written in the text also (in the supplementary and in the manuscript). How was the range of parameters explored? How the authors decided to sample the range of parameters in order to cover the parameter space in the most representative way. Authors should add these information.

The number of tested models, initally indicated in supplementary material, is now also reported in the main text (lines 592-593). We also more clearly indicate that the comparison between model and structural section has been done by a visual fit (lines 245, 590), and that the range of parameters was progressively explored by trial and error (lines 245, 566+590), parameter after parameter to sense its impact on the modeled geometries (supplementary data, Text S1). We recall that we do here a simple forward model (not an inversion). We only aim at providing a viable solution (supplementary material text S1), not a bayesian perspective on these models.

**Technical corrections:**

Manuscript

L12/L14/L45: "western Andes" should be written "Western Andes" with a capital letter (w). Authors should check it in the entire manuscript.

L14-15: "Our results [...] regional data". Authors should consider to rework this sentence that is difficult to follow. I think the part "once our [...] regional data" is not necessary.

Corrected and modified

L45: Authors should consider to add a reference (e.g. Jaillard et al., 2000; Jaillard & Soler, 1996) to justify Andean mountain-building initiation.

L47-49: Authors should considered to cite only 2 or 3 articles here, in order to focus on the most relevant papers that deal with Andean deformation migration. Some of the papers cited here do not deal exclusively with this topic.

We simplified the cited references and kept only those that provide a general review on the topic of initiation of deformation and temporal evolution - and added Jaillard et al (2000) (Lines 45-46).

We also accordingly simplified the citations reported lines 98-99.

L49-53: This sentence is too long and difficult to follow. I suggest to shorten it.

Corrected and modified into 2 sentences (lines 46-48)

L50-51: To justify previous studies in the "various cordilleras to the east" authors should consider to cite Gérard et al. (2021) to support their statements.

Although this contribution is undoubtedly interesting, we prefer not to mention it in our manuscript. Indeed, it is rather a local study in Peru, and if we were to cite this work, we should also cite any other work on the Altiplano-Puna... in contradiction with the simplification requested in our citations (see penultimate comment by RC1).

L57: "locations" instead of "localities" maybe?

Sentence modified (line 51)

L67: "onset" instead of "start" maybe?

Modified (line 62)

L69: Authors should refer to Figure 1 to help the reader. "blanketing Cenozoic deposits and volcanics".

Modified (line 64)

L73: Authors should briefly describe here what kind of quantitative data have been processed.

Modified (line 68)

L86-115: This state of the art is quite complete but, it is too difficult to follow. In order to be clearer, I would suggest to start the geological framework from the full Andes' scale and then to focus on the study area. That is to say, to keep L86-90 as it is. Then to put L97-115 in a row and next, focusing on the study area (L90-96).

We do not really get the point raised by RC1. Indeed the first part (lines 86-90) refers to the overall Andean margin, then the second one (lines 90-96) to the proper Andes and their various units at the large-scale, before indicating the temporal evolution of deformation of these various units from the literature (lines 97-115). As such we already follow the recommendation of RC1 by progressively zooming into our study area. It would make no sense to indicate the temporal evolution of Andean morphotectonics units (lines 97-115) before even defining these units (lines 90-96).

We slightly modified this section, so as to mention the topographic characteristics of the Andes, their crustal thickening and shortening (lines 88-91) after first defining the Andes along the Andean margin (lines 81-87).

L91-92: "elevation" instead of "altitude". Please check throughout the manuscript.

Corrected (line 82) and verified throughout the manuscript.

L95-96: Merge "Figure 1" into the bracket located before. For instance: (...Atacama Bench; Figure 1). Please check throughout the manuscript.

Corrected (line 87)

L98: Authors do not need to quote "following here the terminology of ..." I would suggest just to cite the literature. It will be easier to read.

Corrected (line 87)

L118-119: Authors usually described units from west to east before. Here it is reversed. I suggest to keep this west-east logic, it will be easier for the reader.

We agree with RC1. The logic has been to describe morpho-structural units from west to east before, ie from the subduction trench to the South American continent. We have tried in our earlier versions of the manuscript to keep this geographic logic, but found it to be difficult as it supposes to describe stratigraphic units from the youngest to the oldest, ie the reverse of what is usually done. To keep a stratigraphic (ie temporal) logic from the oldest units to the youngest ones, here but also later in the manuscript when providing the detailed stratigraphy of the investigated field sites, we need to keep describing from east to west our region of interest.
Text modified to explain this stratigraphic logic (lines 111-112)

L124: Please add the city of Calama on Figure 1.

This correction cannot be implemented, as the city of Calama is out of the map of Figure 1. This is why we indicate its latitude in the main text.

L149-153: These sentences are confusing. Please consider to rework this section. There are too many comas.

Simplified (lines 143-147)

L150: Where are located these Sierras on Figure 1? Sierra del Medio and Sierra de Moreno? Authors should consider to add these information on Figure 1.

These sierras are also slightly south of the map of Figure 1, except for part of the Sierra Moreno  (now reported on Figure 1) where the Pinchal zone is located. This part of the text has been simplified (lines 143-147) and the comment does not apply anymore.

L189: Authors should add a reference for "European Pléiades satellites images".

Rather than adding a reference in the text, we added the corresponding web site of the spatial program (https://earth.esa.int/eogateway/missions/pleiades) in the data availability statement (line 824).

L191: A space is missing between "2" and "m".

Corrected (line 186)

L249-251: This is a paragraph for a discussion. Authors should consider to put this section in the interpretation/discussion section and not in the result section.

We understand this comment by RC1. However, because the stratigraphic order observed in the field is the reverse of that published in the earlier geological map of the area (Skarmeta and Marinovic, 1981), with implications in terms of the sense of local structures (syncline instead of anticline) and in terms of regional stratigraphy with the possibility of Trias units, we prefer to mention this particular context early in our presentation of the Pinchal area. In contrast, in the case of the Quebrada Blanca area, we rely on earlier stratigraphic and geologic work.

L254-255: "The first order [...] units". Authors do not need to introduce figures like in a book chapter or in a thesis or report. Just to smoothly integrate it in the text to argue their points. This observation is valid for other sections in the manuscript.

Corrected (line 266-268) and verified throughout the manuscript.

L255-256: To my opinion, and this is valid for the entire manuscript, it would be easier for the reader if authors should refer to the exact figures in the supplementary document (in line with their statements; Figure S1 for instance) instead of just reporting that there is supplementary material here and there.

We do not understand this comment by RC1... as this is what we exactly do by refering to the exact figure to be found in the supplementary material. We verified that this is done throughout the whole text.

L314-315: Authors should not introduce figures at the beginning of sections, just integrate it in the text, according to their statements. It will be easier to read.

Corrected (line 316)

L395: The section 4.1.3 is an interpretation section. Authors must not merged results and interpretation this way. 4.1.3 part has to appear later in the manuscript. Authors should consider to had this section in the Discussion/Interpretation section.

Now section 4.3

We understand this comment by RC1. However, we believe that we need to discuss our field observations in the Pinchal zone in light of existing geological maps and regional stratigraphic knowledge, in particular because these are in contradiction with previous published work, before presenting anything about the Quebrada Blanca. As we present

two different field sites with various degrees of confidence in previous work in light of our field observations, we find it confusing to strictly follow the classical manuscript organisation "data-results-interpretations-discussion" that would imply repetitions in going back and forth between the two field sites. At some point we need to discuss the strength of our field observations before presenting our interpretations, and this for each site, before combining our results together at the regional scale and discussing them. We therefore prefer to keep here this discussion on the strength of our stratigraphic considerations, before getting into the resulting structural interpretations.

L416: Same comment as above. Authors cannot present results and interpretations from Pinchal area and then present results and interpretations from Quebrada Blanca. The organization of the sections has to be reviewed. Authors should present all observations, stratigraphic, tectonic and modeling results for both area; and then interpret and discuss it later in the manuscript. Parts of section 4.2 has to be transferred in a distinct interpretation section (section 6 for instance, with new sub-sections for Quebrada Blanca and for Pinchal).

As the Pinchal and Quebrada Blanca zones are quite different in terms of scale, stratigraphy or structure, in particular with respect to previous works, we preferred to present observations and interpretations for each zone, separately, for clarity purposes and to avoid otherwise inevitable repetitions.

We understand the recommendations of RC1, but for these previous reasons, we would like to keep presenting and discussing the two zones separately, one after the other, at least where it comes to observations.

However, following these suggestions, we revised the structure of the manuscript as follows:

1) first present stratigraphic and structural observations only, for the Pinchal and Quebrada Blanca zones in two separate sections (new sections 4 and 5),

then 2) present all structural and kinematic interpretations of both zones in section 6, including the presentation of the interpreted cross-sections and the first estimates of shortening by folding (revised section 6.2), before getting into the kinematic trishear modeling (sections 6.3 and 6.4).

before 3) a general discussion (section 7) where our results and interpretations are up-scaled to the whole western Andean flank.

Together with these changes in the organization of the manuscript, we also modified figures 5c and 9c. These figures (associated to sections 4 and 5, see point 1 above), now only report surface geological observations and the sub-surface fold-form that can be deduced from there. The deeper interpretation, in terms of fault geometry and associated shortening is not reported anymore here, but appears later in Figure 11 (associated to the interpretative section 6, see point 2 above).

L417-418: In order to get straight to the point, these introductory sentences are not necessary. It makes the reading difficult with unnecessary information. Authors should simply refer to figures and tables to support their statements.

Corrected (line 458)

L418: There is an issue in results presentation. Authors should not refer to table 1 (presenting shortening values computed from cross-section restoration) before figures presenting cross-sections restoration/line-length balancing (Figure 11). It is very difficult to follow.

With the revised organization of the manuscript, this table appears now later in the manuscript, together with interpreted sections and modeling (section 6).

L425: For this assumption, authors should consider to add a reference to support their point.

Corrected (line 518)

L453: Results for Quebrada Blanca area should appear before interpretations from Pinchal area.

L525: Similar comment from above. Parts of this section are interpretations, and it should appear in a distinct and independent section later in the manuscript to avoid merging results and interpretation in the same section.

See previous answer about the proposed revision of the manuscript structure.

Our proposed revision follows in its main lines this particular recommendation. In particular here, we now provide field observations for Pinchal (section 4) and then for Quebrada Blanca (section 5), before getting into structural interpretations and shortening assessments (section 6).

L554: It is not necessary to refer to previous sections: "section 4.1". Authors should remove this information.

Corrected (line 500).

L567: Section 6.2 is a mix between trishear method, results, interpretation and discussion. Results should appear earlier in the manuscript. Authors should review the structure of the manuscript.

Trishear results cannot appear earlier in the manuscript, as we need to present first the structural interpretations (now in section 6.2) of our field sites before modeling them. We agree that part of the text is here a repetition of the trishear method. This has been corrected by complementing section 3.4 (method) and by focusing only here on the model implementation and results. Some points need however to remain discussed here - such as the different spatial scales of the two sites, the meaning of these results with respect to our structural interpretations - so as to keep the discussion section (section 7) focused on up-scaling our results and interpretations and discussing their regional implications.

L574-582: This part should be in the method section. Not in the discussion. Furthermore, authors should had first-order missing information here, such as: - Briefly describe what are parameters investigated (trishear). - What were criteria to define the "best-fitting model". Are specific metrics used? if qualitative only, which parameters have been taken into consideration? Authors should describe a bit more the methodology.

As stated previously, we agree with RC1 and part of the text has been moved to the method section (section 3.4). Also to complement what is written about the model

**implementation,** we moved to this same section some key information initially provided in the supplementary material: what parameters investigated, how the range of tested values has been determined, etc. Finally, as proposed previously, the "best-fitting model" is now referred as the "preferred model" as we do a visual fit between model and structural section, and not a mathematical fit of the two geometries.

L617: "elevation" instead of "altitude".

L797-802: There are too many information here. Authors should consider to shorten this part to focus on first-order information.

L798: Authors may add "Chilean" before "Andes" to help the reader.

L855: The bibliography is not homogeneous. Some titles are in capital letters, typos are present. Please fix issues.

These corrections were all implemented in the revised manuscript.

Figures

Figure 1: The text in the figure caption is not very clear. Authors should consider to rework the caption with smallest sentences. Authors should consider to add the topographic cross-section location on the geological map. Authors should consider to add major tectonic features location on the topographic cross-section to identify potential relationships between faults and topography. Cordillera Domeyko appears in the caption but does not appear on the geological map. Please add it. To simplify the caption, authors should remove abbreviation explanation for ABT, WAT, WATS, Cz, Mz, Px-Pc as it is already written in the figure. Or delete the abbreviation in the figure et keep it in the caption. In the inset, authors should consider to add an arrow to show the direction of convergence (Nazca vs. South American plate).

The suggested corrections were implemented, except for the location of major tectonic features along the topographic profile. **Indeed, some of these structures are oblique to the profile, some are more visible to the south than to the north: reporting them along the topographic profile could be therefore meaningless and confusing.**

Figure 2: Labels "a" and "b" are on the figure but not in the caption. What is the difference between the two pictures ? Authors should mayve select one picture to avoir repetition of information. Figures 2 and 3 could be merged.

**We thank RC1 for these suggested corrections,** which were implemented in the revised manuscript. We chose only one of the two field pictures and completed it to have all needed information in one single picture.

**As of merging figures 2 and 3, we do not agree with this idea as these two figures are quite different: one represents a landscape view (Figure 2), and the other the simplified stratigraphic column of the site (Figure 3).**

Figure 3: Approximative thicknesses of units should be indicated on the figure. Even if the log is not scaled. In the caption: "By analogy to regional description". Please cite the literature here.

We already tried to report the thicknesses of the units, but because the log is not scaled this is not easy to implement for readability purposes, there would be numbers everywhere. All thicknesses are reported in the text as indicated in the caption. Instead of citing the literature corresponding to regional stratigraphic descriptions, we rather refer to the text to avoid lengthening unnecessarily the caption as RC1 usually asks to keep the text as simple as possible in figure captions.

Figure 4: It would be maybe better to call figure 4 before. At the beginning of section 4, to better organize this section. Authors do not need to add information twice (in the figure + in the caption). Field picture locations for instance.

The structural map of Figure 4 cannot be called before the structural descriptions, ie not before section 4.1. The stratigraphic description of section 4.1 is not related to this map but rather to Figure 3.

Any redondant information between figure and caption has been removed. In the case of located field pictures, the associated number still needs to be explained, indicating the figure to which it refers.

Figure 5: Authors should not repeat three times "Figure 4" in the caption. It represents a repetition of the same information. It is difficult to read... I suggest to find a way to shorten these sentences. There are some French words in the figure: "Cisaillement, Schisosité". Please translate it.

Repetitions have been corrected in the revised manuscript.

As of the French word "Cisaillement-Schistosité".. well, this is in fact the technical word used even in English (which explains the "C-S" abbreviation used for the fabric), as some other structural words such as "décollement", etc.

Figure 7: L385/389: In the caption, the word "violet" should be replaced by "purple"? Authors should indicate dip marks of strata on the pictures to help the reader.

The word "violet" has been changed to "purple".

Dip marks were also implemented to help reading the figure.

Figure 10: This figure is not well structured. Authors should add a symbol (on the figure) to show the relationship between picture a and b. Authors should also find a way to remove blank space at bottom right of the figure.

We modified the figure organization to balance the blank space. We now also better illustrate the link between figures a and b with an arrow.

Figure 11: This Figure and associated results have to be presented before in the result section. Not in an interpretation part. L596-603: This text is not needed in the figure caption and should be written in the main text, otherwise the caption is too hard to follow. Parts of the caption are already displayed on the figure. To shorten the caption, authors should consider to remove duplicate information.

We do not agree with RC1. This figure can only appear with section 6 as it presents and uses results from trishear modeling, together with geological time constraints. Therefore it must be in an interpretation section, even though this interpretation integrates modeling results.

**We recognize that time benchmarks, which provide key time constraints for our kinematic interpretation, are already explained in the main text.** We simplified the caption, but kept some explanation about the key time lines that appear in the figure as these are important and to understand the figure - and our final conclusions.

Supporting information

The style of the supplementary document is quite different from the manuscript. I advise authors to homogenize the presentation. English has to be proofread. In the manuscript, authors should refer to the exact figure label to clarify the text. Generally, figure captions should be shorten in order to optimize them.

We homogenized the format of the supplementary material with that of the main text**. We recall here that there is no indication of a specific format to be followed for supplementary material in the author guidelines of Solid Earth.**

English has been verified**. As of referring to the exact figures in the main text, this was already the case in the initially submitted manuscript.**

Section (1)

Figure S14 appears before Figure S13 in the text. Authors should check the order.

This has been verified, we thank RC1 for pointing this.

Figures S15 and S16 are not introduced. Authors should consider to add this information.

Corrected

Figure S4: Aguilef et al. 2019 citation in the caption is not referenced in the bibliography at the end of the supplementary document.

Corrected

Figure S11: In the figure, authors should correct "bancs" into "bed" or "strata". And "Calcareous" into "Limestone".

Corrected

Figure S14: I do not see "view directions" on the map. "Field picture" information appears twice in the legend and in the caption. To help the reader, authors should remove one of this information. "Quebrada" is not only Chilean but used in all the Andes. Authors should write "Spanish word for...". This latest observation is also valid for Figure 4.

Corrected

Text S1: Some of the information presented here have to be written in the method section of the manuscript to help the reader. Please see detailed comments above.

Some key information, initially only in supplementary material, appears now in the method (section 3.4) and results (sections 6.3 and 6.4) sections of the main text: number of models, how the preferred model was established, the kind of testes parameters...

Table S1: This table should be in the method section of the main manuscript. In the caption, authors should remove unnecessary parts (Best results [...] Allmendinger, 2002) and add them instead in the text S1, or even in the manuscript.

We agree with RC1 that this information is important, however not crucial for the main message conveyed in our manuscript. To avoid unnecessarily lengthening the manuscript, we'd rather keep this table in the supplementary material. The table caption is a complement to text S1 so that we did not reduce the information provided here.

Data Set S1: Parts of this text (although nice) are not appropriate for publication in a scientific journal. Although I love to imagine geologists in the field under a beautiful starry night, this is not required here. Authors must remain factual.

We liked the idea of letting the authors free for a couple words of fantasy - in the supplementary material and not in the scientific argumentation presented in the main text. We sincerely hope that the readers and editor will let us keep this slight departure with strict uses since it has no consequence on the science.

Benjamin Gérard
**References cited in this review letter**

Gérard, B., Robert, X., Audin, L., Valla, P. G., Bernet, M., & Gautheron, C. (2021). Differential Exhumation of the Eastern Cordillera in the Central Andes: Evidence for South-Verging Backthrusting (Abancay Deflection, Peru). *Tectonics*, *40*(4), 1–29. https://doi.org/10.1029/2020TC006314

Jaillard, E., & Soler, P. (1996). Cretaceous to early Paleogene tectonic evolution of the northern Central Andes (0-18 degrees S) and its relations to geodynamics. *Tectonophysics*, *259*(2), 41–53. https://doi.org/10.1016/0040-1951(95)00107-7

Jaillard, E., Hérail, G., Monfret, T., Diaz-Martinez, E., Baby, P., Lavenu, A., et al. (2000). Tectonic evolution of the Andes of Ecuador, Peru, Bolivia and northermost Chile. *Tectonic Evolution of South America*, 481–559.

Kley, J. (1996). Transition from basement-involved to thin-skinned thrusting in the Cordillera Oriental of southern Bolivia. *Tectonics*, *15*(4), 763–775. https://doi.org/10.1029/95TC03868

Müller, J. P., Kley, J., & Jacobshagen, V. (2002). Structure and Cenozoic kinematics of the Eastern Cordillera, southern Bolivia (21°S). *Tectonics*, *21*(5), 1-1-1–24. https://doi.org/10.1029/2001tc001340

Sundell, K. E., Saylor, J. E., Lapen, T. J., & Horton, B. K. (2019). Implications of variable late Cenozoic surface uplift across the Peruvian central Andes. *Scientific Reports*, *9*(1), 1–12. https://doi.org/10.1038/s41598-019-41257-3

**Response to the review by Patrice Baby (RC2)**

The paper of Habel et al. presents a structural study of two sites of the western Andes in Chile (20-22°S), where the authors use numerical trishear forward modelling to evaluate minimum horizontal shortening and analyse the kinematic evolution of two fault-related anticlines.

Before being published, this paper must better document the structural observations, which are not always convincing (see below). These data can be used to construct in each area a balanced section to validate structural interpretations and calculate shortenings more rigorously. The authors need to better explain why they chose a fault propagation fold model rather than a tectonic inversion model in their structural interpretation.

These various issues on why we conservatively can't do much better on building cross-sections and calculating from there shortening, or the chosen structural interpretation (tectonic inversion vs. fault propagation fold) will be detailed hereafter. To summarize here in a few words:

- There are no constraints on the structure of the footwall of the faults inferred to generate the folds observed from surface geology. Given this, a more rigorous structural section and the associated shortening can't be proposed. Because of this, we do only propose a possible interpretation and discuss its implications. This is now better explained lines 222-225. To better separate observations from more interpretative outcomes, Figures 5c and 9c now only report field observations, with no inferences at depth - these come later, on Figure 11 when incorporating our trishear modeling results.

- We do not oppose tectonic inversion to fault-propagation folding. In fact, the investigated regions were forming basins during the Mesozoic, and these basins have been inverted during the Cenozoic to form the structures documented here along the Western flank of the Andes. This inversion most probably re-used faults that structured these basins, variably along-strike, and these inverted faults may have formed the fault-propagation folds documented in our work. However, the main question relates here on whether these faults connect or not at depth onto a common detachment (as proposed by e.g., Victor et al 2004 and Armijo et al 2015), or whether they are independent from each other (as proposed in Fuentes et al 2018 and Martinez et al 2021, even though their later interpretation - in Fig 37.9 of Martinez and Fuentes 2022 - gets back to connecting these faults at depth onto an east-dipping detachment). Surface geology of the investigated field sites does not allow to choose between either one of these two models. This is now better explained in section 2.2.2 (lines 155-167) and 7.2.1 (lines 722-725). However, from regional considerations we favor the interpretation that these faults connect at depth onto a common detachment (see revised section 7.2.1, lines 722-731).

Finally, we recall that our proposed modeled structural interpretation and line-length balancing provide upper and lower bounds on shortening estimates, respectively, as now clarified lines 620-622.

**GENERAL COMMENTS:**

The title must be modified. The studied areas are too small to represent the entire western Andes. It would be interesting to locate the two sites on a regional cross-section through the western Andes.

**We agree with RC2 that our two areas are too small to represent the entire Western Andes - this is largely recognized throughout the manuscript. However, based on our findings from these two limited areas, we discuss the implications of these findings for shortening and kinematics of deformation at the scale of the entire Western Andes, from a regional reasoning.**

**The fact that we mention that our work is only "a contribution to the quantification of crustal shortening" does imply that we provide key additional data, not that we solve the problem at the scale of the entire Western Andes.**

Title slightly modified to make it clearer: *A contribution to the quantification of crustal shortening and kinematics of deformation across the Western Andes ( ~20-22°S).*

**A regional general cross-section of the region added to Figure 1 would allow for structurally locating the investigated field sites. A section inspired from that shown in Figure 13 of Armijo et al 2015 would be appropriate. However, providing this section early in the manuscript may give the wrong impression that our interpretation is biased, and give room to unneeded criticism, even before we put forward our various arguments. Later, this section would only be schematic as we only have field data from limited outcrops. Building a new section of the western Andean flank would need more observations and discussions that are beyond the scope of the present paper which is focused on the frontal basement thrust (ABT) and structures just west of it. Our paper is already quite long and detailed. As such, we decided not to add and discuss such a large scale section.**

The stratigraphic and geologic background (2.2.1) needs a figure with a synthetic stratigraphic column.

**This is already schematically provided in the legend of Figure 1, which corresponds to this part of the main text. As this is a very general regional overview, we do not see the need for more at this stage.**

Structural and kinematic context (2.2.2):

In their last paper, Martinez and Fuentes (2022)(https://doi-org.insu.bib.cnrs.fr/10.1016/B978-0-323-85175-6.00037-7) show the importance of tectonic inversion of the Jurassic rift in this region. The analysed seismic sections are just west of the study areas of this paper and must be taken into consideration and discussed.

**We already discussed the work and results of the group of Martinez and Fuentes, by citing their original work and data (Martinez et al 2021 and Fuentes et al 2018) rather than this later summarizing paper that does not provide any new interpretation of the seismic profiles.**

**As mentioned earlier, tectonic inversion is not in opposition with our interpretation. In fact, the continental and marine Mesozoic series described in our field sites were deposited in basins, that were lately inverted during Cenozoic compression to form the structures of the Western flank of the Andes.**

The two main previous opposing views of these structures were about their geometries at depth. On one hand, Martinez et al 2021 and Fuentes et al 2018 propose that they are single planar faults not connected to each other at depth, minimizing shortening estimates. On the other hand, Victor et al 2004 and Armijo et al 2015 propose that these faults form a thrust system connected at depth onto a common decollement, dipping eastward beneath the Andes. Seismic profiles to the west of our field sites are too poorly resolved at depth - let's be frank on this! - and field observations are too sparse and local to favor one or the other model. This is now clarified in the revised version of the manuscript (lines 155-167).

However, as already discussed in Victor et al 2004, and considered by Armijo et al 2015, only the thrust system model is able to provide a satisfactory structural framework for the large-scale structural organisation (deep basement to the east, shallower Mesozoic and Cenozoic units to the west) and topography (high to the east, low to the west, with a westward dipping continuous slope) of the Western Andes. Interestingly, in their latest interpretation, Martinez and Fuentes 2022 propose to connect these various faults at depth onto an east-dipping detachment (See their figure 37.9).

To summarize, only from regional considerations can we favor the interpretation of a thrust system, as indicated in our final discussion (section 7.2.1, slightly modified to clarify this reasoning, and separate local observations from regional interpretations).

**Data and structural observations:**

It is important to better document the field data. For example, it is necessary to localise the field structural data in the structural map of Figure 4 to validate the cross-sections construction and structural interpretations.

All field data are reported on the map of Figure 4 and on the field sections of Figure 5. We cannot place on the map all measurements for obvious readability reasons, this is why we represented (some of) them along the field sections of Figure 5 - a section that is located on the map. We remind here that additional field observations are located on the map of Figure S14.

We now report some of our structural measurements (either from field measurements or extracted from 3D mapping) to the structural map of Figure 4. Those now reported are meant to illustrate the asymmetry of the documented folds.

We would like also to underline the fact that field data do not resume, to our sense, only to numerous analytical field measurements of strike and dip angles, but also encompass large-scale landscape observations - illustrated in the various field pictures provided in the main manuscript and in its supplement.

Field pictures interpretations must be also validated by field data. These field data, as structural dip measurements, must be placed on the pictures. I am not at all convinced by the structural interpretation of the picture in Figure 7b. I can't really see the axis of the anticline of the Quebrada Tambillo, which is a key element of the structural interpretation. This picture interpretation must be absolutely validated by field measurements.

We chose to show field pictures to depict the various observed structures, for instance with the general trend in dip angles on either side of axial planes. This is to our sense

much more meaningful here than numerous local field measurements for our interpretations at the scale of the field sites. Landscape views are true field data, even though qualitative. In figure 7b for instance, the axis of the anticline is defined by the change in the dip orientation of layers, with layers dipping to the east (left of the figure) on one side and layers dipping to the west (right on the figure) on the other side, whatever the absolute values of the dip angles, and whatever the number of dip angle measurements.

We added additional information on the field pictures, whenever appropriate and meaningful, either from direct field measurements, or from the projection of mapped 3D-layers on satellite images and DEMs.

The structural map of the Quebrada Blanca zone shows structural dips values, which is not the case for the structural map of the Pinchal area. I understand that strike and dip measurements are extracted from 3D mapping. These 3D mapping and data extraction must be documented with some detailed illustrations.

See our previous answers about representing structural measurements on the map of Figure 4 (Pinchal Zone). A detailed illustration of the structural inferences that can be made from 3D maps is already provided in the surface section of Figure 5a.

**Structural interpretations:**

I don't understand why the authors didn't try to construct balanced cross-sections, the best way for thrust system modelling and calculation of shortening. The proposed interpretations are not geometrically validated. The footwalls of the thrusts have not been constructed (?).

The footwalls of the thrusts are not drawn in our sections of Figures 5 and 9, as we recognize not to have any indication on their structure. The structure used to build our trishear models is just a proposition, as indicated in the text. As such no definitive balanced section can be honestly proposed just from our field observations.

This is now better explained in the text (lines 222-225, 531-532, 553-554)

The authors propose a model of fault propagation fold (or fault bend fold (?)) for each section. Why? Why not a tectonic inversion? How do you explain such a steep frontal ramp in the cross-section of Figure 9C? Are there lithologies compatible with the levels of detachment?

As already indicated, tectonic inversion is not to be opposed to fault-propagation folding (see previous answers). We agree that structural inheritance from the earlier Andean Basins may have played an important role in localizing the thrusts at depth, or in controlling their steep geometry.

This said, the observed folding is most simply explained by fault-related folding. Fault-propagation folding is favored here as the faults do not reach the surface, and as small-scale folds at the front of the western anticlines are possibly indicative of disharmonic folding (and therefore internal deformation) at the tip of an upward propagating ramp.

As of the lithologies at depth that would favor a decollement level, as can be deduced from our cross-sections (Figures 5c, 9c and 11), these stratigraphic levels do not crop out in the investigated areas and we cannot tell more on the subject.

Our manuscript has been re-organized so as to better clarify these points. We now better separate field observations (revised sections 4 and 5, and updated figures 5c and 9c), modeling and local kinematic considerations (section 6, with Figure 11) and our choices of interpretations from regional considerations (in particular in section 7.2). But in any case this does not question our final conclusions and the field data provided.

The calculation of shortening is confusing ("Folding" + "Folding + thrusting"(?)).

We moved to section 6 all the structural reasoning leading to shortening calculations. By having all this together in one section, we hope that these various results are much more understandable now.

The discussion would require an integration of results in a regional cross-section through the Western Andes.

**This has already been discussed previously.**

**My detailed comments are highlighted in the attached pdf version.**

Please also note the supplement to this comment:
https://egusphere.copernicus.org/preprints/2022/egusphere-2022-629/egusphere-2022-6 29-RC2-supplement.pdf

**We hereafter copy the various comments found throughout the reviewer's annotated PDF. Please note that some text sections were highlighted but not commented by the reviewer.**

Line 1: The studied areas are too small to represent the entire western Andes unit
**See previous answer to the comment on the title**
Modified to ""investigated field sites" (line 14)
Line 17: They are not really restored
**Right, since we do not model the entire sections.**
This has been corrected to "From our interpreted sections " (line 17)

Lines 47-49: Add: Baby et al., 1997
https://doi.org/10.1130/0091-7613(1997)025<0883:NSCTCT>2.3.CO;2
and Rochat et al., 1999
https://www.researchgate.net/publication/233713934_Crustal_balance_and_control_of_the_er osive_and_sedimentary_processes_on_the_Altiplano_formation
Lines 101-102: Baby et al. (1997):
https://doi.org/10.1130/0091-7613(1997)025<0883:NSCTCT>2.3.CO;2
Lines 104-106: Baby et al. (1997):
https://doi.org/10.1130/0091-7613(1997)025<0883:NSCTCT>2.3.CO;2

**We recall that RC1 recommended to limit here the citations only to the few most significant ones.. among the numerous possible publications from the extensive literature on the subject of Andean shortening across the whole range!**

Figure 2: It would be appropriate to represent the Pinchal thrust on both images.

We agree that this could be a good idea, even though the aim of this figure is to present first-order observations on the stratigraphic organization at the scale of the landscape. We tried to add the Pinchal Thrust but we did not find a satisfactory solution that would not load too much the figure - in particular when only keeping only one of the two field pictures as suggested by RC1. Therefore, this suggestion was not implemented.

Figure 3: It is hard to imagine that there are no more precise ages in the bibliography with all these fossils and volcanic levels.It would be appropriate to indicate on the figure more precise ages such as 27-29 Ma at the base of the Cenozoic, or the Triassic and Jurassic (Majala o Chacarilla Fm?).

We recall that the Pinchal zone is really remote and not easily accessible. The only geological map of the area dates back from the early 1980's (Skarmeta and Marinovic, 1981) and we believe from our field experience of this site that their stratigraphic correlation was done from an a priori knowledge of the regional stratigraphy rather by actual dating of the local layers.

Precise dating remains to be done here and we agree that there is a good potential for this. The SERNAGEOMIN (Geological and Mining service of Chile) will be working on this most probably over the next years. This is out of the scope of this manuscript.

Figure 3: Locate the décollement of the cross-section in Figure 5c?

This decollement is a structural interpretation proposed from our field observations, not a field observation and should therefore not be placed on the stratigraphic column of Figure 3.

Also, as suggested by our sections (former Figure 5c, now Figure 11ab-c), the proposed decollement level does not crop out- and cannot therefore be represented on the log where only the series observed in the field are reported. We corrected the caption of Figure 3 to emphasize the fact that only field observations (at the surface) are reported.

Figure 5c: I am not convinced by this anticline. Field data are lacking

See comment above about field data and lanscape views.

Figure 5c: This thrust geometry is very arbitrary. The footwall of the thrust must be constructed to geometrically validate the interpretation. Is there a lithology compatible with this level of décollement? Why not tectonic inversion?

To clarify our point, we now better separate field observations, sub-surface and deeper interpretations. Here, we now only report sub-surface observations on Figure 5c, and leave deeper interpretations for Figure 11.

As already mentioned, we have no direct clue about the structure of the footwall from field geology or even geophysics - this is why we had left it blank. Also, we already explained earlier why the tectonic inversion hypothesis is not to be opposed to fault-related folding. This has been further clarified in the text.

Figure 6: You must indicate values of the structural dips collected on the field and show them on the photo. Scale?

We added an approximate scale on Figure 6a - there is already a scale (Swiss knife) on the picture of Figure 6b). Dip measurements could be added on Figure 6a, but would not be meaningful here, at this scale, in the highly deformed footwall of the Pinchal Thrust. We only added this information on Figure 6b.

Figure 7: You must indicate values of the structural dips collected on the field and show them on the photo. Scale?

We added dip angles, either measured in the field or deduced from 3D mapping on Figure 7. A scale is also now reported, it is approximative because of perspective effects (indicated in figure caption).

Figure 9c: I'm not really convinced by this part. Lack of field data!

Well... the small-scale folds are illustrated in the field picture of Figure 10b.... as indicated in the boxes of Figures 9a-b. To our sense, not much to be added here without a more precise comment.

Figure 9c: This thrust geometry is very arbitrary. The footwall of the thrust must be constructed to geometrically validate the interpretation. Are there lithologies compatible with these level of detachment? Why not tectonic inversion? How do you explain such a steep ramp?

As indicated in our previous responses, we better separate in our revised manuscript surface observations and sub-surface (folds) deductions - as reported here, in particular in revised figure 9c - from deeper interpretations (thrust geometries) - as now only reported in Figure 11.

See previous answers about tectonic inversion and the geometry of faults at depth.

Figure 10: I'm not really convinced by this part. Lack of field data!

Same answer as for the comments on Figures 5c and 7 above, about the frontal anticline of the Pinchal zone. Also this anticline (Chacarilla anticline) has already been documented in previous published (cited) work: Blanco and Tomlinson 2001, Armijo et al 2015, Fuentes et al 2018.

We added some structural measurements, following comments on other field pictures.

Figure 11a: How do you explain such a steep ramp?

Steep ramps are commonly found, in particular in the case of inherited previous structures. This is probably the case here as Mesozoic layers were deposited initially in extensional basins. Here "steep" is only ~40-45°, as reported in table S3... so not that extremely steep. Such dip angle is derived from the geometry of the folded layers of the eastern limb of the Chacarilla anticline, and because of this should be here taken as a minimum.

Figure 11c: It is a fault bend fold!

**We do not agree: it is a fault-propagation fold, as the fault has not yet reached the surface and deformation is disharmonic with layer thickening (and folding) at the tip of the ramp. But we agree that this fault-propagation fold is not far from becoming a fault-bend fold with additional incremental slip and fault propagation.**

Figure 11: How do you explain the difference in the depth of the detachment between the two cross-sections?

**The two sections are ~70-80 km away from each other (Figure 1), with clearly differing stratigraphies, most probably controlled by the details in the earlier local Andean Basin (structural and stratigraphic inheritance). Also the Pinchal section is located in the immediate footwall of the Pinchal Thrust (Figure 5c), and we cannot rule out the possibility that the thrusts generating the observed folding of the Mesozoic layers are here shallower splays of the Pinchal thrust.**

We added a few words on this (lines 609-615).

---

## Referee Report (RR1)

Initial Title: The western Andes at ~20–22°S: A contribution to the quantification of crustal shortening and kinematics of deformation

Revised Title: A contribution to the quantification of crustal shortening and kinematics of deformation across the Western Andes (~20–22°S)
Author(s): Tania Habel et al.
MS No.: egusphere-2022-629
MS type: Research article

Second round of revision. Evaluation of the revised manuscript.

**General comments:**

Habel et al. addressed satisfactorily comments and remarks after the first round of revision. I have no more comments on the science presented in this manuscript (interpretations and discussions). The presentation of the modeling method (trishear) has been improved by adding necessary details (parameters explored, methods used to identify "preferred" models) accordingly. These information make the interpretation section more robust. Suggestions for semantic and visual improvements have been taken into account accordingly. The supplementary document follow editorial guidelines accordingly now. The English has been improved. Grammar mistakes that I was able to identify have been corrected. After this second round of revision, I recommend to accept the manuscript as it is.

**Review criteria of the revised version:**

1. *Does the paper address relevant scientific questions within the scope of SE?*

→ Yes. The understanding of the timing and the quantification of deformation in the Andean Western Cordillera is of a great interest for the geological community working in the Andes. Methods, results and discussion fit with the scope of Solid Earth journal.

2. *Does the paper present novel concepts, ideas, tools, or data?*

→ Yes, new ideas and data are presented in this manuscript.

3. *Are substantial conclusions reached?*

→ Yes. Results have been satisfactorily analyzed, discussed, and conceptualized. The conclusions are of importance for a better understanding of the building of the Central Andes.

4. *Are the scientific methods and assumptions valid and clearly outlined?*

→ Yes. In this revised version, improvements have been made to explain and to justify the methodology.

5. *Are the results sufficient to support the interpretations and conclusions?*

→ Yes. Results were satisfactorily exploited to support authors' statements. Limitations associated to the results are clearly discussed.

6. *Is the description of experiments and calculations sufficiently complete and precise to allow their reproduction by fellow scientists (traceability of results)?*

→ Yes. Improvements have been made to explain and to justify the methodology, especially regarding the modeling process.

7. *Do the authors give proper credit to related work and clearly indicate their own new/original contribution?*

→ Yes.

8. *Does the title clearly reflect the contents of the paper?*

→ The revised title reflects the content of the manuscript.

9. *Does the abstract provide a concise and complete summary?*

→ Yes, the abstract is complete.

10. *Is the overall presentation well-structured and clear?*

→ Yes, in this revised version, improvements have been made in the organization of the sections. The manuscript is clearly argued and articulated in a logical way. Arguments of the authors on manuscript organization are convincing in the context of this study.

11. *Is the language fluent and precise?*

→ The manuscript is easier to follow in comparison to the initial version. Improvements have been made accordingly.

12. *Are mathematical formulae, symbols, abbreviations, and units correctly defined and used?*

→ Yes.

13. *Should any parts of the paper (text, formulae, figures, tables) be clarified, reduced, combined, or eliminated?*

→ No more in this revised version. The manuscript is clear, issues pointed out have been taken into account and errors fixed accordingly.

14. *Are the number and quality of references appropriate?*

→ Yes.

15. *Is the amount and quality of supplementary material appropriate?*

→ Yes, in this revised version, the supplementary material has been clarified and fit now the editorial guidelines.

*Benjamin Gérard*

---

## Referee Report (RR2)

**Response to the response to the review by Patrice Baby**

The paper of Habel et al. presents a structural study of two sites of the western Andes in Chile (20-22°S), where the authors use numerical trishear forward modelling to evaluate minimum horizontal shortening and analyse the kinematic evolution of two fault-related anticlines. Before being published, this paper must better document the structural observations, which are not always convincing (see below). These data can be used to construct in each area a balanced section to validate structural interpretations and calculate shortenings more rigorously. The authors need to better explain why they chose a fault propagation fold model rather than a tectonic inversion model in their structural interpretation.

These various issues on why we conservatively can't do much better on building cross-sections and calculating from there shortening, or the chosen structural interpretation (tectonic inversion vs. fault propagation fold) will be detailed hereafter. To summarize here in a few words:

- There are no constraints on the structure of the footwall of the faults inferred to generate the folds observed from surface geology.

This is the case in most balanced cross-sections. It is precisely the method of balanced cross-sections that makes it possible to propose viable interpretations. In figure 11, you nevertheless propose a footwall geometry !? The advantage of a balanced cross-section constructed along your entire section is that a total minimum shortening can be calculated, including that of the PT.

Given this, a more rigorous structural section and the associated shortening can't be proposed.

Not agree, see above

Because of this, we do only propose a possible interpretation and discuss its implications.
This is now better explained lines 222-225. To better separate observations from more interpretative outcomes, Figures 5c and 9c now only report field observations, with no inferences at depth - these come later, on Figure 11 when incorporating our trishear modeling results.

- We do not oppose tectonic inversion to fault-propagation folding. In fact, the investigated regions were forming basins during the Mesozoic, and these basins have been inverted during the Cenozoic to form the structures documented here along the Western flank of the Andes. This inversion most probably re-used faults that structured these basins, variably along-strike, and these inverted faults may have formed the fault propagation folds documented in our work. However, the main question relates here on whether these faults connect or not at depth onto a common detachment (as proposed by e.g., Victor et al 2004 and Armijo et al 2015), or whether they are independent from each other (as proposed in Fuentes et al 2018 and Martinez et al 2021, even though their later interpretation - in Fig 37.9 of Martinez and Fuentes 2022

- gets back to connecting these faults at depth onto an east-dipping detachment). Surface geology of the investigated field sites does not allow to choose between either one of these two models. This is now better explained in section 2.2.2 (lines 155-167) and 7.2.1 (lines 722-725). However, from regional considerations we favor the interpretation that these faults connect at depth onto a common detachment (see revised section 7.2.1, lines 722-731).

**You have chosen a model of thin-skinned tectonics - with a detachment in the sedimentary cover - which can in no way be associated with a tectonic inversion. Generally, tectonic inversion involves the basement and results in thick-skinned tectonics as in Figure 37.9 of Martinez and Fuentes.**

Finally, we recall that our proposed modeled structural interpretation and line-length balancing provide upper and lower bounds on shortening estimates, respectively, as now clarified lines 620-622.

**It's just the shortening of a fold that is not representative of the cross-section shortening.**

**GENERAL COMMENTS:**

The title must be modified. The studied areas are too small to represent the entire western Andes. It would be interesting to locate the two sites on a regional cross-section through the western Andes.

We agree with RC2 that our two areas are too small to represent the entire Western Andes - this is largely recognized throughout the manuscript. However, based on our findings from these two limited areas, we discuss the implications of these findings for shortening and kinematics of deformation at the scale of the entire Western Andes, from a regional reasoning.
The fact that we mention that our work is only "a contribution to the quantification of crustal shortening" does imply that we provide key additional data, not that we solve the problem at the scale of the entire Western Andes.
Title slightly modified to make it clearer: *A contribution to the quantification of crustal shortening and kinematics of deformation across the Western Andes ( ~20-22°S).*

**OK**

A regional general cross-section of the region added to Figure 1 would allow for structurally locating the investigated field sites. A section inspired from that shown in Figure 13 of Armijo et al 2015 would be appropriate. However, providing this section early in the manuscript may give the wrong impression that our interpretation is biased, and give room to unneeded criticism, even before we put forward our various arguments.

**Anyway, it is clear that your regional interpretation is guided by the work of Armijo et al.**

Later, this section would only be schematic as we only have field data from limited outcrops. Building a new section of the western Andean flank would

need more observations and discussions that are beyond the scope of the present paper which is focused on the frontal basement thrust (ABT) and structures just west of it. Our paper is already quite long and detailed. As such, we decided not to add and discuss such a large scale section.

**This is true and this is the problem. You draw important conclusions from little data.**

The stratigraphic and geologic background (2.2.1) needs a figure with a synthetic stratigraphic column.

This is already schematically provided in the legend of Figure 1, which corresponds to this part of the main text. As this is a very general regional overview, we do not see the need for more at this stage.

**Not agree. Difficult to have a regional view. No regional cross-section, no synthetic stratigraphic column!**

Structural and kinematic context (2.2.2):

In their last paper, Martinez and Fuentes (2022)(https://doiorg. insu.bib.cnrs.fr/10.1016/B978-0-323-85175-6.00037-7) show the importance of tectonic inversion of the Jurassic rift in this region. The analysed seismic sections are just west of the study areas of this paper and must be taken into consideration and discussed.

We already discussed the work and results of the group of Martinez and Fuentes, by citing their original work and data (Martinez et al 2021 and Fuentes et al 2018) rather than this later summarizing paper that does not provide any new interpretation of the seismic profiles.

**This paper is a Chapter of the Elsevier Book "Andean Structural Styles. A Seismic Atlas", where seismic interpretations have been validated by experts. The authors oh this Chapter write: "The inversion of these east-dipping faults produced the uplift of the hanging wall fault blocks and the development of west-verging asymmetrical anticlines with short and steeply dipping frontal limbs and large and gently dipping backlimbs (Figs. 37.6 and 37.7)". This is very similar to the anticlines in Fig. 11, especially the one in Fig. 11a which propagates on a steeply dipping fault.**

As mentioned earlier, tectonic inversion is not in opposition with our interpretation. In fact, the continental and marine Mesozoic series described in our field sites were deposited in basins, that were lately inverted during Cenozoic compression to form the structures of the Western flank of the Andes.

**You have chosen a model of thin-skinned tectonics - with a detachment in the sedimentary cover - which can in no way be associated with a tectonic inversion.**

The two main previous opposing views of these structures were about their geometries at depth. On one hand, Martinez et al 2021 and Fuentes et al 2018 propose that they are single planar faults not connected to each other at depth, minimizing shortening estimates. On the other hand, Victor et al 2004 and Armijo et al 2015 propose that these faults form a thrust system connected at depth onto a common decollement, dipping eastward beneath the Andes.

**You cannot compare with the deep geometries that are done on a regional scale and involve the basement.**

Seismic profiles to the west of our field sites are too poorly resolved at depth - let's be frank on this! –

**Not agree. Seismic profiles clearly show tectonic inversions.**

and field observations are too sparse and local to favor one or the other model. This is now clarified in the revised version of the manuscript (lines 155-167).

However, as already discussed in Victor et al 2004, and considered by Armijo et al 2015, only the thrust system model is able to provide a satisfactory structural framework for the large-scale structural organisation (deep basement to the east, shallower Mesozoic and Cenozoic units to the west) and topography (high to the east, low to the west, with a westward dipping continuous slope) of the Western Andes. Interestingly, in their latest interpretation, Martinez and Fuentes 2022 propose to connect these various faults at depth onto an east-dipping detachment (See their figure 37.9). To summarize, only from regional considerations can we favor the interpretation of a thrust system, as indicated in our final discussion (section 7.2.1, slightly modified to clarify this reasoning, and separate local observations from regional interpretations).

**I don't really understand your reasoning. At the mountain scale, inverted faults are part of the thrust system and often control deep basement structure uplifts.**

**Data and structural observations:**

It is important to better document the field data. For example, it is necessary to localise the field structural data in the structural map of Figure 4 to validate the cross-sections construction and structural interpretations.

All field data are reported on the map of Figure 4 and on the field sections of Figure 5. We cannot place on the map all measurements for obvious readability reasons, this is why we represented (some of) them along the field sections of Figure 5 - a section that is located on the map. We remind here that additional field observations are located on the map of Figure S14. We now report some of our structural measurements (either from field measurements or extracted from 3D mapping) to the structural map of Figure 4. Those now reported are meant to illustrate the asymmetry of the documented folds.

**We would like also to underline the fact that field data do not resume, to our sense, only to numerous analytical field measurements of strike and dip angles, but also encompass large-scale landscape observations - illustrated in the various field pictures provided in the main manuscript and in its supplement.**

Field pictures interpretations must be also validated by field data. These field data, as structural dip measurements, must be placed on the pictures. I am not at all convinced by the structural interpretation of the picture in Figure 7b. I can't really see the axis of the anticline of the Quebrada Tambillo, which is a key element of the structural interpretation. This picture interpretation must be absolutely validated by field measurements.

**We chose to show field pictures to depict the various observed structures, for instance with the general trend in dip angles on either side of axial planes. This is to our sense much more meaningful here than numerous local field measurements for our interpretations at the scale of the field sites. Landscape views are true field data, even though qualitative. In figure 7b for instance, the axis of the anticline is defined by the change in the dip orientation of layers, with layers dipping to the east (left of the figure) on one side and layers dipping to the west (right on the figure) on the other side, whatever the absolute values of the dip angles, and whatever the number of dip angle measurements.**

**We are in the 21st century, and structural cross-sections are no longer constructed from landscapes, but from robust quantitative structural data, and using 3D software.**

We added additional information on the field pictures, whenever appropriate and meaningful, either from direct field measurements, or from the projection of mapped 3D layers on satellite images and DEMs.

The structural map of the Quebrada Blanca zone shows structural dips values, which is not the case for the structural map of the Pinchal area. I understand that strike and dip measurements are extracted from 3D mapping. These 3D mapping and data extraction must be documented with some detailed illustrations.

**See our previous answers about representing structural measurements on the map of Figure 4 (Pinchal Zone). A detailed illustration of the structural inferences that can be made from 3D maps is already provided in the surface section of Figure 5a.**

**Structural interpretations:**

I don't understand why the authors didn't try to construct balanced cross-sections, the best way for thrust system modelling and calculation of shortening. The proposed interpretations are not geometrically validated. The footwalls of the thrusts have not been constructed (?).

The footwalls of the thrusts are not drawn in our sections of Figures 5 and 9, as we recognize not to have any indication on their structure. The structure used to build our trishear models is just a proposition, as indicated in the text. As such no definitive balanced section can be honestly proposed just from our field observations.

**You could have made the same proposition using the balanced cross-section method, but at a more regional scale, which would have allowed you to validate it or not.**

This is now better explained in the text (lines 222-225, 531-532, 553-554)

The authors propose a model of fault propagation fold (or fault bend fold (?)) for each section. Why? Why not a tectonic inversion? How do you explain such a steep frontal ramp in the cross-section of Figure 9C? Are there lithologies compatible with the levels of detachment?

As already indicated, tectonic inversion is not to be opposed to fault-propagation folding (see previous answers). We agree that structural inheritance from the earlier Andean Basins may have played an important role in localizing the thrusts at depth, or in controlling their steep geometry. This said, the observed folding is most simply explained by fault-related folding. Fault propagation folding is favored here as the faults do not reach the surface, and as small scale folds at the front of the western anticlines are possibly indicative of disharmonic folding (and therefore internal deformation) at the tip of an upward propagating ramp.

**These are not valid arguments.**

As of the lithologies at depth that would favor a decollement level, as can be deduced from our cross-sections (Figures 5c, 9c and 11), these stratigraphic levels do not crop out in the investigated areas and we cannot tell more on the subject.

**You could have deduced this from the regional sedimentary thicknesses.**

Our manuscript has been re-organized so as to better clarify these points. We now better separate field observations (revised sections 4 and 5, and updated figures 5c and 9c), modeling and local kinematic considerations (section 6, with Figure 11) and our choices of interpretations from regional considerations (in particular in section 7.2). But in any case this does not question our final conclusions and the field data provided.

The calculation of shortening is confusing ("Folding" + "Folding + thrusting"(?)).
We moved to section 6 all the structural reasoning leading to shortening calculations. By having all this together in one section, we hope that these various results are much more understandable now.
The discussion would require an integration of results in a regional cross-section through the Western Andes.

**This has already been discussed previously.**

**If you don't want to introduce the regional cross-section at the beginning, you could have at least put it at the end.**

**My detailed comments are highlighted in the attached pdf version.**

Please also note the supplement to this comment:
https://egusphere.copernicus.org/preprints/2022/egusphere-2022-629/egusphere-2022-6 29-
RC2-supplement.pdf
**We hereafter copy the various comments found throughout the reviewer's annotated PDF.**
**Please note that some text sections were highlighted but not commented by the reviewer.**

Line 1: The studied areas are too small to represent the entire western Andes unit
**See previous answer to the comment on the title**
Modified to ""investigated field sites" (line 14)

Line 17: They are not really restored
**Right, since we do not model the entire sections.**
This has been corrected to "From our interpreted sections " (line 17)

Lines 47-49: Add: Baby et al., 1997
https://doi.org/10.1130/0091-7613(1997)025<0883:NSCTCT>2.3.CO;2
and Rochat et al., 1999
https://www.researchgate.net/publication/233713934_Crustal_balance_and_control_of_the_erosive_and_sedimentary_processes_on_the_Altiplano_formation
Lines 101-102: Baby et al. (1997):
https://doi.org/10.1130/0091-7613(1997)025<0883:NSCTCT>2.3.CO;2
Lines 104-106: Baby et al. (1997):
https://doi.org/10.1130/0091-7613(1997)025<0883:NSCTCT>2.3.CO;2
**We recall that RC1 recommended to limit here the citations only to the few most significant ones among the numerous possible publications from the extensive literature on the subject of Andean shortening across the whole range!**

**It would have been interesting to diversify the authors for once.**

Figure 2: It would be appropriate to represent the Pinchal thrust on both images.

**We agree that this could be a good idea, even though the aim of this figure is to present first-order observations on the stratigraphic organization at the scale of the landscape. We tried to add the Pinchal Thrust but we did not find a satisfactory solution that would not load too much the figure - in particular when only keeping only one of the two field pictures as suggested by RC1. Therefore, this suggestion was not implemented.**

**As with my other suggestions!**

Figure 3: It is hard to imagine that there are no more precise ages in the bibliography with all these fossils and volcanic levels. It would be appropriate to indicate on the figure more precise ages such as 27-29 Ma at the base of the Cenozoic, or the Triassic and Jurassic (Majala o Chacarilla Fm?).

**We recall that the Pinchal zone is really remote and not easily accessible. The only geological map of the area dates back from the early 1980's (Skarmeta and Marinovic, 1981) and we believe from our field experience of this site that their stratigraphic correlation was done from an a priori knowledge of the regional stratigraphy rather by actual dating of the local layers.**

**Precise dating remains to be done here and we agree that there is a good potential for this. The SERNAGEOMIN (Geological and Mining service of Chile) will be working on this most probably over the next years. This is out of the scope of this manuscript.**

Figure 3: Locate the décollement of the cross-section in Figure 5c?

**This decollement is a structural interpretation proposed from our field observations, not a field observation and should therefore not be placed on the stratigraphic column of Figure 3.**

**Also, as suggested by our sections (former Figure 5c,** now Figure 11ab-c)**, the proposed decollement level does not crop out- and cannot therefore be represented on the log where only the series observed in the field are reported.**

We corrected the caption of Figure 3 to emphasize the fact that only field observations (at the surface) are reported.

Figure 5c: I am not convinced by this anticline. Field data are lacking

**See comment above about field data and lanscape views.**

**See my response**

Figure 5c: This thrust geometry is very arbitrary. The footwall of the thrust must be constructed to geometrically validate the interpretation. Is there a lithology compatible with this level of décollement? Why not tectonic inversion?

To clarify our point, we now better separate field observations, sub-surface and deeper interpretations. Here, we now only report sub-surface observations on Figure 5c, and leave deeper interpretations for Figure 11.

**As already mentioned, we have no direct clue about the structure of the footwall from field geology or even geophysics - this is why we had left it blank. Also, we already explained earlier why the tectonic inversion hypothesis is not to be opposed to fault related folding.** This has been further clarified in the text

Figure 6: You must indicate values of the structural dips collected on the field and show them on the photo. Scale?

We added an approximate scale on Figure 6a **- there is already a scale (Swiss knife) on the picture of Figure 6b). Dip measurements could be added on Figure 6a, but would not be meaningful here, at this scale, in the highly deformed footwall of the Pinchal Thrust**.

**This is not satisfactory**

We only added this information on Figure 6b.

Figure 7: You must indicate values of the structural dips collected on the field and show them on the photo. Scale?
We added dip angles, either measured in the field or deduced from 3D mapping on Figure 7. A scale is also now reported, it is approximative because of perspective effects (indicated in figure caption).

Figure 9c: I'm not really convinced by this part. Lack of field data!

**Well... the small-scale folds are illustrated in the field picture of Figure 10b.... as indicated in the boxes of Figures 9a-b. To our sense, not much to be added here without a more precise comment.**

**Lack of quantitative field data!**

Figure 9c: This thrust geometry is very arbitrary. The footwall of the thrust must be constructed to geometrically validate the interpretation. Are there lithologies compatible with these level of detachment? Why not tectonic inversion? How do you explain such a steep ramp?
As indicated in our previous responses, we better separate in our revised manuscript surface observations and sub-surface (folds) deductions - as reported here, in particular in revised figure 9c - from deeper interpretations (thrust geometries) - as now only reported in Figure 11.
**See previous answers about tectonic inversion and the geometry of faults at depth.**

**How do you explain such a steep ramp?**

Figure 10: I'm not really convinced by this part. Lack of field data!

**Same answer as for the comments on Figures 5c and 7 above, about the frontal anticline of the Pinchal zone. Also this anticline (Chacarilla anticline) has already been documented in previous published (cited) work: Blanco and Tomlinson 2001, Armijo et al 2015, Fuentes et al 2018.**
We added some structural measurements, following comments on other field pictures.

Figure 11a: How do you explain such a steep ramp?

**Steep ramps are commonly found, in particular in the case of inherited previous structures. This is probably the case here as Mesozoic layers were deposited initially in extensional basins. Here "steep" is only ~40-45°, as reported in table S3... so not that extremely steep. Such dip angle is derived from the geometry of the folded layers of the eastern limb of the Chacarilla anticline, and because of this should be here taken as a minimum.**

**I don't understand the argument**

Figure 11c: It is a fault bend fold!

We do not agree: it is a fault-propagation fold, as the fault has not yet reached the surface and deformation is disharmonic with layer thickening (and folding) at the tip of the ramp.
But we agree that this fault-propagation fold is not far from becoming a fault-bend fold with additional incremental slip and fault propagation.

**The staircase trajectory of the fault is typical of a fault bend fold**

Figure 11: How do you explain the difference in the depth of the detachment between the two cross-sections?

The two sections are ~70-80 km away from each other (Figure 1), with clearly differing stratigraphies, most probably controlled by the details in the earlier local Andean Basin (structural and stratigraphic inheritance). Also the Pinchal section is located in the immediate footwall of the Pinchal Thrust (Figure 5c), and we cannot rule out the possibility that the thrusts generating the observed folding of the Mesozoic layers are here shallower splays of the Pinchal thrust.
We added a few words on this (lines 609-615).

---

## Author Response (AR2)

Martine Simoes
Institut de physique du globe de Paris
1 rue Jussieu
75238 Paris cedex05
France
tel: +33 1 83 95 76 26
e-mail: simoes@ipgp.fr

Federico Rossetti
Topical Editor of *Solid Earth* (EGU)

Dear Editor,

Please find enclosed our re-revised manuscript, entitled "A contribution to the quantification of crustal shortening and kinematics of deformation across the Western Andes (~20-22°S)" by T. Habel and co-authors, submitted to the journal *Solid Earth*.

We thank Benjamin Gerard (RC1) and Patrice Baby (RC2) for their evaluation of our previously revised manuscript. RC1 has greatly appreciated our revisions and provided a very positive feedback. We note that such is still not the case for RC2, unfortunately, but at this stage we do have the feeling that the raised remarks are part of the classical scientific debate, with agreements and unavoidable disagreements, which allows science to move forward. We remind here that all reviews and responses to these reviews are public in the case of the journal *Solid Earth*, allowing any reader to make his/her own opinion from the manuscript and from the various arguments of the discussion.

To answer remaining critics from RC2, we verified that all needed nuances were well reported in the manuscript, wherever needed. We also now provide a schematic regional cross-section of the Western Andes (new Figure 12) to help the reader locate and place field observations and results at the scale of the discussed region.

Instead of replying to each remark by RC2, point by point, we hereafter provide a synthetic answer to his various comments by grouping them (various subjects annoted A to E). RC2's full review is provided afterwards and the various discussion groups (A to E) are indicated to the left of any one of his remarks. We believe that the discussion will be more easy to follow this way.

We also provide a marked-up manuscript specifying our slight revisions. Revisions to our manuscript are highlighted in yellow in this document. The highlighted figure caption indicates the figure that has been added (Figure 12: schematic regional cross-section).

We hope that you'll find now our manuscript suitable for publication in Solid Earth.

Martine SIMOES (on behalf of all co-authors)

**Response to the review by Patrice Baby (RC2)**

**A) Tectonic inversion**

RC2 suggests that we still disregard the possibility of tectonic inversion to interpret our sections across the Quebrada Blanca and Pinchal zones and discuss their regional implications. Here the argument is that we propose to root the faults at the base of the sedimentary cover and not into basement, as tectonic inversion implies thick-skinned tectonics.

We recall that surface geology (which is the same for all authors, with folds where they outcrop) and seismic profiles (where surface geology is not available) do to allow for documenting precisely the geometry of the underlying faults, even less to reveal how deep they root. This is most probably clearer for readers in the case of what can be said from surface geology, as illustrated in this study. It is however also the case from seismic profiles, if the reader tries to forget about interpretative lines drawn by various authors in published work. We do have the raw data in hand, and definitely nothing can be rigorously said for layers and possible faults beneath the well-layered less deformed (most probably Cenozoic) shallow levels - at least we would not dare to do so.

Therefore it is not possible from existing data, to our sense, to discriminate whether tectonics are here thick- or thin-skinned, as stated systematically thoughout the text where we report the most simple possible fault geometry (favored in our study) but conservatively acknowledge that we cannot discard the possibility that faults are steeper and/or root deeper.

However, when reasoning is done at a regional scale, we agree with the idea already put forward by previous authors (Victor et al 2004, Armijo et al 2015), that an east-dipping master fault is needed to generate and sustain the overall stratigraphic and structural organization as well as the continuous westward rising topographic slope, with shallow young levels at low altitudes to the west, and deeper older levels at higher altitudes to the east. Whether this east-dipping master fault is rooted in basement or at the base of the sedimentary cover is not constrained by any data.

We do not disregard tectonic inversion, as we agree that the faults that structured the previous Andean Basins have most probably been re-activated during the Andean orogeny and controlled subsequent compressive tectonics. However, we can not provide tight conclusions on how deep these faults are rooted. Our structural local models are the most simple local solutions - as acknowledged throughout the text -, we build our reasoning on them but conservatively consider them as an upper bound.

**B) Data**

RC2 considers most often that our sections are not documented by data - or enough data. Landscapes views are even considered to be an old-fashioned way to do structural geology.

We do not consider that our study would have been more rigorous and better documented with even more strike and dip measurements. These measurements were done in the field where accessible - and where appropriate. In addition, we consider that extracting structural data from high-resolution satellite imagery and derived-DEMs, through 3D-modeler tools, is quite a modern way to do structural geology ;)

**C) Building fully balanced sections, and estimating shortening**

RC2 keeps encouraging us to build fully-balanced sections to properly determine shortening estimates. Also RC2 questions the high dip angle of the proposed ramps, and the fact that Quebrada Blanca is not a fault-propagation fold but a fault-bend fold.

We insist here, once more, that this cannot be done without proper constraints on the geometries of layers at depth, in particular within the footwall of the inferred faults. Because of this, we previously revised our manuscript to clearly separate field and surface observations (Figures 5 and 9) from our proposed sections (Figure 11), which are based on strong hypotheses (and not data!) on the faults' footwalls - and are therefore only *possible* solutions, not *the* solutions. We verified throughout the text that these nuances were clearly stated.

We therefore estimate minimum shortening from folding only - a strict minimum that is independent on the inferences that can be made at depth on the faults' geometries or on slip on these faults - , and a maximum view on this shortening from the most simple structural solutions that can be honestly proposed from surface geology (Figure 11).

The dip angles of the faults underlying the Quebrada Blanca and Pinchal anticlines are at least equal to the dip angles of the backlimb layers of these folds - or steeper. We do not discuss why these faults are at least that steep -  this is not the scope of our work as we do not have data to specifically discuss this - , we just document the possible geometries from structural geology.

As of the Quebrada Blanca, we agree that this structure resembles that of a fault-bend fold. However, there is limb rotation at the front (as observed from tilted Cenozoic growth layers) and the fault does not yet reach the surface. We therefore keep on a fault-propagation fold interpretation, however recognizing that this structure has almost sufficiently evolved to a more mature fault-bend fold, as expected with increasing cumulative shortening (Bernard et al, 2007).

**D) Regional implications of our results**

RC2 recommends a regional cross-section at the beginning or at the end of the manuscript. Also, he finds that we build strong inferences on little data at this regional scale.

Building our own regional structural section would require that we re-evaluate in detail previously published sections, in particulary further west where only seismic profiles are available. Additionnally, as already mentioned, these seismic profiles are not sufficient to resolve how deep the needed east-dipping master fault beneath the Western Cordillera roots. Such section would unavoidably bring uneeded passionate debate.

Therefore, at this stage, we honestly cannot build a regional balanced section. However, we recognize that a regional section, even schematic would be helpful for the reader to get a better sense of the structural context and to understand better the changes in scale from local field results to a more regional reasoning. Rather than a balanced section, we therefore propose a structural sketch at the scale of of the western flank of the Andes in Northern Chili (new figure 12). Given the various unknowns, a sketch is most probably a more honest solution than a balanced section.

Such sketch will helpfully help better understand our reasoning and the changes in scale, and clarify the separation between data and inferences. Data are only provided where there are outcrops - and in this case, only about surface geology. We build upon these local data to propose inferences at the regional scale. These inferences are based on strong hypotheses and should be considered as such, as mentioned throughout the text.

**E) Not answered**

RC2 expresses at times his disagreement, however without providing further arguments. In this case, we cannot discuss and provide answers to foster discussion. At places, we also note some disrespectful remarks.
These various points were/could not be answered.

**Response to the response to the review by Patrice Baby**

The paper of Habel et al. presents a structural study of two sites of the western Andes in Chile (20-22°S), where the authors use numerical trishear forward modelling to evaluate minimum horizontal shortening and analyse the kinematic evolution of two fault-related anticlines. Before being published, this paper must better document the structural observations, which are not always convincing (see below). These data can be used to construct in each area a balanced section to validate structural interpretations and calculate shortenings more rigorously. The authors need to better explain why they chose a fault propagation fold model rather than a tectonic inversion model in their structural interpretation.

These various issues on why we conservatively can't do much better on building cross-sections and calculating from there shortening, or the chosen structural interpretation (tectonic inversion vs. fault propagation fold) will be detailed hereafter. To summarize here in a few words:

- There are no constraints on the structure of the footwall of the faults inferred to generate the folds observed from surface geology.

**This is the case in most balanced cross-sections. It is precisely the method of balanced cross-sections that makes it possible to propose viable interpretations. In figure 11, you nevertheless propose a footwall geometry !? The advantage of a balanced cross-section constructed along your entire section is that a total minimum shortening can be calculated, including that of the PT.**

Given this, a more rigorous structural section and the associated shortening can't be proposed.

**Not agree, see above**

Because of this, we do only propose a possible interpretation and discuss its implications.
This is now better explained lines 222-225. To better separate observations from more interpretative outcomes, Figures 5c and 9c now only report field observations, with no inferences at depth - these come later, on Figure 11 when incorporating our trishear modeling results.

- We do not oppose tectonic inversion to fault-propagation folding. In fact, the investigated regions were forming basins during the Mesozoic, and these basins have been inverted during the Cenozoic to form the structures documented here along the Western flank of the Andes. This inversion most probably re-used faults that structured these basins, variably along-strike, and these inverted faults may have formed the fault propagation folds documented in our work. However, the main question relates here on whether these faults connect or not at depth onto a common detachment (as proposed by e.g., Victor et al 2004 and Armijo et al 2015), or whether they are independent from each other (as proposed in Fuentes et al 2018 and Martinez et al 2021, even though their later interpretation - in Fig 37.9 of Martinez and Fuentes 2022

**C**

- gets back to connecting these faults at depth onto an east-dipping detachment). Surface geology of the investigated field sites does not allow to choose between either one of these two models. This is now better explained in section 2.2.2 (lines 155-167) and 7.2.1 (lines 722-725). However, from regional considerations we favor the interpretation that these faults connect at depth onto a common detachment (see revised section 7.2.1, lines 722-731).

**A** **You have chosen a model of thin-skinned tectonics - with a detachment in the sedimentary cover - which can in no way be associated with a tectonic inversion. Generally, tectonic inversion involves the basement and results in thick-skinned tectonics as in Figure 37.9 of Martinez and Fuentes.**

Finally, we recall that our proposed modeled structural interpretation and line-length balancing provide upper and lower bounds on shortening estimates, respectively, as now clarified lines 620-622.

**C** **It's just the shortening of a fold that is not representative of the cross-section shortening.**

**GENERAL COMMENTS:**

The title must be modified. The studied areas are too small to represent the entire western Andes. It would be interesting to locate the two sites on a regional cross-section through the western Andes.

We agree with RC2 that our two areas are too small to represent the entire Western Andes - this is largely recognized throughout the manuscript. However, based on our findings from these two limited areas, we discuss the implications of these findings for shortening and kinematics of deformation at the scale of the entire Western Andes, from a regional reasoning.
The fact that we mention that our work is only "a contribution to the quantification of crustal shortening" does imply that we provide key additional data, not that we solve the problem at the scale of the entire Western Andes. Title slightly modified to make it clearer: *A contribution to the quantification of crustal shortening and kinematics of deformation across the Western Andes ( ~20-22°S).*

**OK**

A regional general cross-section of the region added to Figure 1 would allow for structurally locating the investigated field sites. A section inspired from that shown in Figure 13 of Armijo et al 2015 would be appropriate. However, providing this section early in the manuscript may give the wrong impression that our interpretation is biased, and give room to unneeded criticism, even before we put forward our various arguments.

**D** **Anyway, it is clear that your regional interpretation is guided by the work of Armijo et al.**

Later, this section would only be schematic as we only have field data from limited outcrops. Building a new section of the western Andean flank would

need more observations and discussions that are beyond the scope of the present paper which is focused on the frontal basement thrust (ABT) and structures just west of it. Our paper is already quite long and detailed. As such, we decided not to add and discuss such a large scale section.

**D** **This is true and this is the problem. You draw important conclusions from little data.**

The stratigraphic and geologic background (2.2.1) needs a figure with a synthetic stratigraphic column.

This is already schematically provided in the legend of Figure 1, which corresponds to this part of the main text. As this is a very general regional overview, we do not see the need for more at this stage.

**D** **Not agree. Difficult to have a regional view. No regional cross-section, no synthetic stratigraphic column!**

Structural and kinematic context (2.2.2):

In their last paper, Martinez and Fuentes (2022)(https://doiorg. insu.bib.cnrs.fr/10.1016/B978-0-323-85175-6.00037-7) show the importance of tectonic inversion of the Jurassic rift in this region. The analysed seismic sections are just west of the study areas of this paper and must be taken into consideration and discussed.

We already discussed the work and results of the group of Martinez and Fuentes, by citing their original work and data (Martinez et al 2021 and Fuentes et al 2018) rather than this later summarizing paper that does not provide any new interpretation of the seismic profiles.

**A** **This paper is a Chapter of the Elsevier Book "Andean Structural Styles. A Seismic Atlas", where seismic interpretations have been validated by experts. The authors oh this Chapter write: "The inversion of these east-dipping faults produced the uplift of the hanging wall fault blocks and the development of west-verging asymmetrical anticlines with short and steeply dipping frontal limbs and large and gently dipping backlimbs (Figs. 37.6 and 37.7)". This is very similar to the anticlines in Fig. 11, especially the one in Fig. 11a which propagates on a steeply dipping fault.**

As mentioned earlier, tectonic inversion is not in opposition with our interpretation. In fact, the continental and marine Mesozoic series described in our field sites were deposited in basins, that were lately inverted during Cenozoic compression to form the structures of the Western flank of the Andes.

**A** **You have chosen a model of thin-skinned tectonics - with a detachment in the sedimentary cover - which can in no way be associated with a tectonic inversion.**

The two main previous opposing views of these structures were about their geometries at depth. On one hand, Martinez et al 2021 and Fuentes et al 2018 propose that they are single planar faults not connected to each other at depth, minimizing shortening estimates. On the other hand, Victor et al 2004 and Armijo et al 2015 propose that these faults form a thrust system connected at depth onto a common decollement, dipping eastward beneath the Andes.

**A,D** **You cannot compare with the deep geometries that are done on a regional scale and involve the basement.**

Seismic profiles to the west of our field sites are too poorly resolved at depth - let's be frank on this! –

**A** **Not agree. Seismic profiles clearly show tectonic inversions.**

and field observations are too sparse and local to favor one or the other model. This is now clarified in the revised version of the manuscript (lines 155-167).

However, as already discussed in Victor et al 2004, and considered by Armijo et al 2015, only the thrust system model is able to provide a satisfactory structural framework for the large-scale structural organisation (deep basement to the east, shallower Mesozoic and Cenozoic units to the west) and topography (high to the east, low to the west, with a westward dipping continuous slope) of the Western Andes. Interestingly, in their latest interpretation, Martinez and Fuentes 2022 propose to connect these various faults at depth onto an east-dipping detachment (See their figure 37.9).
To summarize, only from regional considerations can we favor the interpretation of a thrust system, as indicated in our final discussion (section 7.2.1, slightly modified to clarify this reasoning, and separate local observations from regional interpretations).

**A,D** **I don't really understand your reasoning. At the mountain scale, inverted faults are part of the thrust system and often control deep basement structure uplifts.**

**Data and structural observations:**

It is important to better document the field data. For example, it is necessary to localise the field structural data in the structural map of Figure 4 to validate the cross-sections construction and structural interpretations.

All field data are reported on the map of Figure 4 and on the field sections of Figure 5. We cannot place on the map all measurements for obvious readability reasons, this is why we represented (some of) them along the field sections of Figure 5 - a section that is located on the map. We remind here that additional field observations are located on the map of Figure S14.
We now report some of our structural measurements (either from field measurements or extracted from 3D mapping) to the structural map of Figure 4. Those now reported are meant to illustrate the asymmetry of the documented folds.

We would like also to underline the fact that field data do not resume, to our sense, only to numerous analytical field measurements of strike and dip angles, but also encompass large-scale landscape observations - illustrated in the various field pictures provided in the main manuscript and in its supplement.

Field pictures interpretations must be also validated by field data. These field data, as structural dip measurements, must be placed on the pictures. I am not at all convinced by the structural interpretation of the picture in Figure 7b. I can't really see the axis of the anticline of the Quebrada Tambillo, which is a key element of the structural interpretation. This picture interpretation must be absolutely validated by field measurements.

We chose to show field pictures to depict the various observed structures, for instance with the general trend in dip angles on either side of axial planes. This is to our sense much more meaningful here than numerous local field measurements for our interpretations at the scale of the field sites. Landscape views are true field data, even though qualitative. In figure 7b for instance, the axis of the anticline is defined by the change in the dip orientation of layers, with layers dipping to the east (left of the figure)
on one side and layers dipping to the west (right on the figure) on the other side, whatever the absolute values of the dip angles, and whatever the number of dip angle measurements.

**B,E** **We are in the 21st century, and structural cross-sections are no longer constructed from landscapes, but from robust quantitative structural data, and using 3D software.**

We added additional information on the field pictures, whenever appropriate and meaningful, either from direct field measurements, or from the projection of mapped 3D layers on satellite images and DEMs.

The structural map of the Quebrada Blanca zone shows structural dips values, which is not the case for the structural map of the Pinchal area. I understand that strike and dip measurements are extracted from 3D mapping. These 3D mapping and data extraction must be documented with some detailed illustrations.

See our previous answers about representing structural measurements on the map of Figure 4 (Pinchal Zone). A detailed illustration of the structural inferences that can be made from 3D maps is already provided in the surface section of Figure 5a.

**Structural interpretations:**

I don't understand why the authors didn't try to construct balanced cross-sections, the best way for thrust system modelling and calculation of shortening. The proposed interpretations are not geometrically validated. The footwalls of the thrusts have not been constructed (?).

The footwalls of the thrusts are not drawn in our sections of Figures 5 and 9, as we recognize not to have any indication on their structure. The structure used to build our trishear models is just a proposition, as indicated in the text. As such no definitive balanced section can be honestly proposed just from our field observations.

**D** **You could have made the same proposition using the balanced cross-section method, but at a more regional scale, which would have allowed you to validate it or not.**

This is now better explained in the text (lines 222-225, 531-532, 553-554)

The authors propose a model of fault propagation fold (or fault bend fold (?)) for each section. Why? Why not a tectonic inversion? How do you explain such a steep frontal ramp in the cross-section of Figure 9C? Are there lithologies compatible with the levels of detachment?

As already indicated, tectonic inversion is not to be opposed to fault-propagation folding (see previous answers). We agree that structural inheritance from the earlier Andean Basins may have played an important role in localizing the thrusts at depth, or in controlling their steep geometry. This said, the observed folding is most simply explained by fault-related folding. Fault propagation folding is favored here as the faults do not reach the surface, and as small scale folds at the front of the western anticlines are possibly indicative of disharmonic folding (and therefore internal deformation) at the tip of an upward propagating ramp.

**E** **These are not valid arguments.**

As of the lithologies at depth that would favor a decollement level, as can be deduced from our cross-sections (Figures 5c, 9c and 11), these stratigraphic levels do not crop out in the investigated areas and we cannot tell more on the subject.

**C** **You could have deduced this from the regional sedimentary thicknesses.**

Our manuscript has been re-organized so as to better clarify these points. We now better separate field observations (revised sections 4 and 5, and updated figures 5c and 9c), modeling and local kinematic considerations (section 6, with Figure 11) and our choices of interpretations from regional considerations (in particular in section 7.2). But in any case this does not question our final conclusions and the field data provided.

The calculation of shortening is confusing ("Folding" + "Folding + thrusting"(?)).
We moved to section 6 all the structural reasoning leading to shortening calculations. By having all this together in one section, we hope that these various results are much more understandable now.
The discussion would require an integration of results in a regional cross-section through the Western Andes.

**This has already been discussed previously.**

**D**  **If you don't want to introduce the regional cross-section at the beginning, you could have at least put it at the end.**

**My detailed comments are highlighted in the attached pdf version.**

Please also note the supplement to this comment:
https://egusphere.copernicus.org/preprints/2022/egusphere-2022-629/egusphere-2022-6 29-
RC2-supplement.pdf
**We hereafter copy the various comments found throughout the reviewer's annotated PDF.**
**Please note that some text sections were highlighted but not commented by the reviewer.**

Line 1: The studied areas are too small to represent the entire western Andes unit
**See previous answer to the comment on the title**
Modified to ""investigated field sites" (line 14)

Line 17: They are not really restored
**Right, since we do not model the entire sections.**
This has been corrected to "From our interpreted sections " (line 17)

Lines 47-49: Add: Baby et al., 1997
https://doi.org/10.1130/0091-7613(1997)025<0883:NSCTCT>2.3.CO;2
and Rochat et al., 1999
https://www.researchgate.net/publication/233713934_Crustal_balance_and_control_
of_the_erosive_and_sedimentary_processes_on_the_Altiplano_formation
Lines 101-102: Baby et al. (1997):
https://doi.org/10.1130/0091-7613(1997)025<0883:NSCTCT>2.3.CO;2
Lines 104-106: Baby et al. (1997):
https://doi.org/10.1130/0091-7613(1997)025<0883:NSCTCT>2.3.CO;2
**We recall that RC1 recommended to limit here the citations only to the few most significant ones among the numerous possible publications from the extensive literature on the subject of Andean shortening across the whole range!**

**E**  **It would have been interesting to diversify the authors for once.**

Figure 2: It would be appropriate to represent the Pinchal thrust on both images.

**We agree that this could be a good idea, even though the aim of this figure is to present first-order observations on the stratigraphic organization at the scale of the landscape. We tried to add the Pinchal Thrust but we did not find a satisfactory solution that would not load too much the figure - in particular when only keeping only one of the two field pictures as suggested by RC1. Therefore, this suggestion was not implemented.**

**E**  **As with my other suggestions!**

Figure 3: It is hard to imagine that there are no more precise ages in the bibliography with all these fossils and volcanic levels. It would be appropriate to indicate on the figure more precise ages such as 27-29 Ma at the base of the Cenozoic, or the Triassic and Jurassic (Majala o Chacarilla Fm?).

**We recall that the Pinchal zone is really remote and not easily accessible. The only geological map of the area dates back from the early 1980's (Skarmeta and Marinovic, 1981) and we believe from our field experience of this site that their stratigraphic correlation was done from an a priori knowledge of the regional stratigraphy rather by actual dating of the local layers.**
**Precise dating remains to be done here and we agree that there is a good potential for this. The SERNAGEOMIN (Geological and Mining service of Chile) will be working on this most probably over the next years. This is out of the scope of this manuscript.**

Figure 3: Locate the décollement of the cross-section in Figure 5c?

**This decollement is a structural interpretation proposed from our field observations, not a field observation and should therefore not be placed on the stratigraphic column of Figure 3.**
**Also, as suggested by our sections (former Figure 5c,** now Figure 11ab-c)**, the proposed decollement level does not crop out- and cannot therefore be represented on the log where only the series observed in the field are reported.**
We corrected the caption of Figure 3 to emphasize the fact that only field observations (at the surface) are reported.

Figure 5c: I am not convinced by this anticline. Field data are lacking
**See comment above about field data and lanscape views.**

E   **See my response**

Figure 5c: This thrust geometry is very arbitrary. The footwall of the thrust must be constructed to geometrically validate the interpretation. Is there a lithology compatible with this level of décollement? Why not tectonic inversion?
To clarify our point, we now better separate field observations, sub-surface and deeper interpretations. Here, we now only report sub-surface observations on Figure 5c, and leave deeper interpretations for Figure 11.
**As already mentioned, we have no direct clue about the structure of the footwall from field geology or even geophysics - this is why we had left it blank. Also, we already explained earlier why the tectonic inversion hypothesis is not to be opposed to fault related folding.** This has been further clarified in the text

Figure 6: You must indicate values of the structural dips collected on the field and show them on the photo. Scale?
We added an approximate scale on Figure 6a **- there is already a scale (Swiss knife) on the picture of Figure 6b). Dip measurements could be added on Figure 6a, but would not be meaningful here, at this scale, in the highly deformed footwall of the Pinchal Thrust**.

E   **This is not satisfactory**

We only added this information on Figure 6b.

Figure 7: You must indicate values of the structural dips collected on the field and show them on the photo. Scale?

We added dip angles, either measured in the field or deduced from 3D mapping on Figure 7. A scale is also now reported, it is approximative because of perspective effects (indicated in figure caption).

Figure 9c: I'm not really convinced by this part. Lack of field data!

**Well... the small-scale folds are illustrated in the field picture of Figure 10b.... as indicated in the boxes of Figures 9a-b. To our sense, not much to be added here without a more precise comment.**

**B,E**  **Lack of quantitative field data!**

Figure 9c: This thrust geometry is very arbitrary. The footwall of the thrust must be constructed to geometrically validate the interpretation. Are there lithologies compatible with these level of detachment? Why not tectonic inversion? How do you explain such a steep ramp?

As indicated in our previous responses, we better separate in our revised manuscript surface observations and sub-surface (folds) deductions - as reported here, in particular in revised figure 9c - from deeper interpretations (thrust geometries) - as now only reported in Figure 11.

**See previous answers about tectonic inversion and the geometry of faults at depth.**

**C**  **How do you explain such a steep ramp?**

Figure 10: I'm not really convinced by this part. Lack of field data!

**Same answer as for the comments on Figures 5c and 7 above, about the frontal anticline of the Pinchal zone. Also this anticline (Chacarilla anticline) has already been documented in previous published (cited) work: Blanco and Tomlinson 2001, Armijo et al 2015, Fuentes et al 2018.**

We added some structural measurements, following comments on other field pictures.

Figure 11a: How do you explain such a steep ramp?

**Steep ramps are commonly found, in particular in the case of inherited previous structures. This is probably the case here as Mesozoic layers were deposited initially in extensional basins. Here "steep" is only ~40-45°, as reported in table S3... so not that extremely steep. Such dip angle is derived from the geometry of the folded layers of the eastern limb of the Chacarilla anticline, and because of this should be here taken as a minimum.**

**E**  **I don't understand the argument**

Figure 11c: It is a fault bend fold!

We do not agree: it is a fault-propagation fold, as the fault has not yet reached the surface and deformation is disharmonic with layer thickening (and folding) at the tip of the ramp.
But we agree that this fault-propagation fold is not far from becoming a fault-bend fold with additional incremental slip and fault propagation.

C    **The staircase trajectory of the fault is typical of a fault bend fold**

Figure 11: How do you explain the difference in the depth of the detachment between the two cross-sections?

The two sections are ~70-80 km away from each other (Figure 1), with clearly differing stratigraphies, most probably controlled by the details in the earlier local Andean Basin (structural and stratigraphic inheritance). Also the Pinchal section is located in the immediate footwall of the Pinchal Thrust (Figure 5c), and we cannot rule out the possibility that the thrusts generating the observed folding of the Mesozoic layers are here shallower splays of the Pinchal thrust.
We added a few words on this (lines 609-615).